# Enhanced Atmospheric Oxidation and Particle Reductions Driving Changes to Nitrate Formation Mechanisms across Coastal and Inland Regions of North China

Zhenze Liu[1,2], Jianhua Qi[1,2], Yuanzhe Ni[1], Likun Xue[3], Xiaohuan Liu[1,2]

[1] Key Laboratory of Marine Environment and Ecology, Ministry of Education, Ocean University of China, Qingdao 266100, China

[2] Laboratory for Marine Ecology and Environmental Science, Qingdao Marine Science and Technology Center, Qingdao 266237, China

[3] Environment Research Institute, Shandong University, Qingdao, Shandong, 266237, China

*Correspondence to*: Jianhua Qi (qjianhua@ouc.edu.cn), Xiaohuan Liu (liuxh1983@ouc.edu.cn)

**Abstract.**

Nitrate ($NO_3^-$) has surpassed sulfate as the dominant secondary inorganic ion, posing a significant challenge to air quality improvement measures in China. We utilized the WRF–CMAQ model and isotope

analysis to investigate the nitrate formation mechanisms driving regional changes in inland and coastal cities in North China during the winters of 2013 and 2018. Among the nitrate formation pathways, the oxidation reaction of OH radicals with $NO_2$ (OH + $NO_2$) and the heterogeneous reaction of $N_2O_5$ (het$N_2O_5$) were determined to be the dominant pathways (88%–95.5%), whereas the other pathways contributed less than 12.4% to the total amount of nitrate formation. In inland cities, 63.7%–85.6% of nitrate was

formed via OH + $NO_2$ and 8.3%–27.7% was formed from het$N_2O_5$. In coastal cities, approximately half of the nitrate (48.2%–56.5%) was produced from OH + $NO_2$, whereas het$N_2O_5$ contributed 37.0%–45.7% due to higher $N_2O_5$ concentrations and longer $NO_3$ radical lifetimes. Compared with that in 2013, the OH + $NO_2$ contribution in 2018 increased by 7.6% in inland cities and 3.6% in coastal cities due to the increased atmospheric oxidizing capacity. Scenario simulations indicated that a 60% reduction in $NO_x$

emissions led to a 4.5% decrease in nitrate concentrations in Beijing. The reduction reached 32.4% reduction in Qingdao. A 60% combined reduction in $NH_3$, $NO_x$, and VOCs yielded 44.2% and 60.0% reductions in nitrate in Beijing and Qingdao, respectively, underscoring the necessity of multipollutant control strategies.

**1 Introduction**

As a key component of fine particulate matter ($PM_{2.5}$), nitrate ($NO_3^-$) exacerbates health risks (Tie et al., 2009; Sun et al., 2014) and affects the physical and chemical properties of particulates, such as their hygroscopicity, light absorption, and acidity (Cao et al., 2013; Wang et al., 2023). These properties directly influence the atmospheric radiation balance (Ramanathan and Feng, 2009; Tegen et al., 2000), atmospheric visibility and air quality. Furthermore, nitrate serves as a key source of cloud condensation

nuclei, affecting cloud formation and precipitation patterns, which subsequently influence the global water cycle and climate regulation (Yu et al., 2020; Kalkavouras et al., 2019). Moreover, the photolysis process of nitrate in atmospheric boundary layers is highly active, serving as an important source of NOx and regulating the atmospheric oxidation capacity, thus influencing the formation of secondary pollutants such as sulfate ($SO_4^{2-}$) and brown carbon (BrC) (Ye et al., 2016; Xue et al., 2019; Zheng et al., 2020;

Yang et al., 2021). Therefore, nitrate is closely related to regional haze pollution occurrence (Zhai et al., 2021; Zhang et al., 2021; Xu et al., 2019a; Fu et al., 2020; Xu et al., 2019b). Understanding the formation mechanism of nitrate is essential not only for advancing the atmospheric chemical processes but also for mitigating regional haze pollution.

In tropospheric atmospheres, nitrate formation primarily follows two pathways (Figure 1). During

the daytime, $NO_2$ is oxidized by hydroxyl radicals (OH) to produce gaseous $HNO_3$ (R1). Conversely, at night, $O_3$ oxidizes $NO_2$, leading to the formation of the $NO_3$ radical, which combines with $NO_2$ to form $N_2O_5$. This compound can subsequently be adsorbed onto particles via heterogeneous reactions, resulting in the formation of $HNO_3$ (R2) (Hallquist et al., 1999; Pathak et al., 2011). These dynamics highlight the intrinsic dependence of atmospheric nitrate formation on NOx and oxidants, especially $O_3$ and OH

radicals. The generation of OH and $O_3$ is intricately linked to the photochemical reactions of NOx and volatile organic compounds (VOCs) (Atkinson, 2000). OH radicals and hydroperoxyl radicals ($HO_2$) are produced via the photolysis of ozone ($O_3$), nitrous acid (HONO), oxygenated volatile organic compounds (OVOCs), and hydrogen peroxide ($H_2O_2$), and via reactions between $O_3$ and VOCs. OH subsequently reacts with VOCs to generate organic peroxy radicals ($RO_2$) and $HO_2$, which are then recycled back to

OH via their interaction with nitrogen monoxide (NO). During this cycle, NO is transformed into $NO_2$, whichyields $O_3$ upon photolysis (Fu et al., 2020) .

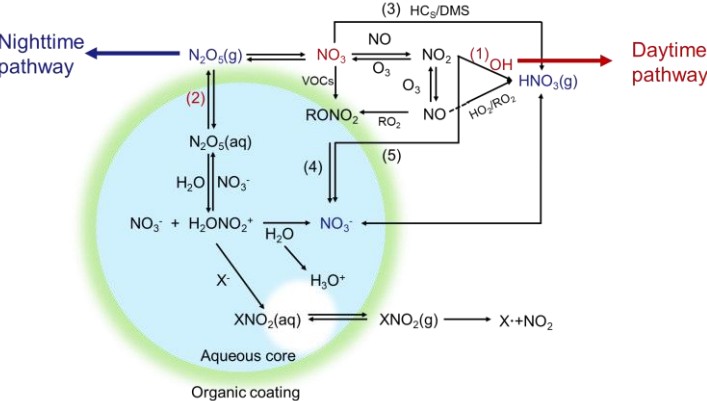

**Figure 1 Formation mechanisms of nitrate in the atmosphere (Wang et al., 2023)**

Nitrate formation mechanisms exhibit significant regional differences, especially in winter. Some studies based on isotope analysis have shown that the OH pathway is important for nitrate formation in inland cities. For example, in Beijing, the OH pathway contributes 66–92% to winter nitrate formation, whereas the heterogeneous reaction of the $N_2O_5$ (het$N_2O_5$) pathway contributes 8–34% to nitrate (Chen et al., 2020). In Xi'an, the contribution of the het$N_2O_5$ pathway to winter nitrate formation ranges from 13% to 35% (Wu et al., 2021). In coastal areas, which are influenced by high humidities, high sea salt levels and the combined effects of marine emissions and air masses (Zhong et al., 2023; Athanasopoulou et al., 2008; Zhao et al., 2024), the contribution of the OH pathway is smaller than that in inland cities, whereas the het$N_2O_5$ pathway plays a more significant role. For example, isotope analysis revealed that the contribution of the OH + $NO_2$ pathway to winter nitrate formation is 48–74% in Shanghai (He et al., 2020), whereas the contributions of the OH, het$N_2O_5$, and $NO_3$ + HC pathways to winter nitrate formation are 20.2%, 38.2%, and 21.6%, respectively, in Xiamen (Li et al., 2022). Broad coastal studies have revealed similar significant regional differences. Michalski et al. (2003) noted that in coastal California, approximately 90% of winter nitrate originates from the het$N_2O_5$ pathway. In the polar environment of GreenlandKunasek et al. (2008), nearly all nitrate is formed via the $NO_3$ + HC/DMS and het$N_2O_5$ pathways in winter, with contributions of 60% and 40%, respectively. Furthermore, the impacts of continental and marine air masses on nitrate formation mechanisms differ. In the South Yellow Sea, under the dominance of marine air masses, the contributions of the OH + $NO_2$, $NO_3$ + HC/DMS, and het$N_2O_5$ pathways to nitrate formation are 43.9%, 22.4%, and 33.6%, respectively, whereas the contributions (12.6%, 59.0%, and 28.4%, respectively) in the Bohai and North Yellow Seas are affected by continental pollution (Zhao et al., 2024). These studies have confirmed substantial regional differences in nitrate formation mechanisms.

In addition to isotope analysis, model simulation studies have provided significant support for elucidating nitrate formation mechanisms. Global models such as GEOS-Chem have revealed significant land–ocean differences in nitrate formation pathways. Alexander et al. (2020), using the GEOS-Chem model at a global scale, reported that the $OH + NO_2$ reaction (41%–42%) and $N_2O_5$ hydrolysis (28%–41%) were the main pathways involved below a 1 km altitude, with $N_2O_5$ hydrolysis dominating in winter in the mid to high-latitude northern continents; however, in remote marine regions, $XNO_3$ hydrolysis emerged as the primary pathway. However, owing to their relatively low resolution (typically 0.5°×0.625° or coarser) and simplified treatment of aerosol-surface heterogeneous reactions, global models clearly have limitations in capturing regional differences (Heald et al., 2012). Consequently, the average outputs from global models may not accurately reflect the nitrate formation mechanisms of specific regions (Alexander et al., 2020), leading to potential overestimations or underestimations of local nitrate formation, particularly in terms of ammonia emissions and nitrate production rates. Compared with global models, regional models, such as WRF–Chem and WRF–CMAQ, can offer greater accuracy in capturing local chemical processes. For example, Sun et al. (2022) used the WRF–CMAQ model to study the Yangtze River Delta (YRD) and reported that during winter, the $OH + NO_2$ reaction contributed 69.3% of nitrate formation in Shanghai (a coastal area) and 66.9% in Hefei (an inland area), whereas the contribution of $hetN_2O_5$ in Hefei was 27.1%. Similarly, Li et al. (2021) used the WRF–Chem model to investigate the NCP and YRD and reported that near the surface, the $OH + NO_2$ reaction dominated nitrate formation (60%–92%), with $hetN_2O_5$ contributing 8%–40%. Moreover, Kim et al. (2014b) applied the WRF–CMAQ model to the Great Lakes region of North America and reported that $hetN_2O_5$ accounted for 57% of nitrate formation, highlighting the regional variability. For targeted regions, especially heavily polluted urban areas or those with unique ecosystems, employing high-resolution regional simulations and more detailed chemical mechanisms allows for a more precise characterization of nitrate formation processes.

The North China Plain (NCP) region is a focal point for studying air pollution due to its severe winter haze issues, which consistently rank among the areas with the highest $PM_{2.5}$ concentrations globally (Chen et al., 2016). Before 2013, sulfate was the dominant pollutant during haze events in the NCP, accounting for a significant proportion of the total $PM_{2.5}$ (Li et al., 2019). However, following the implementation of the Clean Air Action Plan (CAAP) in 2013, which aimed to reduce $PM_{2.5}$

concentrations and improve air quality (Air Pollution Prevention and Control Action Plan, 2013), the

composition of winter haze pollution shifted from sulfate-dominated to nitrate-dominated (Li et al., 2019;

Xu et al., 2019a). Data from the Multiresolution Emission Inventory for China (MEIC) (Zheng et al.,

2018) show that from 2013 to 2017, anthropogenic emissions of $SO_2$, $NO_x$, CO, and $NH_3$ decreased by

59%, 21%, 23%, and -3%, respectively. Correspondingly, the national annual average $PM_{2.5}$ level

decreased by 30–50% from 2013–2018 (Zhai et al., 2019). While these emission reduction measures led

to significant declines in CO and $SO_2$ levels and in particulate sulfate concentrations, the reduction in

particulate nitrate levels was smaller than expected. In fact, nitrate mass concentrations have increased

in some major cities (Xu et al., 2019b; Shao et al., 2018; Zhou et al., 2019; Fu et al., 2020). This

divergence in the responses of sulfate and nitrate to emission controls underscores the complex chemistry

of nitrate formation and its precursors. Most studies have focused on nitrate formation mechanisms in

inland cities such as Beijing–Tianjin–Hebei (Fan et al., 2020b; Fu et al., 2020; Yang et al., 2024; Chen et

al., 2020); thus, the formation mechanisms in coastal cities remain poorly understood. The coastal regions

of China experience monsoon transitions, where marine air masses are transported inland under the

influence of sea-land breezes, affecting the formation of nitrate over terrestrial areas.

To understand the factors controlling nitrate concentration variations in inland and coastal cities,

especially during periods dominated by different primary pollutants, we selected the winters of 2013,

when the CAAP was initiated, and 2018, after the project had concluded, to explore the nitrate formation

mechanisms in five inland cities and two coastal cities on the NCP via a regional multiscale model

combined with isotope analysis. The primary objectives of this study were to quantify the contributions

of different nitrate formation pathways, such as $OH + NO_2$ and $hetN_2O_5$, in inland and coastal cities,

investigate the impacts of emission control measures on nitrate formation, and provide insights into

regional differences in nitrate control.

## 2 Methods

### 2.1 Model configuration

The Community Multiscale Air Quality (CMAQ) model (version 5.3.3) was used to simulate the

chemical reactions and physical processes that contributed to $NO_3^-$ and $TNO_3$ ($HNO_3 + NO_3^-$) formation

in the NCP region. Two distinct periods were considered, i.e., one period from December 1, 2013, to

February 28, 2014, representing the period when the CAAP was initiated, and one period from December

1, 2018, to February 28, 2019, representing the period after the CAAP was completed. As shown in

Figure 2, a dual-layer grid nesting method was implemented in our simulations. Domain 1 (D01)

encompassed the majority of China with a 36-km horizontal resolution, whereas domain 2 (D02) covered

the NCP with a 12km horizontal resolution. The major cities within the NCP included Beijing (BJ),

Tianjin (TJ), Shijiazhuang (SJZ), Jinan (JN), Zhengzhou (ZZ), Qingdao (QD), and Yantai (YT). The

terrain elevation data used in Figure 2 came from the GEBCO 2024 Grid provided by the GEBCO

Compilation Group (2024; doi:10.5285/1c44ce99-0a0d-5f4f-e063-7086abc0ea0f; retrieved from

https://www.gebco.net/data_and_products/gridded_bathymetry_data/). The model was vertically

segmented into 14 layers, stretching from the Earth's surface to the troposphere, with the first layer height

at approximately 31 m. The CB6 chemical mechanism was chosen to simulate gas-phase chemistry

(Luecken et al., 2019; Yan et al., 2021) and the aerosol mechanism. Aerosol module version 7 (AERO7,

Appel et al. (2020)) was used to simulate particulate-phase chemistry. The initial and boundary conditions

used as the inputs of D01 were provided via the GEOS-Chem global simulation results, and D02 was

provided via D01. To minimize the impact of the initial conditions, we initiated the model run 7 days

before the analysis period. Anthropogenic emission data were sourced from the Multiresolution

Emission Inventory for China(MEIC)source emission inventory maintained by Tsinghua University

(http://meicmodel.org/), while biogenic emission estimates were derived from the Model of Emissions

of Gases and Aerosols from Nature version 2.0.4 (MEGANv2.0.4, http://lar.wsu.edu/megan/; Guenther

et al., 2006).

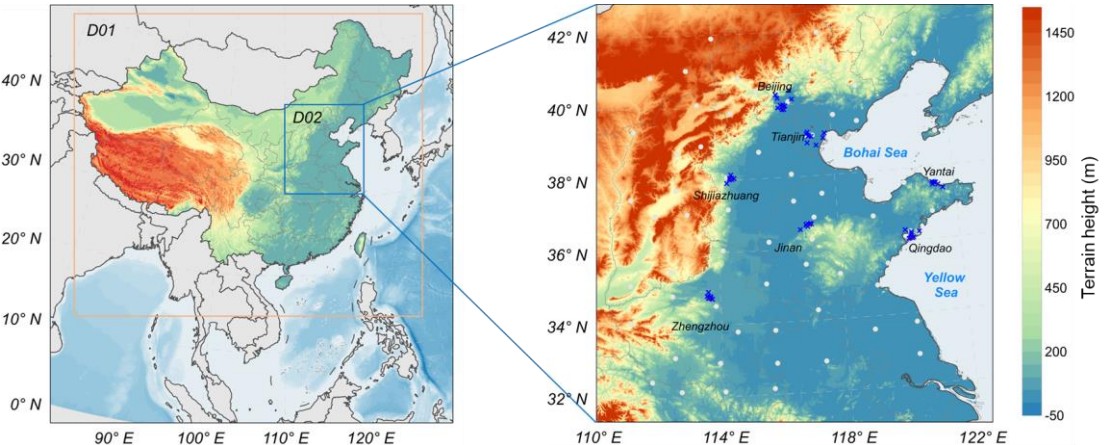

**Figure 2 Terrain heights of the NCP region and locations of the major cities (Beijing, Tianjin, Shijiazhuang,**
**Jinan, Zhengzhou, Qingdao, and Yantai) within the study domain. The white dots indicate meteorological**
**observation stations, and the blue crosses indicate air quality monitoring stations.**

The required meteorological data were generated with the Weather Research and Forecasting Model

(WRF) version 3.7, a system for predicting mesoscale weather patterns. The selection of physics options

for this model conformed with the methodologies applied in our earlier studies (Liu et al., 2021; Chen et

al., 2021). The WRF simulations in this study depended on the final operational global analysis (FNL)

datasets, which are global reanalysis data with temporal and spatial resolutions of 6 hours and $1° \times 1°$,

respectively, provided by the U.S. Environmental Prediction Center and the National Center for

Atmospheric Research (https://rda.ucar.edu/datasets/ds083-2/). The WRF model outputs were

subsequently processed via Model-3/CMAQ Modeling System Interface Processor (MCIP) version 4.3,

ensuring compatibility with the CMAQ model format.

**2.2 Process analysis**

Within the CMAQ modeling framework, the Integrated Reaction Rate (IRR) module of the Process

Analysis (PA) tool was used to simulate the formation reactions of $TNO_3$ ($HNO_3 + NO_3^-$). The reaction

rates of the chemical reactions at each moment were provided by the IRR module, enabling the

quantitative evaluation of the target reactions. The complex chemical formation of $TNO_3$ involved seven

reaction pathways, which were categorized into three main pathways on the basis of their significance:

$OH + NO_2$ (R1), $hetN_2O_5$ (R2), and other (R3 + R4 + R5 + R6 + R7). The chemical equations for these

seven production reaction pathways could be expressed as follows:

$$NO_2\,(g) + OH(g) \rightarrow HNO_3\,(g) \tag{R1}$$

$$N_2O_5 + H_2O(l) \rightarrow 2HNO_3(aq) \tag{R2}$$

$$N_2O_5(g) + H_2O(g) \rightarrow 2HNO_3\,(g) \tag{R3}$$

$$NO_3(g) + VOCs(g) \rightarrow HNO_3(g) \tag{R4}$$

$$2NO_2(g) + 2H_2O(l) \rightarrow HNO_3(aq) + HONO(aq) \tag{R5}$$

$$ClNO_3\,(g) + H_2O(l) \rightarrow HNO_3(aq) + HOCl\,(aq) \tag{R6}$$

$$RONO_2(g) + H_2O(l) \rightarrow HNO_3(aq) + ROH(l) \tag{R7}$$

**2.3 Observation data**

In this study, the simulation results, including meteorological parameters and atmospheric pollutants,

for the major cities in the NCP were validated in depth. The meteorological parameters included the 2-m

temperature (T2), relative humidity (RH), 10-m wind speed (WS10), and 10-m wind direction (WD10).

The atmospheric pollutants included the maximum 8-hour moving average ozone ($MDA8O_3$), $NO_2$,

PM$_{2.5}$, and PM$_{2.5}$ components.

### 2.3.1 Meteorological data and sources

Meteorological data, comprising T2, WS10, WD10, and RH, were sourced from the National Climatic Data Center (NCDC) of the National Oceanic and Atmospheric Administration (NOAA) (https://www.ncdc.noaa.gov/cdo-web/, last access: 17 October 2023). Data were collected from a total of 68 stations, with a temporal resolution of either 3 hours or 1 hour.

### 2.3.2 Atmospheric pollutant concentrations

Data on O$_3$, NO$_2$, and PM$_{2.5}$ concentrations were retrieved from the real-time national urban air quality dataset disseminated by the China Environmental Monitoring Center (https://air.cnemc.cn:18007/). The SO$_4^{2-}$, NH$_4^+$ and NO$_3^-$ concentration data were sourced from our own observations in Qingdao and published data (detailed information can be found in Table S3).

In Qingdao city, we collected total suspended particle (TSP) samples during two winter seasons. The TSP samples were collected on preheated quartz fiber filters with a high-volume (1.05 m$^3$ min$^{-1}$) aerosol sampler (Qingdao Laoshan Electronics Co., Ltd., China). The sampling site was situated on the roof of Darwin Hall (36°06′N, 120°33′E, 16 m) at Ocean University of China, approximately 1 km from the beach. Detailed information can be found in Ding et al. (2019). Water-soluble inorganic ions, such as NO$_3^-$, SO$_4^{2-}$, and NH$_4^+$, were extracted from the samples using ultrapure water (> 18.2 MΩ·cm), and their concentrations were measured with a Thermo Scientific Dionex ICS-1100 IC system, as described in a previous study (Qi et al., 2020). The ionic concentrations in the TSP samples were calibrated by subtracting the concentrations in the blanks. The δ$^{18}$O and δ$^{15}$N values of NO$_3^-$ in the TSP samples were determined via the bacterial denitrifier method (Casciotti et al., 2002; Sigman et al., 2001) with the Gasbench-IRMS system (Delta V model, Thermo Scientific). For analytical calibration, four international reference standards were employed, namely, USGS32, USGS34, USGS35, and IAEA-NO$_3^-$. The standard deviations of the replicates were ±0.2‰ for δ$^{15}$N–NO$_3^-$ and ±0.5‰ for δ$^{18}$O–NO$_3^-$. The analytical procedures for determining δ$^{15}$N–NO$_3^-$ and δ$^{18}$O–NO$_3^-$ were detailed in a previous study (Luo et al., 2021). The blank contributedless than 5% to the sample results. Additional details are provided in Supplementary Text S1.

**2.4 Dual-isotopic Bayesian mixing model**

The Bayesian isotope mixing model (Stable Isotope Analysis in R, SIAR) and dual-isotopic compositions ($\delta^{15}$N-NO$_3^-$ and $\delta^{18}$O-NO$_3^-$) determined in Qingdao and adapted from a reference study in Beijing (Zong et al., 2020; Fan et al., 2020b) were employed to estimate the contributions of OH radicals and the hetN$_2$O$_5$ pathway to particulate NO$_3^-$. Eqs. (S1) –(S7) for calculating the isotopic compositions of nitrogen oxides in the atmosphere are provided in the Supporting Information. The atmospheric $\delta^{15}$N–NO$_3^-$ and $\delta^{18}$O–NO$_3^-$ can be expressed by Eqs. (S1) and (S2), respectively. The end-members of [$\delta^{15}$N–

HNO$_3$]$_{OH}$, [$\delta^{15}$N–HNO$_3$]$_{N2O5}$, [$\delta^{18}$O–HNO$_3$]$_{OH}$ and [$\delta^{18}$O–HNO$_3$]$_{N2O5}$ can were expressed by Eqs. (S3), (S4), (S5) and (S6), respectively (Walters and Michalski, 2016). This study was focused on the seasonal scale, where observations of $\delta^{18}$O–NO$_3^-$ did not reveal a significant diurnal difference (Feng et al., 2020; Zhang et al., 2022), as shown in Figure S1. Therefore, we did not account for the diurnal variation in $\delta^{18}$O–NO$_2$ in the calculation of end-members (Walters et al., 2018; Zhang et al., 2025; Albertin et al.,

2024; Albertin et al., 2021). The $\delta^{15}$N values of tropospheric NOx and the $\delta^{18}$O values of tropospheric H$_2$O$_{(g)}$, NOx, O$_3$, and ·OH occurred within certain ranges, as described in Text S2 and Table S1. Therefore, the end-members of $\delta^{15}$N and $\delta^{18}$O for the two pathways could be estimated according to f$_{NO2}$ (the molar ratio of NO$_2$ and NOx) and the isotope fractionation values of nitrogen and oxygen, i.e., $\alpha_{NO2/NO}$, $\alpha_{OH/H2O}$, and $\alpha_{N2O5/NO2}$. The nitrogen and oxygen isotope fractionations were temperature dependent and could be

estimated via Eq. (S7) and Table S2. On the basis of the end-members of [$\delta^{15}$N-HNO$_3$]$_{OH}$, [$\delta^{15}$N-HNO$_3$]$_{N2O5}$, [$\delta^{18}$O-HNO$_3$]$_{OH}$ and [$\delta^{18}$O-HNO$_3$]$_{N2O5}$, the contribution of the OH radical formation pathway ($\gamma$) was estimated via the dual-isotopic Bayesian mixing model (Luo et al., 2021). The uncertainty of the Bayesian model mainly stemmed from the standard deviations in the end-member calculations for each formation pathway and the posterior probability distributions output by the model (Moore and Semmens,

240 2008).

**2.5 Model evaluation**

The following statistical indicators were used to evaluate the simulation effect: the mean deviation (MB), normalized mean deviation (NMB), normalized mean error (NME), correlation coefficient (R), root mean square error (RMSE), and index of agreement (IOA). The definitions and standards of all the

statistical indicators are provided in Table S3. To evaluate model performance, Emery et al. (2001) provided benchmarks of the above statistical indicators for major meteorological parameters such as T2,

WS10 and WD10, which are presented in Table 1. Similarly, Emery et al. (2017) and Huang et al. (2021) proposed benchmarks for the concentrations of major air pollutants, including $PM_{2.5}$, $NO_2$, $MDA8O_3$, and $NO_3^-$ , as comprehensively detailed in Table 2.

For the comparison between the model simulations and observational data, the model grid cell corresponding to the geographic coordinates of each observation station was first identified. At each station, the observed values were compared with the simulated values from the corresponding grid cell on a point-by-point, hour-by-hour basis. These paired comparisons were then aggregated across all stations, and statistical evaluation metrics were generated from the entire dataset to assess model

performance comprehensively. All data processing and comparisons were conducted on an hourly basis.

**2.6 Emission reduction scenario simulation design**

     Notably, emission reduction scenarios were specifically designed to investigate how reductions in $NH_3$, $NO_x$, and VOCs emissions affected $PM_{2.5}$ nitrate concentrations in Beijing and Qingdao, which are typical inland and coastal cities, respectively, in northern China. The simulations included single-

pollutant reduction strategies for $NH_3$, NOx, and VOCs and combined reduction scenarios. For each pollutant, emissions were reduced by 20%, 40%, and 60%, with a focus on assessing the resulting influence on nitrate formation. The combined reduction scenarios entailed simultaneous reductions in $NH_3$, NOx, and VOCs to evaluate synergistic effects. The nitrate concentration responses to these reductions were analyzed to determine the most effective strategies for controlling winter nitrate levels.

**3 Results and discussion**

**3.1 Model evaluation**

**3.1.1 Evaluation of meteorological parameters**

**Table 1 Statistical performance of the modeled meteorological parameters in the NCP during the winters of 2013 and 2018 (68 sites).**

| Parameters | Winter, 2013 | | | | Winter, 2018 | | | | Benchmark | | | |
|---|---|---|---|---|---|---|---|---|---|---|---|---|
| | MB | RMSE | IOA | R | MB | RMSE | IOA | R | MB | RMSE | IOA | R |
| T2/°C | 0.60 | 3.25 | 0.93 | 0.88 | 1.15 | 3.14 | 0.94 | 0.90 | ≤±0.5 | / | ≥±0.8 | / |
| RH/% | -3.67 | 17.53 | 0.82 | 0.69 | -5.35 | 19.36 | 0.81 | 0.68 | / | / | / | / |
| WS10/(m·s⁻¹) | 1.55 | 2.79 | 0.61 | 0.47 | 1.47 | 2.72 | 0.60 | 0.46 | ≤±0.5 | ≤±2.0 | ≥0.6 | / |
| WD10/° | 5.44 | 115.02 | 0.70 | 0.43 | -1.65 | 121.00 | 0.69 | 0.42 | ≤±10 | / | / | / |

* T2 denotes the 2-m temperature, RH denotes the relative humidity, WS10 denotes the 10-m wind speed, and WD10 denotes the 10-m wind direction.

     The statistical model performance for meteorological parameters such as T2, WS10, WD10, and

RH in the NCP is summarized in Table 1, covering the winter months from December 2013 to February 2014 (winter, 2013) and from December 2018 to February 2019 (winter, 2018). The simulated T2 and RH values exhibited satisfactory reproducibility, with temperature simulations exhibiting MB values slightly higher than the recommended threshold (MB≤±0.5). The simulated wind speeds during both winters were slightly overestimated, with the RMSE slightly exceeding the criterion (2 m/s; Emery et al. (2001)), whereas the wind direction results fully satisfied the criterion. The observed wind speed overestimation by the WRF model could be attributed to its inability to accurately capture the impact of high aerosol loadings on shortwave radiation in winter, which likely reduced the near-surface wind speed (Tan et al., 2017; Jacobson and Kaufman, 2006). In general, the meteorological field simulations of the WRF model were reliable and could effectively reveal the changes in various meteorological elements in the NCP.

### 3.1.2 Evaluation of atmospheric pollutants

Table 2 Model performance for the major air pollutants in typical cities of the NCP during the winters of 2013 and 2018.

| City | Pollutants ($\mu g/m^3$) | 2013 winter | | | | 2018 winter | | | |
|---|---|---|---|---|---|---|---|---|---|
| | | NMB | NME | MB | R | NMB | NME | MB | R |
| Beijing (BJ) | MDA8 $O_3$ | -5% | 25% | -2.04 | 0.76 | -13% | 25% | -6.80 | 0.68 |
| | $NO_2$ | 1% | 19% | 0.08 | 0.89 | 32% | 46% | 13.85 | 0.73 |
| | $PM_{2.5}$ | -16% | 28% | -17.41 | 0.88 | -10% | 45% | -5.11 | 0.64 |
| Tianjin (TJ) | MDA8 $O_3$ | -4% | 30% | -1.23 | 0.73 | -29% | 33% | -14.19 | 0.63 |
| | $NO_2$ | 8% | 22% | 5.53 | 0.76 | 26% | 39% | 14.16 | 0.68 |
| | $PM_{2.5}$ | 32% | 41% | 34.73 | 0.81 | 2% | 51% | 1.75 | 0.62 |
| Shijiazhuang (SJZ) | MDA8 $O_3$ | -15% | 29% | -5.49 | 0.64 | -25% | 35% | -11.25 | 0.58 |
| | $NO_2$ | -5% | 27% | -3.66 | 0.73 | 12% | 32% | 7.40 | 0.61 |
| | $PM_{2.5}$ | -25% | 34% | -48.32 | 0.80 | -20% | 39% | -23.13 | 0.59 |
| Jinan (JN) | MDA8 $O_3$ | 2% | 25% | 0.87 | 0.77 | -11% | 31% | -6.41 | 0.54 |
| | $NO_2$ | -14% | 20% | -10.20 | 0.73 | -8% | 30% | -5.01 | 0.61 |
| | $PM_{2.5}$ | -7% | 21% | -9.54 | 0.85 | -10% | 33% | -8.82 | 0.70 |
| Zhengzhou (ZZ) | MDA8 $O_3$ | -6% | 49% | -2.65 | 0.03 | 1% | 34% | 0.16 | 0.63 |
| | $NO_2$ | 12% | 24% | 7.52 | 0.69 | 11% | 34% | 6.10 | 0.61 |
| | $PM_{2.5}$ | 24% | 33% | 30.63 | 0.85 | -9% | 36% | -10.34 | 0.63 |
| Qingdao (QD) | MDA8 $O_3$ | -1% | 21% | -0.66 | 0.82 | -3% | 22% | -1.91 | 0.47 |
| | $NO_2$ | 11% | 23% | 7.52 | 0.69 | -4% | 28% | -1.99 | 0.76 |
| | $PM_{2.5}$ | 4% | 21% | 3.70 | 0.89 | -3% | 38% | -1.84 | 0.70 |
| Yantai (YT) | MDA8 $O_3$ | 5% | 17% | 2.48 | 0.79 | -10% | 22% | -6.50 | 0.27 |
| | $NO_2$ | -30% | 32% | -15.34 | 0.81 | -9% | 33% | -3.31 | 0.78 |
| | $PM_{2.5}$ | -6% | 25% | -4.09 | 0.84 | -19% | 33% | -10.84 | 0.78 |
| Benchmark | MDA8 $O_3$ | ≤±15% | < 25% | / | > 0.50 | / | / | / | / |
| | $NO_2$ | ≤±30% | ≤75% | / | / | / | / | / | / |
| | $PM_{2.5}$ | ≤±30% | < 50% | / | > 0.40 | / | / | / | / |

* The benchmarks for these pollutants, including $NO_2$ according to Us-Epa (2007); MDA8 $O_3$ and $PM_{2.5}$, were proposed by Emery et al. (2017) and Huang et al. (2021).

As indicated in Table 2, the model exhibited a favorable simulation ability for the concentrations of $NO_2$, MDA8$O_3$, and $PM_{2.5}$ pollutant in the NCP region. Specifically, the simulations accurately captured

NO$_2$ concentrations, with NMB values ranging from -30% to 32%. Notably, during the winter of 2018, the simulated NO$_2$ concentration in Beijing was slightly higher than the recommended range (by 2%), whereas that in other cities remained within this range. The MDA8O$_3$ simulations generally exhibited slight underestimation, whereas in Tianjin, Shijiazhuang, and Zhengzhou, the simulations slightly

exceeded the benchmark. This difference was primarily due to the uncertainty in anthropogenic VOC emissions (Wang et al., 2014), making accurate ozone simulations particularly challenging (Sun et al., 2022; Yang et al., 2024). The PM$_{2.5}$ simulations were generally accurate, although the simulated concentrations in Tianjin during the winter of 2013 were slightly elevated, with an NMB of 32%, exceeding the standard by 2%; however, those in the other cities met the standard requirements.

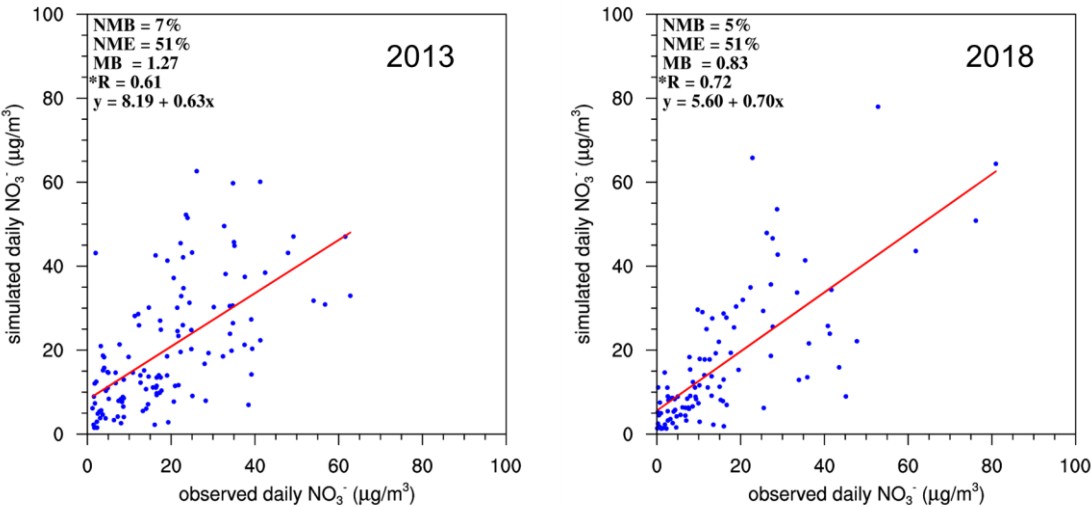


**Figure 3 Scatter plots of the simulated and observed daily mean NO$_3^-$ concentrations in the NCP during the winter seasons of 2013 and 2018; the * symbol indicates R values that are statistically significan (*P* < 0.05).**

Figure 3 shows a comparison between the simulated and observed NO$_3^-$ concentrations in the six NCP cities during the winters of 2013 and 2018. Yantai was excluded from the comparison because of

the inadequacy of the observation data. The data sources are listed in Table S4. The comparison results indicated that the simulated NO$_3^-$ trends in the NCP were accurate during the winters of 2013 and 2018, with R values of 0.61 and 0.72, respectively, meeting the benchmark of 0.60. However, the simulation for 2013 yielded slightly overestimated values, with an MB of 1.27, corresponding to NMB = 7% and NME = 51%. In contrast, the simulation results for 2018 were closer to the observed values, with NMB

= 5% and NME = 51%. Overall, the model demonstrated suitable stability and accuracy in simulating atmospheric pollutants in the NCP region, providing a solid foundation for future analysis.

**3.2 Variations in the concentrations of PM₂.₅ and its components between 2013 and 2018**

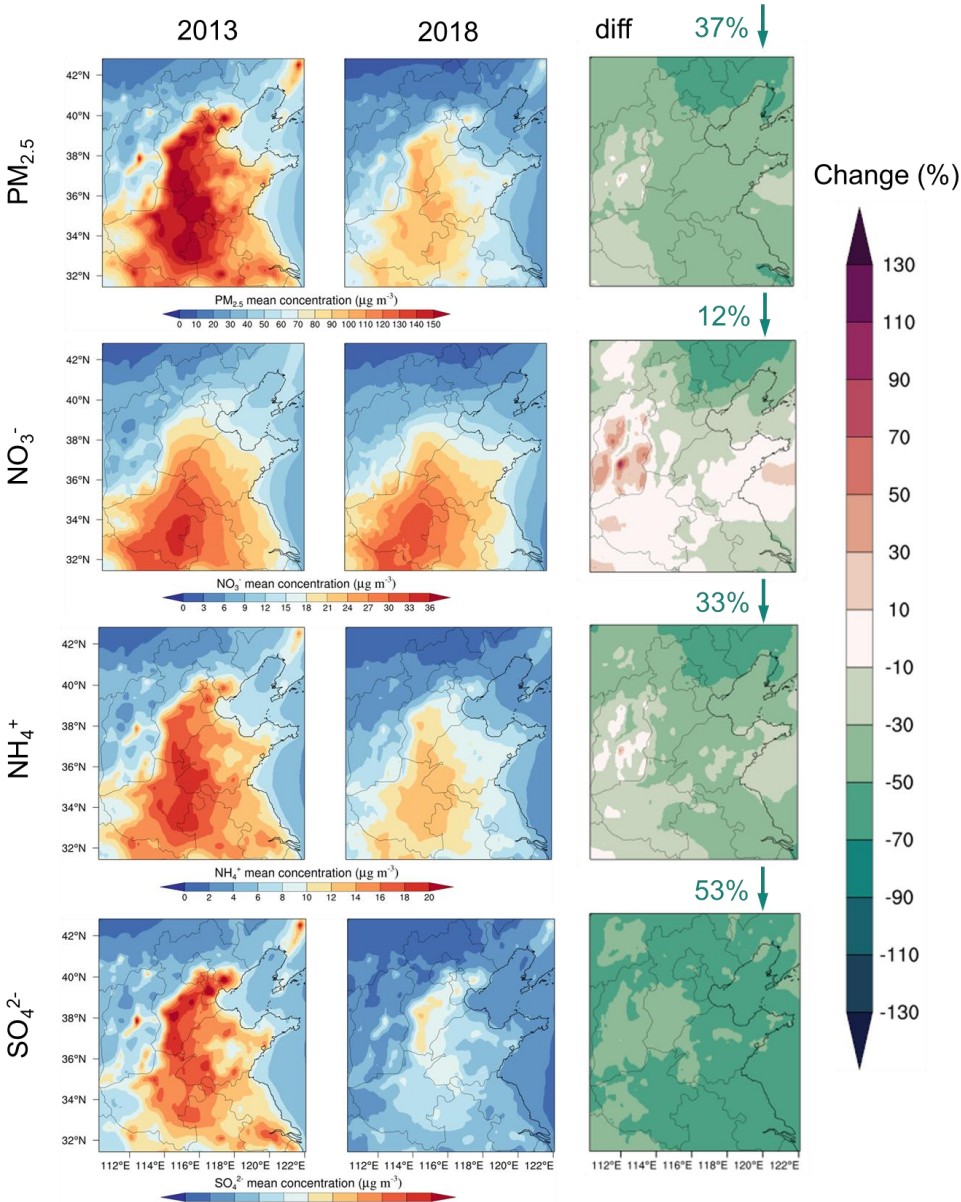

**Figure 4 Spatial distribution and changes in PM₂.₅ and its components in the NCP during the winters of 2013 and 2018. The up arrows indicate increases, and the down arrows indicate decreases.**

During the winter of 2013, the average PM₂.₅ concentration in the NCP reached $80.52 \pm 77.13$ µg/m³, with high-concentration areas mainly concentrated in the provinces of Beijing, Tianjin, southern Hebei, Henan, Shandong, and Anhui (Figure 4). By 2018, the average PM₂.₅ concentration had decreased to $50.74 \pm 49.26$ µg/m³, representing a significant reduction of $37\% \pm 10\%$. Previously, scholars reported similar trends in PM₂.₅ reduction across China (Li et al., 2023; Zhai et al., 2019). We found that the $SO_4^{2-}$ concentration decreased by $53\% \pm 5\%$, whereas the $NO_3^-$ and $NH_4^+$ concentrations decreased by $13\% \pm 20\%$ and $33\% \pm 13\%$, respectively (refer to Figure S2 for city-specific changes). Our analysis revealed a

pronounced difference in nitrate concnentration. Specifically, inland cities experienced a more pronounced reduction in nitrate concentrations (4.6% to 41.7%), whereas coastal cities showed relatively

small changes, with Yantai experiencing a 4.8% decrease and Qingdao exhibiting a 3.9% increase. The substantial reduction in the three inorganic components ($SO_4^{2-}$, $NH_4^+$, and $NO_3^-$) led to significant changes in the $PM_{2.5}$ composition, with the proportion of nitrate increasing relative to that of sulfate (Figure 5). Nitrate concentrations demonstrated significantly different patterns of reduction between coastal and inland regions, necessitating the identification of their dominant controlling factors to

facilitate effective regional pollution control strategies.

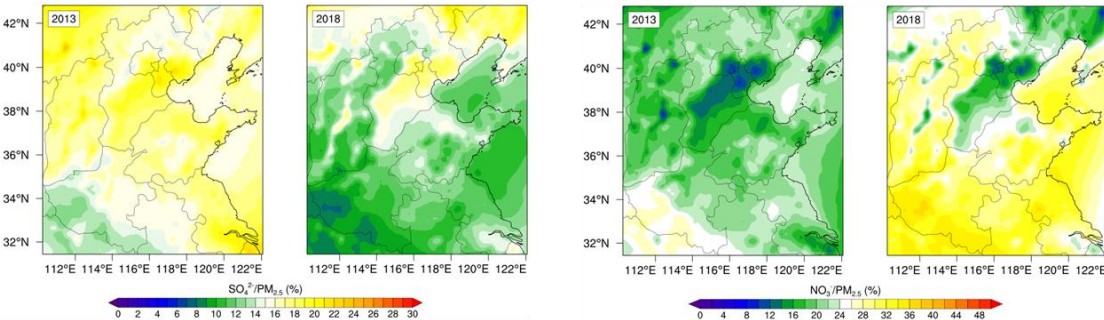

**Figure 5 Spatial distributions of the nitrate and sulfate proportions in $PM_{2.5}$ in the NCP region during the winters of 2013 and 2018.**

**3.3 Rates, contributions and diurnal variations in nitrate formation reactions in the major cities**

**Table 3 $TNO_3$ production rates (ppb/h) in the surface layer and contributions (%) of the major production pathways in the surface atmosphere in seven representative cities of the NCP in 2013 and 2018**

| City | Year | Production rates (ppb/h) | | | Daytime Contribution (%) | | | Nighttime Contribution (%) | | | Total Contribution (%) | | |
|---|---|---|---|---|---|---|---|---|---|---|---|---|---|
| | | $TNO_3$ | $OHNO_2$ | $hetN_2O_5$ | $OHNO_2$ | $hetN_2O_5$ | Other | $OHNO_2$ | $hetN_2O_5$ | Others | $OHNO_2$ | $hetN_2O_5$ | Others |
| BJ | 2013 | 0.19±0.15 | 0.15±0.16 | 0.03±0.02 | 90.2% | 5.2% | 4.6% | 33.4% | 46.9% | 19.7% | 77.8% | 14.3% | 7.9% |
| | 2018 | 0.17±0.15 | 0.15±0.16 | 0.02±0.01 | 95.1% | 2.6% | 2.3% | 48.4% | 39.4% | 12.2% | 85.6% | 9.9% | 4.5% |
| TJ | 2013 | 0.22±0.17 | 0.15±0.19 | 0.04±0.03 | 85.7% | 6.5% | 7.8% | 15.2% | 58.8% | 25.9% | 67.8% | 19.7% | 12.4% |
| | 2018 | 0.17±0.15 | 0.13±0.17 | 0.03±0.02 | 92.2% | 3.9% | 3.9% | 22.4% | 57.1% | 20.4% | 77.5% | 15.1% | 7.4% |
| SJZ | 2013 | 0.19±0.16 | 0.14±0.17 | 0.03±0.01 | 87.8% | 5.5% | 6.6% | 21.2% | 46.8% | 32.0% | 73.8% | 14.2% | 12.0% |
| | 2018 | 0.18±0.17 | 0.15±0.18 | 0.02±0.01 | 93.8% | 2.0% | 4.2% | 30.3% | 41.1% | 28.6% | 83.6% | 8.3% | 8.1% |
| JN | 2013 | 0.26±0.14 | 0.16±0.18 | 0.07±0.05 | 85.1% | 8.9% | 6.0% | 21.1% | 65.4% | 13.5% | 63.7% | 27.7% | 8.5% |
| | 2018 | 0.22±0.15 | 0.16±0.18 | 0.04±0.03 | 90.7% | 4.4% | 4.9% | 27.6% | 59.5% | 12.9% | 72.9% | 19.9% | 7.1% |
| ZZ | 2013 | 0.24±0.21 | 0.19±0.23 | 0.03±0.01 | 88.9% | 4.7% | 6.4% | 26.8% | 37.4% | 35.8% | 77.6% | 10.6% | 11.8% |
| | 2018 | 0.24±0.20 | 0.19±0.22 | 0.03±0.02 | 91.9% | 3.1% | 5.0% | 32.8% | 47.5% | 19.6% | 79.2% | 12.7% | 8.2% |
| QD | 2013 | 0.27±0.09 | 0.14±0.14 | 0.11±0.07 | 75.8% | 17.8% | 6.4% | 16.5% | 75.6% | 7.9% | 51.5% | 41.5% | 7.0% |
| | 2018 | 0.25±0.09 | 0.14±0.15 | 0.09±0.06 | 82.2% | 12.0% | 5.8% | 17.5% | 74.8% | 7.6% | 56.5% | 37.0% | 6.5% |
| YT | 2013 | 0.24±0.06 | 0.12±0.12 | 0.11±0.07 | 76.0% | 18.1% | 5.9% | 14.7% | 78.9% | 6.4% | 48.2% | 45.7% | 6.1% |
| | 2018 | 0.21±0.06 | 0.11±0.11 | 0.09±0.07 | 79.9% | 14.7% | 5.4% | 14.9% | 78.4% | 6.7% | 50.4% | 43.5% | 6.0% |

The $NO_3^-$ concentration slightly decreased from 2013 to 2018, due to a decrease in the $TNO_3$ production rate (0–0.05 ppb/h), which could be attributed to the production rate of the $hetN_2O_5$ pathway (0–0.03 ppb/h). According to Table 3, the contribution increased in Zhengzhou from 2013 to 2018, whereas it decreased in six cities, including Beijing (-4.4%), Tianjin (-4.6%), Shijiazhuang (-5.9%), Jinan (-7.8%), Qingdao (-4.5%), Yantai (-2.2%), and Zhengzhou (+2.1%). For the $OH + NO_2$ reaction pathway, the rate remained unchanged in all but four cities (Beijing, Jinan, Zhengzhou, and Qingdao); it varied between -0.02 and 0.01 ppb/h in the other cities. However, the contribution of the $OH + NO_2$ reaction pathway to $TNO_3$ formation increased overall due to the decreased rate of the $hetN_2O_5$ reaction pathway, although the production rate remained almost unchanged. Moreover, the contribution increase was greater in inland cities than in coastal cities. Among the five inland cities (Beijing, Tianjin, Shijiazhuang, Jinan, and Zhengzhou), the $OH + NO_2$ reaction pathway was the main nitrate formation mechanism, with its contribution ratio ranging from 63.7%–77.8% in 2013 and increasing to 72.9%–85.6% in 2018. In contrast, in the two coastal cities (Qingdao and Yantai), the $OH + NO_2$ reaction pathway contributed to 48.2%–56.5% of nitrate formation. However, the contribution of the $hetN_2O_5$ pathway ranged from 37.0%–45.7%. Therefore, both pathways were equally important in nitrate formation in coastal cities. These findings highlighted notable differences in nitrate formation mechanisms between inland cities and coastal cities, demonstrating that nitrate formation mechanisms were strongly affected by the ocean.

Since the $OH + NO_2$ and $hetN_2O_5$ reactions were the major pathways affecting nitrate formation, we comprehensively analyzed the diurnal variations in these two $TNO_3$ formation reactions in the NCP, considering the inland city of Beijing and the coastal city of Qingdao as examples. Figure 6 shows the average diurnal variations in the $TNO_3$ production rates via the different pathways in Qingdao and Beijing. We found that the $OH + NO_2$ reaction pathway exhibited similar diurnal characteristics in these two cities. Specifically, from night to the early morning (0:00 to 8:00), the reaction rate under this pathway remained almost zero because of the lack of photochemical processes that generated OH radicals; these processes were limited by solar radiation (Liu et al., 2020; Sun et al., 2022; Tan et al., 2021). Observational data from the Pearl River Delta region supported this result, showing that OH radicals only began to accumulate significantly after 6:00 AM (Hofzumahaus et al., 2009; Lu et al., 2012). After approximately 8:00 AM, the reaction rate rapidly increased, reaching a peak between 11:00 and 13:00 at approximately 0.4 ppb/h (~2.8 $\mu g/m^3$/h), resembling the findings for the NCP and Shanghai (Tan et al., 2021; Sun et al.,

2022; Liu et al., 2020). The reaction rate decreased with decreasing sunlight during the afternoon and returned to low levels at night due to a decrease in the number of photochemical processes that occurred. Compared with those during the winter of 2013, the peak reaction rates of the OH pathway decreased from 0.43 to 0.42 ppb/h and from 0.45 to 0.44 ppb/h in 2018 in Qingdao and Beijing, respectively. The

contributions of the OH + $NO_2$ reaction pathway to $TNO_3$ formation were in the ranges of 90.2%–95.1% and 75.8%–82.2% in Beijing and Qingdao, respectively, during the daytime. However, the other cities in the NCP exhibited similar diurnal variations. Therefore, the OH + $NO_2$ reaction pathway dominated during the daytime in both the inland and coastal cities.

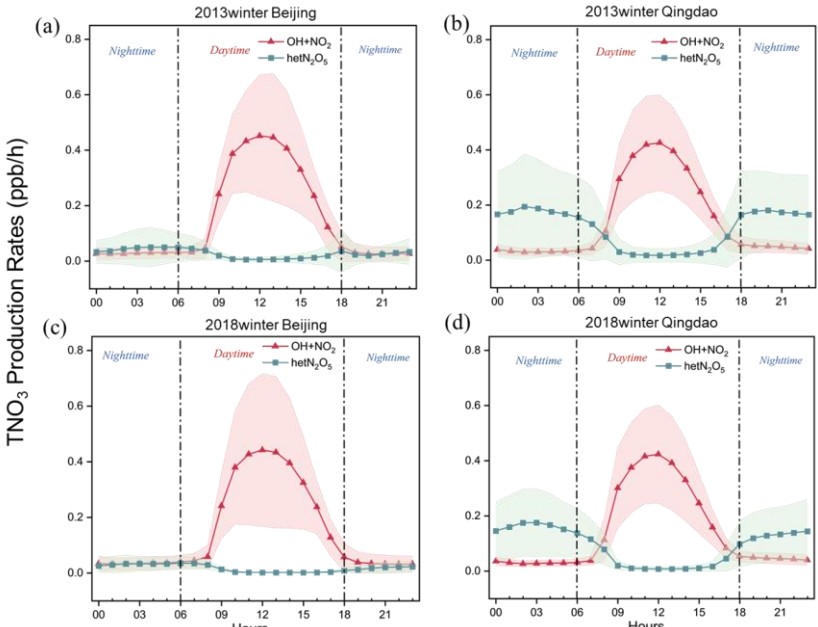

**Figure 6 Average diurnal variation in the $TNO_3$ production rate under different pathways (daytime: 06:00–18:00 BJT; nighttime: 18:00–06:00 BJT): (a) Beijing, 2013; (b) Qingdao, 2013; (c) Beijing, 2018; and (d) Qingdao, 2018**

At night, the het$N_2O_5$ reaction pathway notably affected $TNO_3$ formation. The rate of the het$N_2O_5$ reaction pathway showed similar bimodal diurnal variations in Qingdao and Beijing. However, the peak

values occurred at 2:00 and 20:00 in Qingdao, with much higher peak values than those in Beijing, where the peak values occurred from 4:00–5:00 and at 18:00. The het$N_2O_5$ reaction rate was relatively high at night but decreased rapidly by early morning, reaching its lowest point during the daytime (9:00 to 16:00). This phenomenon occurred because the $N_2O_5$ hydrolysis reaction dependeds heavily on $NO_3$ radicals and $N_2O_5$, which were only stable in weakly illuminated environments (Tie et al., 2003b; Tie et al., 2003a;

Atkinson et al., 2004).

High concentrations of $NO_2$ and $O_3$, precursors for $N_2O_5$, were considered key in this process (Dentener and Crutzen, 1993):

$$NO_2 + O_3 \rightarrow NO_3 + O_2 \tag{R8}$$

$$NO_3 + NO_2 \leftrightharpoons N_2O_5 \tag{R9}$$

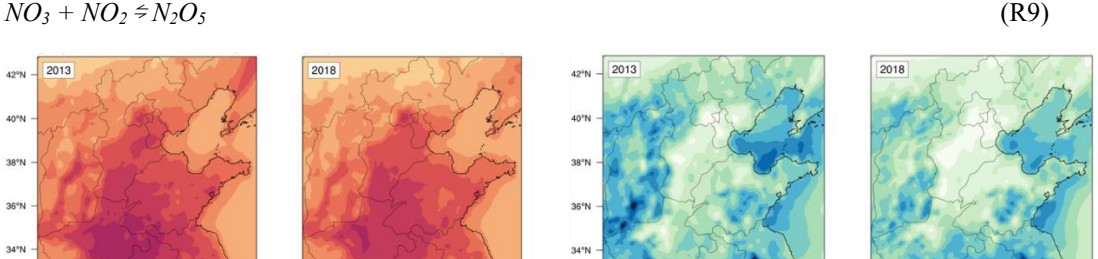


**Figure 7 Spatial distributions of the mean reaction rates for the nitrate formation pathways (OH + NO₂ and hetN₂O₅) in the NCP during the winters of 2013 and 2018.**

However, owing to high rates of photolysis during the day, $NO_3$ radicals and $N_2O_5$ could accumulate only at night (Zhao et al., 2023; Atkinson et al., 2004). In Beijing, the hetN₂O₅ reaction rate was relatively

low, with a peak value of only 0.03 ppb/h and a nighttime contribution rate of 46.9%. In contrast, Qingdao presented a peak hetN₂O₅ reaction rate of 0.2 ppb/h (approximately 1.4 μg/m³/h) at night, with an average contribution of 75.2%. This difference arose primarily due to the higher $N_2O_5$ concentrations and RH levels in Qingdao than in Beijing, as detailed in Section 3.5.3. The other inland cities exhibited diurnal variations similar to those in Beijing, whereas the coastal cities presented trends similar to those in

Qingdao. Compared with those during the winter of 2013, the reaction rates of the hetN₂O₅ pathway in both cities during the winter of 2018 decreased, from 0.19 to 0.18 ppb/h in Qingdao and from 0.07 to 0.03 ppb/h in Beijing. Comparative analysis revealed significant variations in the nitrate formation pathways between the coastal and inland areas, particularly in the hetN₂O₅ reaction rate (Figure 7). The factors influencing these variations are discussed in Section 3.5.3, which specifically addresses the

underlying mechanisms responsible for the differences in hetN₂O₅ reaction rates between coastal and inland regions.

**3.4 Assessment of the simulated nitrate formation reactions via the isotopic method**

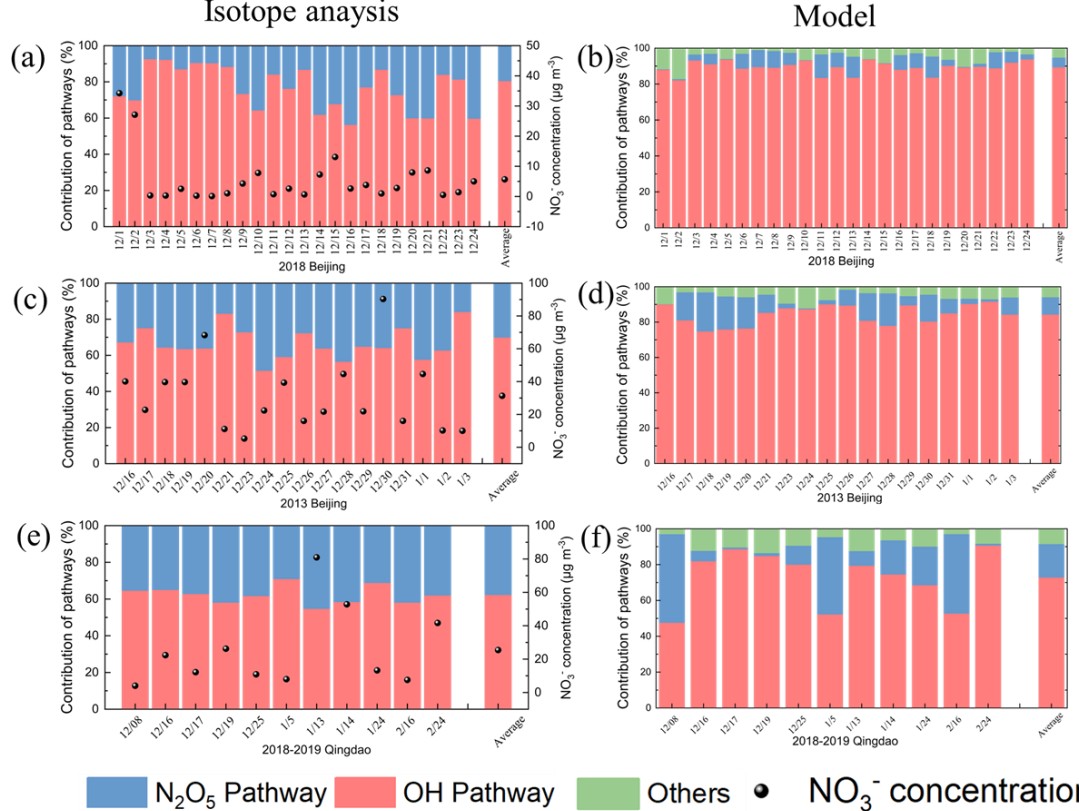


**Figure 8 Time series of the contributions of the atmospheric NO₃⁻ formation pathways: (a, b) Beijing 2018, (c, d) Beijing 2013, and (e, f) Qingdao 2018, based on (a, c, e) dual-isotope analysis and (b, d, f) model simulations.**

To verify the accuracy of nitrate formation reaction simulations, we conducted a comprehensive isotope analysis in Qingdao and Beijing. The isotope analysis conducted in Qingdao and Beijing for the

2013 and 2018 data (Figure 8) revealed that the model-simulated and isotope-based contributions of NO₃⁻ formation pathways generally exhibited consistent trends, although some differences were observed. In 2013 in Qingdao, the CMAQ model yielded contribution values of 72.9% (OH+NO₂) and 18.7% (hetN₂O₅), whereas the isotope analysis suggested 62.4 ± 22.5% and 37.6 ± 22.5%, respectively. By 2018 in Beijing, the model yielded 89.5% vs. an isotope-derived 76.5 ± 15.8% for OH+NO₂ and 5.4%

vs. 23.5 ± 15.8% for hetN₂O₅. Overall, although the differences persisted, the model captured the general trends, and the deviations fell within an acceptable range, indicating that the model simulations generally agreed with the isotope analysis. Several factors could explain these discrepancies. First, the isotope analysis carried intrinsic uncertainties, commonly approximately 10–30%, due to the standard deviations (SDs) in each pathway's endmember calculations and the posterior probability distributions generated by

the Bayesian model (Moore and Semmens, 2008). Second, the isotope analysis considered only two principal pathways; OH+NO₂ and hetN₂O₅. Although the other pathways only contributed to a small

fraction of $NO_3^-$ formation, their omission could cause biases, especially under specific meteorological or pollution conditions. Third, the IRR module did not consider long-range transport effects, which couldy further contribute to the discrepancies in the isotope results. In brief, the model simulation results

align reasonably well with the isotope analysis findings, despite some discrepancies between the two methods.

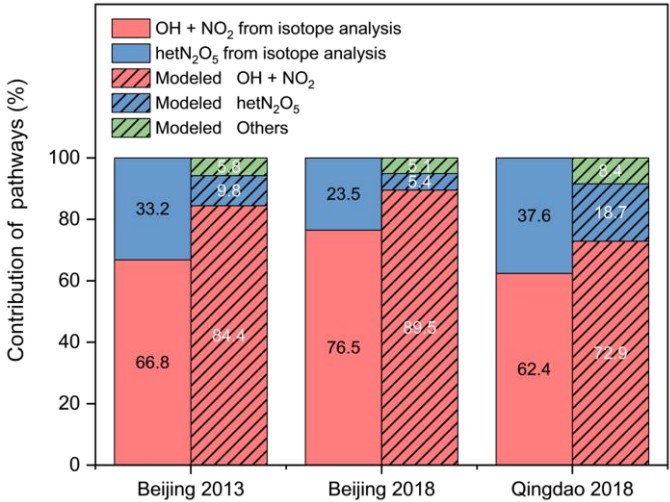

**Figure 9 Comparison of the contributions of the atmospheric $NO_3^-$ formation pathways based on the dual-isotope results and model simulations for Beijing in 2013 and 2018 and for Qingdao in 2018.**

Notably, both the model and isotope results indicated an increasing trend in the $OH+NO_2$ pathway contribution in Beijing from 2013 to 2018 (Figure 9), with the model and isotope analysis showing increases of 5.1% and 9.7%, respectively. This trend reflecteds the increased atmospheric oxidation capacity in the region. Regional comparative analysis revealed significant differences in nitrate formation pathways between Qingdao and Beijing (Figure 9). In 2018, the Qingdao' model results showed

contributions of 72.9% and 18.7% to the total nitrate formation for the $OH+NO_2$ and $hetN_2O_5$ pathways, respectively, compared with the isotope results of 62.4 ± 22.5% and 37.6 ± 22.5%, respectively. In contrast, the Beijing' model results indicated contributions of 89.5% and 5.4% to the total nitrate formation for the $OH+NO_2$ and $hetN_2O_5$ pathways, respectively, whereas the isotope results were 76.5 ± 15.8% and 23.5 ± 15.8%, respectively. Both methods demonstrated that the $OH+NO_2$ pathway dominated

nitrate formation in Beijing, while the $hetN_2O_5$ pathway played a substantial role in Qingdao, highlighting regional differences in the mechanism of nitrate formation and resulting in different changes in the nitrate concentration.

As discussed above, field observation data could include contributions from long-range transport (Walters et al., 2024), however, the CMAQ model's IRR module confined nitrate formation to individual grid cells. To quantify this discrepancy, we employed a backward trajectory analysis method incorporating temporal and spatial constraints to calculate the source regions and their contributions to the nitrate formation pathway.

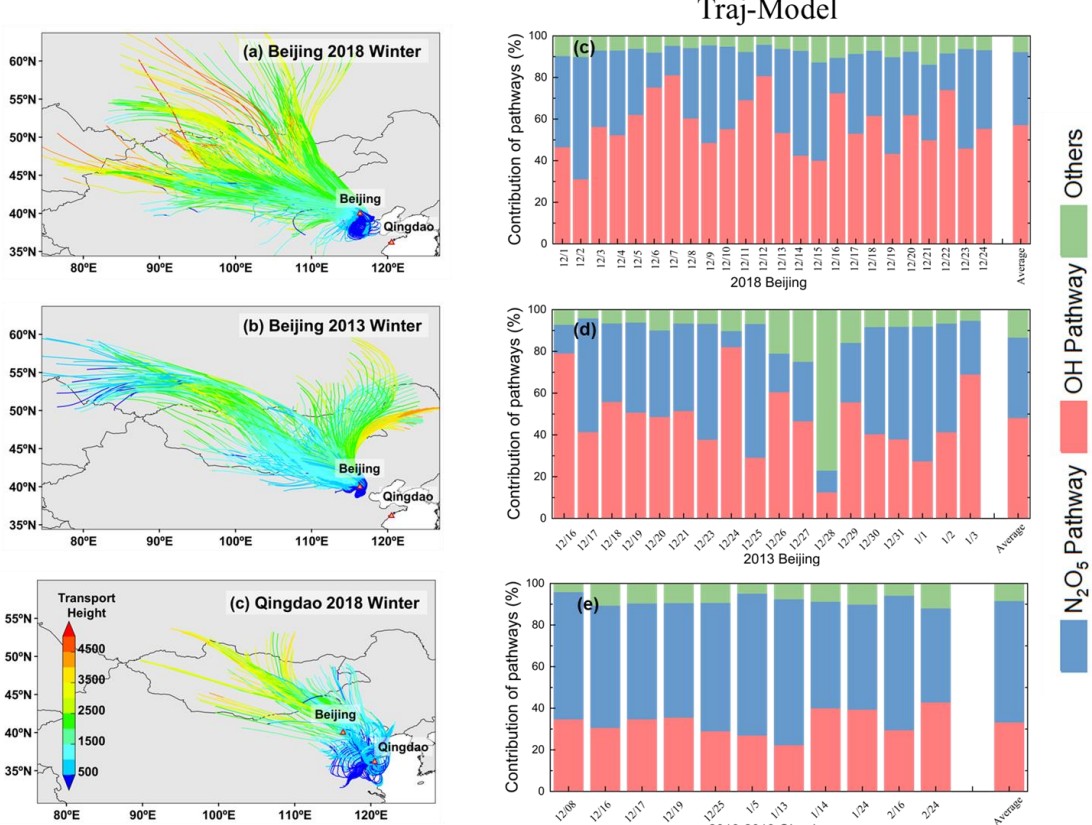

**Figure 10 Backward trajectories and nitrate formation pathways: (a-c) Backward trajectories for (a) Beijing 2018, (b) Beijing 2013, and (c) Qingdao 2018; time series of the contributions of the atmospheric NO₃⁻ formation pathways based on backward-trajectory-constrained model simulations for (c) Beijing 2018, (d) Beijing 2013, and (e) Qingdao 2018.**

The backward trajectory correction significantly altered the estimated nitrate formation pathways (Figure 10). In Qingdao, the corrected model results revealed an average OH+NO₂ pathway contribution of 33.3%, which was substantially lower than the isotope analysis result of 62.4 ± 22.5%, whereas the hetN₂O₅ pathway contribution was 58.4%, which was significantly greater than the isotope result of 37.6 ± 22.5%. In Beijing, the corrected model in 2013 indicated an OH+NO₂ pathway contribution of 48.3% (isotope: 66.8 ± 26.1%) and a hetN₂O₅ pathway contribution of 38.5% (isotope: 33.2 ± 26.1%). In 2018, the corrected model showed an OH+NO₂ pathway contribution of 57.2% (isotope: 76.5 ± 15.8%) and a

hetN$_2$O$_5$ pathway contribution of 35.1% (isotope: 23.5 ± 15.8%). Compared with the uncorrected model, the backward trajectory correction significantly improved the estimation of the hetN$_2$O$_5$ pathway contribution (Qingdao: from -18.9% to +20.8%; Beijing: from -18.1% to +11.6%) but exacerbated the discrepancy in the OH+NO$_2$ pathway contribution (Qingdao: from +10.5% to -29.1%; Beijing: from +13.0% to -19.3%). The effects of this correction exhibited regional differences: the improvement in the

hetN$_2$O$_5$ pathway was more pronounced in Qingdao, where regional transport had a greater impact, whereas the overadjustment of the OH+NO$_2$ pathway was more evident in Beijing, where local processes were dominant.

Remarkably, Walters et al. (2024) achieved relatively accurate results using the unmodified IRR module without considering transport effects, reporting an overall RMSE of 2.6‰ for $\Delta^{17}$O(HNO$_3$)

between modeled and observed values. This finding indicated that if the trends in nitrate formation pathways were similar across regions and vertical layers, the impact of transport on the overall simulation accuracy could be limited (Walters et al., 2024). Our results highlighted that in regions with significant spatial heterogeneity in formation pathways, such as Qingdao and Beijing, incorporating transport effects through backward trajectory analysis remained crucial for improving model performance.

These results demonstrated that backward trajectory correction played a positive role in improving the estimation of the hetN$_2$O$_5$ pathway, particularly in regions significantly influenced by regional transport. However, the overadjustment of the OH+NO$_2$ pathway highlighted the need for further optimization of the correction method. Key areas for improvement would  include refining OH radical concentration estimates and optimizing the quantification of regional transport. Currently, the overly

broad consideration of transport regions could obscure local formation signals, suggesting a need for increasingly precise spatial delineation in transport modeling. These enhancements would be essential for improving the model's applicability of the model across different regions and under varied meteorological conditions. Currently, owing to the lack of successful transport correction methodologies, we discussed the influencing factors without applying correctional procedures.

**3.5 Factors influencing nitrate formation pathways**

To explore the reasons for the observed coastal–inland differences in nitrate formation, we further investigated the influences of several key factors, including NH$_3$, NO$_2$, OH, O$_3$, RH, NO, and N$_2$O$_5$, on these pathways.

### 3.5.1 Influence of NH₃ on nitrate formation

The reaction of NH₃ with nitric acid (HNO₃) to form ammonium nitrate (NH₄NO₃) was considered an important process in the formation of atmospheric aerosols. In the absence of sufficient NH₃, the material would first react with H₂SO₄ to form (NH₄)₂SO₄ and then with HNO₃ to form NH₄NO₃ (Zhai et al., 2021). The availability of NH₃ was considered a critical factor governing the partitioning of HNO₃ to particulate nitrate (pNO₃).The gas ratio (GR) (Fu et al., 2020; Ansari and Pandis, 1998) was used to study whether NH₃ limits nitrate formation. The GR could be calculated as follows:

$$GR = \frac{([NH_3] + [NH_4^+]) - 2 \times [SO_4^{2-}]}{[NO_3^-] + [HNO_3]}$$

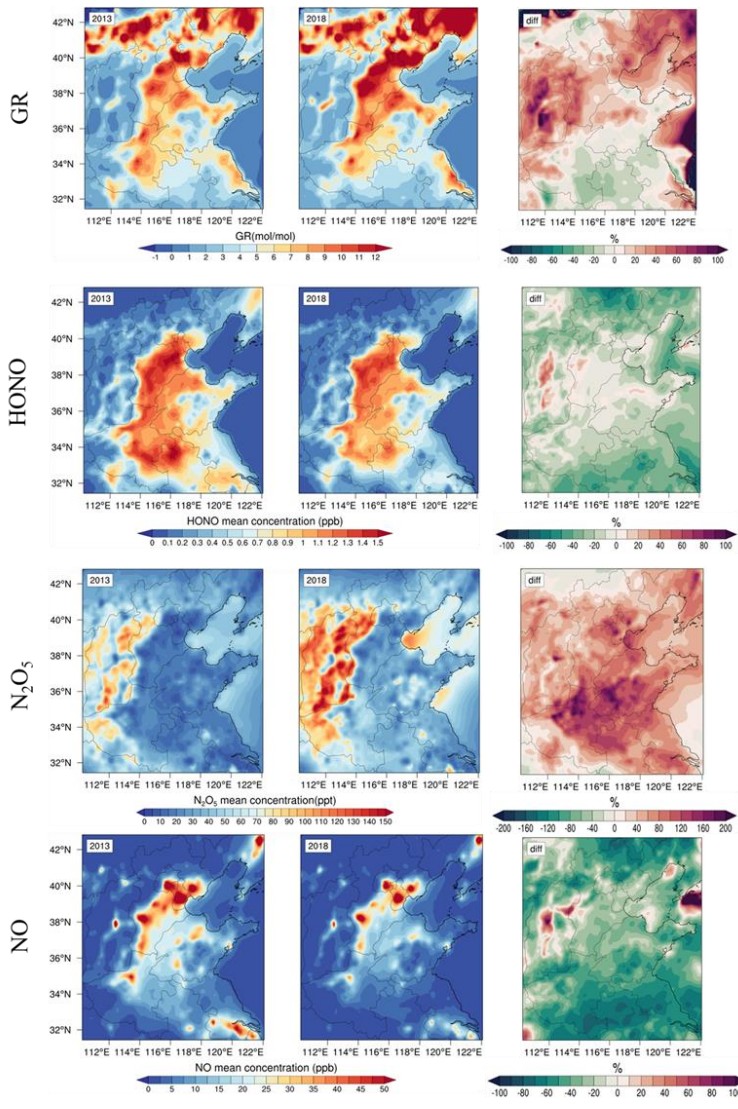

**Figure 11 Spatial distributions and interannual variations in the GR, HONO, N₂O₅, and NOₓ concentrations over the North China Plain during the winters of 2013 and 2018. The percentage changes (diff) represent the relative differences between 2018 and 2013.**

When the GR exceeded 1, the NH$_3$ concentration in the atmosphere was considered sufficient. A GR value between 0 and 1 indicated NH$_3$-neutral conditions, whereas a value less than 0 suggested NH$_3$-poor conditions, thus limiting NO$_3^-$ formation due to insufficient NH$_3$. In the NCP region, the GR values were generally greater than 2 in 2013 and 2018 (Figure 11), indicating that the region mainly exhibited a state of sufficient NH$_3$. Similar phenomena were observed by other researchers in the NCP region (Zhai et al., 2021; Xu et al., 2019c; Li et al., 2018). Therefore, in the NCP region, the formation of NH$_4$NO$_3$ was not limited by NH$_3$ in terms of supply.

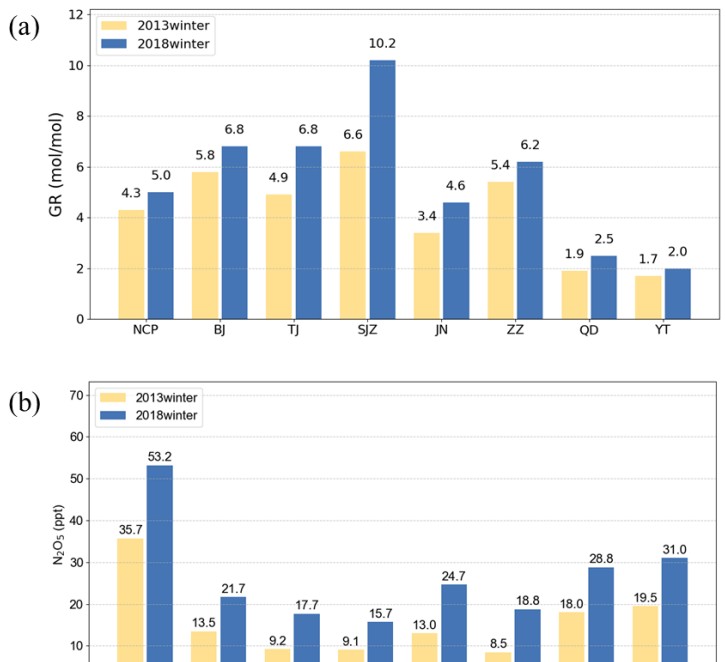

**Figure 12 (a) GR and (b) N$_2$O$_5$ concentrations across the North China Plain and seven major cities during the winters of 2013 and 2018.**

The GR in 2018 was greater than that in 2013 indicating an increasing surplus of NH$_3$. We believed that the variation in GR was caused by the significant reduction in sulfate and slight reduction in nitrate in PM$_{2.5}$ (Figure 12(a)) because of the minimal changes in NH$_3$ emissions. This hypothesis could be verified from an emission perspective. The total ammonia emissions in China increased from 9.64 to 9.75 Tg from 2013–2015 and then gradually decreased to 9.12 Tg by 2018 (Liao et al., 2022). Additionally, there were coastal–inland differences in NH$_3$ surplus levels. Notably, the NH$_3$ surplus ratio was consistently greater than 3.4 in inland regions, whereas it ranged from 1.7 to 2.5 in coastal cities. This difference was attributed primarily to intensive industrial activities related to fertilizer production in

inland areas, which resulted in ncreased $NH_3$ emissions. Data showed that $NH_3$ emissions from fertilizer production in the inland regions of the NCP ranged from 1.0 to 40.0 Gg/year, whereas emissions in coastal areas were significantly lower, ranging only from 0 to 0.1 Gg/year (Chen et al., 2022).

  In summary, $NH_3$ played a critical role in influencing particulate nitrate concentrations by affecting the gas–particle conversion of $HNO_3$ in the NCP region, although its availability was

sufficient. Furthermore, variations in $NH_3$ emissions and distinct land–ocean transport patterns significantly affected the formation and spatial distribution of particulate nitrate.

### 3.5.2 Factors influencing the OH + $NO_2$ reaction rate

  As the OH + $NO_2$ reaction significantly contributed to $TNO_3$ formation, the OH and $NO_2$ concentrations served as key factors. The $NO_2$ concentration in the NCP was 30.3 μg/m$^3$ during the winter

of 2013 and decreased to 24.7 μg/m$^3$ during the winter of 2018, and its concentration remained several orders of magnitude greater than that of OH radicals. Figure S3 shows that the $NO_2$/OH molar ratio in the NCP region generally exceeded $10^8$, indicating that the $NO_2$ concentration was in excess under emission reduction measures. Therefore, the reaction primarily depended on the OH radical concentration. Next, we examined the impact of the OH radical concentration on the reaction rate. OH radicals were

produced primarily via photochemical reactions, such as by the decomposition of HONO into OH and NO under sunlight (Song et al., 2023), which could account for 20% to 90% of the total primary production of OH radicals (Song et al., 2023; Xue et al., 2020; Kim et al., 2014a). Additionally, OH radicals could be produced indirectly via $O_3$ photolysis in the presence of water vapor (Fu et al., 2020; Kim et al., 2014a).

**Table 4 Observations and CMAQ model simulation results for the winter OH radical concentration**

| Species | Location | Obs. period | Obs. average | Sim. average | Reference |
|---|---|---|---|---|---|
| OH ($10^6$ molecules cm$^{-3}$) | Huairou_Beijing (40.41°N, 116.68°E) | Jan-Mar, 2016 | Peak 2.4 | Peak 0.8 | Tan et al. (2018) |
| | Beijing PKU (40°N, 116.3°E) | Winter, 2017 | Peak1.5 ~ 2.0 Daily average 0.3 | Peak 0.8 Daily average 0.2 | Ma et al. (2019) |

  We compare the simulated and observed OH radical concentrations in Table 4. The simulated daily average OH radical concentrations were relatively close to the observed values. However, the peak concentrations of OH radicals were lower than those observed. Underprediction of the OH peak concentration was determined to be a common issue in current model simulations (Czader et al., 2013;

Stone et al., 2012; Xue et al., 2020). Although the CMAQ model often underestimated high daytime concentrations (Czader et al., 2013), it satisfactorily captured the diurnal variation in OH radicals, likely because it underestimated HONO concentrations (Zhang et al., 2023b). The data (Figure 13 a-c) revealed that the peak value of OH radicals in Beijing significantly increased by $0.98 \times 10^6$ molecules cm$^{-3}$ in 2018 compared with that in 2013. A similar trend was observed in Qingdao, with an increase of $0.58 \times 10^6$

molecules cm$^{-3}$. The GEOS–Chem simulations Zhang et al. (2023a) indicated a stable upward trend in OH radical concentrations across the NCP from 2014 to 2017, ranging from 0.05 to $0.17 \times 10^6$ molecules cm$^{-3}$ a$^{-1}$. Our study data suggested that despite China's emission reduction measures, these efforts did not effectively control OH radical concentrations. The continuous increase in OH radicals accelerated the nitrate formation rate via the OH + NO$_2$ pathway.

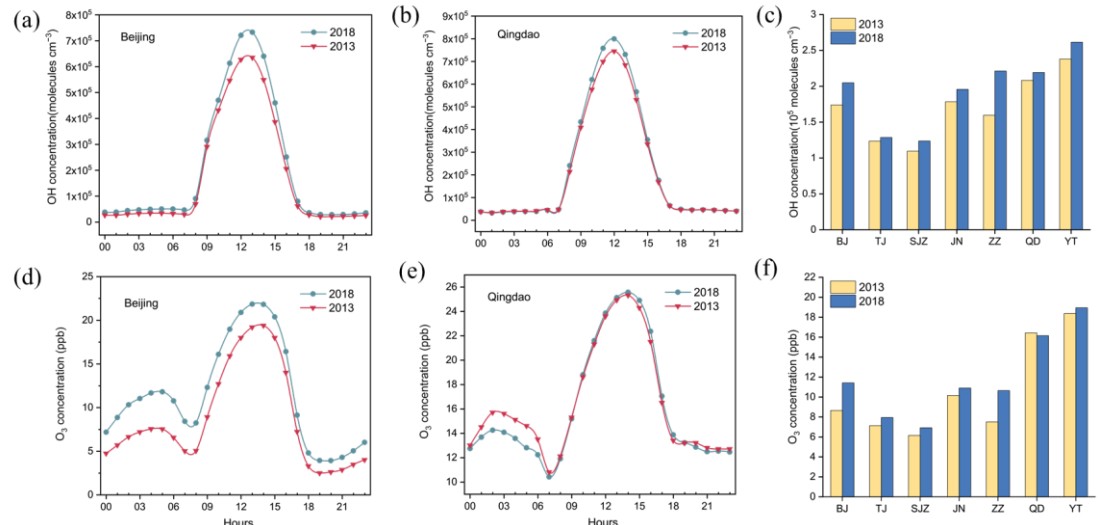


**Figure 13 Diurnal variations in OH radicals and O$_3$ in Beijing and Qingdao and average concentrations across seven major cities in the NCP (2013, 2018): (a) diurnal variation in OH radicals in Beijing, (b) diurnal variation in OH radicals in Qingdao, (c) average OH radical concentration in the seven cities, (d) diurnal variation in O$_3$ in Beijing, (e) diurnal variation in O$_3$ in Qingdao, and (f) average O$_3$ concentration in the seven**

**cities.**

One of the primary pathways for the production of OH radicals was the photolysis of nitrous acid (HONO) in the tropospheric atmosphere. This photolytic reaction could be expressed as follows:

HONO + hv (λ < 400 nm) → OH + NO (R10)

However, the data indicated that the HONO concentration decreased between 2013 and 2018

(Figure 11), suggesting that this production pathway did not significantly contribute to the increase in the OH radical concentration.

In the atmosphere, another source of OH was the reaction between water vapor (H$_2$O) and excited

oxygen atoms (O($^1$D)). Notably, excited oxygen atoms would typically be produced via $O_3$ photolysis (Kim et al., 2014a; Tan et al., 2019).

$O_3 + hv \quad (\lambda < 340\ mm) \quad \rightarrow O_2 + O(^1D)$ (R11)

The excited oxygen atoms (O($^1$D)) subsequently reacted with water vapor to produce OH radicals.

$H_2O + O(^1D) \rightarrow 2OH$ (R12)

Therefore, $O_3$ photolysis played an indirect but crucial role in the formation process. $O_3$ could oxidize NO to $NO_2$, which then reacted with OH radicals to form $HNO_3$, significantly influencing the

$OH + NO_2$ pathway. As shown in Figure 13(d–f), the increase in the $O_3$ concentration indirectly increased the $OH + NO_2$ reaction rate for nitrate formation. This effect was greater in inland cities, where the increase in the $O_3$ concentration was greater than that in coastal cities, leading to a greater contribution of the $OH + NO_2$ pathway. Thus, the OH and $O_3$ concentrations affected the formation rate and contribution of the $OH + NO_2$ pathway to nitrate formation due to the excess $NO_2$ concentration, although

the $NO_2$ emissions decreased. Therefore, inland cities should prioritize controlling the emissions of VOCs and NOx. These pollutants could generate OH radicals through photochemical reactions, which would subsequently dominate nitrate formation via the $OH + NO_2$ pathway. By reducing precursor emissions (VOCs and NOx), the production of OH radicals could be suppressed, thereby slowing the reaction kinetics and mitigating atmospheric nitrate accumulation..

**3.5.3 Factors influencing hetN$_2$O$_5$ reaction rates**

The heterogeneous $N_2O_5$ reaction was considered a major pathway for nitrate formation influenced by many factors, including the $N_2O_5$ concentration, aerosol surface area, and uptake coefficient of $N_2O_5$ ($\gamma N_2O_5$). The value of $\gamma N_2O_5$ varied widely, ranging from an extremely low value of $10^{-4}$ to a relatively high value of 0.1 (Wang et al., 2023). Experimental results indicated that environmental factors, such as

temperature and humidity, the composition of aerosols, such as $NO_3^-$, $Cl^-$, and $SO_4^{2-}$, the liquid water content in aerosols, the presence of organic compounds, and the mixing state of aerosols, were critical factors influencing $\gamma N_2O_5$ (Folkers et al., 2003; Mitroo et al., 2019; Thornton and Abbatt, 2005; Wahner et al., 1998; Wang et al., 2020). These factors collectively influenced the reaction rate and yield of heterogeneous $N_2O_5$ reactions.

With increasing concentrations of atmospheric $O_3$ (Figure 13(f)), the generation rate of $N_2O_5$ significantly increased. Consequently, the average $N_2O_5$ concentration in the NCP was 35.7 ppt in 2013

and increased to 53.2 ppt in 2018 (Figure 12(b)). However, the $N_2O_5$ concentrations were generally higher in ocean areas and western mountainous regions than in central inland areas. We found that this spatial distribution of $N_2O_5$ was caused by NO. In urban areas, especially during severe haze nighttime periods,

the rapid titration of NO with $O_3$ led to near-zero concentrations of surface $O_3$ (Zang et al., 2022), thereby inhibiting the in situ generation of $NO_3$ radicals and $N_2O_5$ (Zhao et al., 2023). As shown in Figure 11, the NO concentration exhibited a spatial distribution inversely correlated with that of $N_2O_5$, where higher NO concentrations corresponded to lower $N_2O_5$ levels and vice versa. This inverse relationship explains the low $N_2O_5$ concentrations in inland cities. Additionally, observations have showed that the lifetime of

$NO_3$ radicals in marine atmospheres could reach 30 minutes because of the low NOx mixing ratio in marine air masses, whichwas longer than 1 minute in inland atmospheres (Crowley et al., 2011) and contributed to the high $N_2O_5$ concentrations in coastal cities. Observations in Hebei Province revealed low $N_2O_5$ concentrations in summer, with lifetimes ranging from 0.1 to 10 minutes (Tham et al., 2018). Therefore, the spatial differences in het$N_2O_5$ reaction rates depended on NO and the lifetime of $NO_3$

radicals.

Moreover, we speculated that the decreased rate of heterogeneous $N_2O_5$ was influenced not only by $N_2O_5$ but also by other factors, such as RH and the surface area of aerosol particles, due to the interannual variations in RH and PM. Compared with 2013, the RH in the NCP decreased by 3.1% in 2018 (Figure S4). The sensitivity experiments revealed that both the $N_2O_5$ reaction rate and its contribution to total

nitrates remained essentially unchanged when the humidity decreased by 10%. This finding indicated that humidity changes were not the main factor influencing the decrease in the reaction rate of het$N_2O_5$ in the NCP. The total dry surface area of the particles in the CMAQ model was calculated as the sum of the modal dry surface areas, corresponding to the particles in the Aitken, the accumulation levels, and coarse modes (Bergin et al., 2022). During the winter of 2013 in the NCP, the average surface reaction

factor (SRF) value was 689 μm²/cm³, which decreased to 425 μm²/cm³ during the winter of 2018, indicating a reduction of 38.3%. This significant reduction in the aerosol surface area likely contributed to the observed decrease in the reaction rate of heterogeneous $N_2O_5$. Air quality monitoring data revealed that $PM_{10}$ decreased by 20% and $PM_{2.5}$ decreased by 28% in China from 2014 to 2018 (Fan et al., 2020a). Therefore, we considered that the reduction in the $PM_{2.5}$ concentration was a major factor contributing

to the decrease in the het$N_2O_5$ generation rate.

In brief, hetN$_2$O$_5$ reaction rates were affected by three key factors: N$_2$O$_5$ concentrations, aerosol surface area, and the spatial distribution of NO. The decline in hetN$_2$O$_5$ reaction rates across the NCP between 2013 and 2018 was driven primarily by a substantial reduction in aerosol surface area, particularly due to decreased PM$_{2.5}$ concentrations. Furthermore, regional variations in NO concentrations and NO$_3$ radical lifetimes significantly influenced N$_2$O$_5$ concentrations, resulting in distinct spatial patterns. To further mitigate nitrate pollution, coastal cities should prioritize reducing particulate matter concentrations, which would reduce hetN$_2$O$_5$ reaction rates and enhance air quality.

**3.6 Impact of reducing NOx and VOCs emissions on the nitrate concentration**

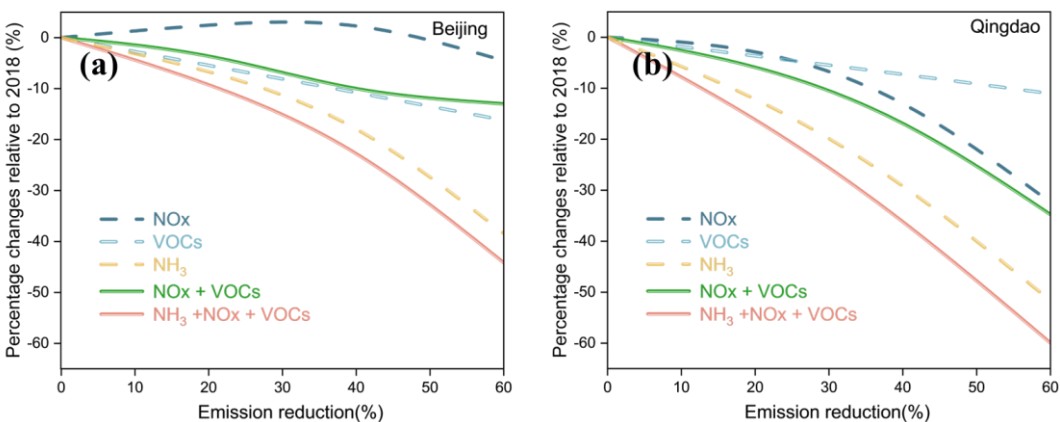

**Figure 14 Relative changes in the PM$_{2.5}$ nitrate contribution in response to the emission reduction scenarios in (a) Beijing and (b) Qingdao, compared with the winter 2018 baseline conditions.**

To reduce the nitrate content in PM$_{2.5}$, effective control strategies must target key factors, such as OH radicals, O$_3$, and the photochemical reactions of NOx and VOCs. However, a previous study revealed that even with a 30% reduction in VOCs and NOx emissions, the winter nitrate concentration in PM$_{2.5}$ in the NCP decreased by only 8.6% (Fu et al., 2020). NH$_3$ played a crucial role in the gas-to-particle conversion process of HNO$_3$ in the NCP region. Therefore, we designed single and combined pollutant reduction strategies for NH$_3$, NOx, and VOCs to examine the effects of the emission of these species on nitrate in PM$_{2.5}$.

The effectiveness of emission reduction scenarios notably differed between Beijing and Qingdao (Figure 14). In Beijing, single-pollutant reduction strategies yielded outcomes depending on target factors. The city showed limited sensitivity to NO$_x$ reduction, with nitrate concentrations initially increasing before showing a 4.5% decrease in nitrate concentration until reaching a 60% NO$_x$ reduction threshold. A 60% reduction in VOCs emissions led to a 10.5% decrease in nitrate concentration, whereas NH$_3$

reduction led to a 38.4% decrease at the same level. In contrast, Qingdao demonstrated relatively strong responsiveness to $NO_x$ controls, with a 60% reduction resulting in a 32.4% decrease in nitrate concentration. However, Qingdao showed relatively weak sensitivity to VOCs controls, with a 60% reduction in VOCs emissions resulting in an 11.1% decrease in nitrate concentration. Qingdao exhibited the strongest response to $NH_3$ reduction measures, where a 60% decrease in $NH_3$ emissions led to a substantial 51.7% reduction in nitrate concentration.

The comprehensive reduction scenarios revealed more pronounced intercity differences. Beijing showed relatively weak response to combined $NO_x$ and VOCs reductions, with only a 13.0% decrease in nitrate concentration at a 60% reduction in both pollutants. Qingdao, however, achieved a significantly greater reduction of 34.7% under the same conditions. The most effective strategy involved simultaneous reductions in $NH_3$, VOCs, and $NO_x$, with Qingdao achieving a 60.0% reduction and Beijing achieving a 44.2% reduction at 60% emission cuts. These findings demonstrated that effective nitrate control measures would require the coordinated reduction of multiple precursors. While multipollutant strategies yielded optimal results, $NH_3$ reduction were determined to be the most effective single-pollutant control measure for both cities.

**4 Conclusion**

In this study, nitrate formation mechanisms and influencing factors in seven cities across the NCP during the winters of 2013 and 2018 were investigated. Despite inherent uncertainties, the model simulations and isotopic measurements showed remarkable agreement, demonstrating the reliability of the modeling approach in elucidating nitrate formation mechanisms. Our findings revealed that nitrate formation was significantly influenced by $NH_3$, $NO_2$, OH radicals, $O_3$, NO, and $N_2O_5$, with distinct differences between regions. In inland cities, the OH + $NO_2$ reaction pathway dominated, contributing 63.7%–85.6%of nitrate formation. This phenomenon was largely driven by the increased concentrations of OH radicals and $O_3$, resulting in a 7.6% greater contribution of this pathway in inland cities in 2018 than in 2013. In contrast, coastal cities presented a greater contribution of the het$N_2O_5$ pathway (37.0% to 45.7%) because of the higher $N_2O_5$ concentrations and longer $NO_3$ radical lifetimes. High NO concentrations in inland areas facilitated $O_3$ titration, inhibiting $N_2O_5$ formation and further differentiating nitrate formation processes between the two regions.

Our emission reduction experiments revealed distinct regional responses and highlighted the need for tailored, multipollutant strategies to effectively control nitrate pollution. Among the single-pollutant control strategies, $NH_3$ reduction demonstrated the highest effectiveness, with a 60% emission reduction leading to a 38.4% decrease in nitrate concentrations in Beijing and a substantial 51.7% reduction in Qingdao. The combined emission control strategy, which inovlved simultaneous reductions in multiple pollutants, was more effective than $NH_3$ reduction alone, achieving a 44.2% nitrate reduction in Beijing and a 60.0% decrease in Qingdao. This study advanced our understanding of regional variations in atmospheric nitrogen chemistry and provided essential insights for developing targeted measures to mitigate nitrate pollution and improve air quality in northern China. To further advance research in this field, it is recommended that researchers in the future should focus on developing robust methodologies to comprehensively account for the influence of atmospheric transport on nitrate formation.

**Declaration of Competing Interest**

The authors declare that they have no known competing financial interests or personal relationships that could have appeared to influence the work reported in this paper.

**CRediT authorship contribution statement**

**Zhenze Liu**: Visualization, Formal analysis, Writing–original draft. **Xiaohuan Liu**: Project administration, Writing–review & editing. **Yuanze Ni**: Validation. **Likun Xue**: Resources. **Jianhua Qi**: Conceptualization, Methodology, Writing-review & editing

**Data availability**

All datasets supporting the findings of this study are available through the following channels. Primary datasets and analysis results are available from the corresponding authors upon reasonable request. All plotting data and essential research data have been deposited in a publicly accessible repository on the Baidu Cloud (Data link: https://pan.baidu.com/s/153rcdB-vTidH-14PPaXu-A; Access code: egus).

**Acknowledgments**

This work was financially supported by the Key Program of the National Natural Science Foundation of China (42430606). The authors are extremely grateful to Prof. Xiaodong Li for his help with the nitrate data.

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
