# Peer review of "Enhanced Atmospheric Oxidation and Particle Reductions Driving Changes to Nitrate Formation Mechanisms across Coastal and Inland Regions of North China"

_EGUsphere, 2024_

## Author Comment (AC2)

**Responses to reviewers (original comments by reviewers are in blue).**

**Reviewer #2:**

Summary:

The authors present an interesting study on wintertime atmospheric nitrate formation in the North China Region for 2013 and 2018, using model simulations aimed at validating the model against isotope observations. This research is highly valuable given the increasing contribution of nitrate to particulate matter, especially during the winter months. The authors have conducted detailed work on the topic; however, the presentation of their findings appears somewhat unfocused. Much of the model simulations and interpretations regarding nitrate changes reiterate findings from previous studies. The novel aspect of this research lies in the validation of model chemistry through comparison with isotope observations. However, most of the isotope data and modeling results are included in the supplementary material, which was not accessible for review. This component is critical for the interpretation of the work and requires thorough examination. The comparisons between model simulations and isotope observations were presented in broad terms, raising concerns about the reliability of using $d^{18}O$ and $d^{15}N$ values to determine oxidation pathways, given that the $d^{18}O$ values of atmospheric oxidants remain poorly constrained and recent documentation of potential $d^{18}O$ source effects from nighttime NO emissions. Moreover, the study lacked discussion on uncertainties in the isotope data and their potential impact on the interpretation within the modeling framework. Additionally, the model constrained nitrate production to a single grid cell, while the nitrate observations were derived from field samples that likely included contributions from long-range transport of nitrate produced upwind of the grid cell. Addressing this discrepancy is crucial for robust interpretation. Overall, while I appreciate the detailed efforts of the authors, the study appears unfocused due to the lack of integration and discussion of the model outputs, particularly the concentration and isotope comparisons. I recommend revisiting and refocusing the work to strengthen its coherence and clarity before it can be considered suitable for publication in ACP.

**Reply:** Thank you for your detailed feedback and valuable suggestions on our research. We understand the reviewer's concerns regarding the presentation of the study and some issues. We have made significant revisions to the manuscript, and addressed the following points in the revision:

**1. Comparison between model simulations and isotope observations**

To improve the comparison between model simulations and isotope data, we have significantly expanded the comparative analysis of model simulations and isotope data in Section 3.4, providing a more detailed analysis. Additionally, we have addressed the concerns regarding the reliability of using $d^{18}O$ and $d^{15}N$ values to determine oxidation pathways. Furthermore, in Section 2.4, we have incorporated a comprehensive discussion on the uncertainties in isotope data and their potential impact on the interpretation within the modeling framework.

**2. Use of isotope data**

The isotope data are presented in the supplementary materials, which the reviewer was unable to access for uncertain reasons. In the revised version, the supplementary materials are accessible, and we have added additional details on the isotope data and model results in the main text to better explain the study's findings.

**3. Regarding the constraint of using a single grid cell in the model**

We agree with the reviewer's point that the model's restriction of nitrate formation to a single grid

cell may introduce discrepancies compared with to field observations, which likely include contributions from long-range transport of nitrate produced upwind. To address this issue, we supplemented a detailed discussion in the manuscript (Section 3.4, lines 447–488) highlighting the potential discrepancies between the model results and actual observational data due to the exclusion of long-range transport effects. Additionally, we employed a backward trajectory analysis method to trace the source regions of nitrate and quantify the contributions of long-range transport to nitrate formation pathways. Furthermore, we conducted an error analysis between the observational and simulated results to assess the impact of long-range transport on model errors. Finally, we explored methods to quantitatively evaluate the influence of transport processes and refine the model's representation of regional transport dynamics.

**4. Overall structure and focus of the paper**

We agree that the overall structure and focus of the paper require further optimization to enhance coherence and clarity. In the revised version, we made several improvements: we revised the introduction to include a discussion on the applications and limitations of global models (lines 82–92), removed redundant results (e.g., findings already established in Section 3.2), consolidated figures and tables (e.g., Figure 11), highlighted the land–sea differences in the results, and refined the content and language throughout the manuscript to improve readability and precision.

**Comments:**

Lines 60 – 64: How was this determined?

**Reply:** The differences in nitrate formation pathways between inland and coastal cities during winter, as mentioned in lines 59–70, were derived from an analysis of the literature. Scholars have revealed significant variations in nitrate formation mechanisms between inland cities (e.g., Beijing) and coastal cities (e.g., Shanghai) during winter by isotope observations. In Beijing, the OH pathway contributes to 66–92% of nitrate formation, whereas the heterogeneous reaction of $N_2O_5$ (het$N_2O_5$) accounts for 8–34% (Chen et al., 2020). In contrast, in Shanghai, the OH pathway contributes to 48-74% of nitrate formation, which is significantly lower than that in Beijing (He et al., 2020). This discrepancy highlights the distinct nitrate formation mechanisms between inland and coastal cities. We have revised the phrasing in the manuscript accordingly.

Line 64 – 66: This sentence doesn't make sense to me.

**Reply:** Thank you for your careful and constructive feedback on our work. We have revised the text to clarify the role of coastal conditions in nitrate formation. The updated text now reads: "In coastal areas, which are influenced by high humidities, high sea salt levels and the combined effects of marine emissions and air masses (Zhong et al., 2023; Athanasopoulou et al., 2008; Zhao et al., 2024), the contribution of the OH pathway is smaller than that in inland cities, whereas the het$N_2O_5$ pathway plays a more significant role." (lines 64–67).

Lines 55 – 98: This appears to be a block of text and clear paragraph breaks are not apparent. This makes it hard for the reader to follow the main points in the introduction section.

**Reply:** Thank you for your valuable feedback. We have restructured the paragraph to improve readability by splitting it into two separate sections, which can now be found in lines 59–80 and lines 105–124.

Lines 78 – 79:   The motivation for the work should be stronger than pointed out in this line.   It is unclear to the readers how the nitrate formation cited in the works of this paragraph are from model studies or from some other mechanistic constraint.   Further, there have been global model studies of nitrate formation that would enable some insight into the land-ocean influence on nitrate formation.

**Reply:** Thank you for the valuable suggestions. While there are global model studies of nitrate formation, global models typically have coarse resolutions (generally 0.5°×0.625° or higher), making it difficult for them to capture the complex chemical processes occurring in localized areas. As a result, these models tend to reflect global average conditions and may not accurately represent local nitrate formation mechanisms (Alexander et al., 2020). Moreover, the treatment of aerosol chemistry in global models is relatively simplified, particularly concerning heterogeneous reactions on aerosol surfaces. Although these models have updated the probabilities for aerosol absorption reactions, significant uncertainties still exist regarding key parameters such as the $N_2O_5$ uptake coefficient and $HNO_3$ deposition rate (Heald et al., 2012). These uncertainties can lead to either an overestimation or underestimation of nitrate formation at local scales, especially in terms of ammonia emissions and nitrate production rates.

In contrast, regional multiscale models offer greater accuracy, which is why we focused on nitrate formation mechanisms in localized areas, particularly in heavily polluted urban centers or specific regions. By employing higher-resolution simulations and more detailed chemical reaction mechanisms, local nitrate formation processes can be captured more accurately.

Following your suggestion, we have revised the motivation section of the manuscript to highlight the advantages of regional-scale models (lines 92 – 94). Additionally, we have incorporated a discussion of the progress of both global and regional models in the study of nitrate formation, adjusting the sequence and emphasizing the contrast between the two approaches (lines 81–104). Furthermore, we have placed greater emphasis on the focus of this study in the final paragraph of the introduction (lines 125–132).

Alexander, B., Sherwen, T., Holmes, C. D., Fisher, J. A., Chen, Q., Evans, M. J., and Kasibhatla, P.: Global inorganic nitrate production mechanisms: comparison of a global model with nitrate isotope observations, Atmos. Chem. Phys., 20, 3859-3877, 10.5194/acp-20-3859-2020, 2020.
Heald, C. L., Collett Jr, J. L., Lee, T., Benedict, K. B., Schwandner, F. M., Li, Y., Clarisse, L., Hurtmans, D. R., Van Damme, M., Clerbaux, C., Coheur, P. F., Philip, S., Martin, R. V., and Pye, H. O. T.: Atmospheric ammonia and particulate inorganic nitrogen over the United States, Atmos. Chem. Phys., 12, 10295-10312, 10.5194/acp-12-10295-2012, 2012.

Lines 82 – 98: The jump from a discussion of nitrate formation (prior to lines 82) to the role of nitrate during haze events (Lines 82-98), back to nitrate formation mechanisms (Lines 99-108), is hard for the reader to follow.

**Reply:** Thank you for raising this issue. Following your suggestion, we have revised the text to improve clarity and logical flow, particularly in the discussion of nitrate formation mechanisms and their role in winter haze events. Specifically, we rephrased the introduction of the North China Plain (NCP) region (lines 105–107), restructured the discussion to highlight the shift from sulfate-dominated to nitrate-dominated haze following the Clean Air Action Plan (CAAP) implementation (lines 107–112), and expanded on nitrate formation mechanisms in coastal regions, emphasizing the

influences of marine air masses and sea–land and breezes (lines 120–124).

Lines 99 – 101:    Do coastal cities have a nitrate concentration change that is different compared to the inland cities?    I am still unclear the motivation to explore mechanism differences between inland and coastal cities as it relates to nitrate concentration changes.    Do we expect potential differences in chemistry to influence the rate of nitrate concentration change from these types of locations?

[Figure]

Figure R1 Concentrations of PM$_{2.5}$ and its components in seven major cities in the NCP region during the winters of 2013 and 2018

**Reply:** Thank you for your thoughtful and constructive feedback on our work. We observed that from 2013 to 2018, nitrate concentrations in inland cities decreased by 4.6%–41.7% during the emission reduction period, whereas the decline in coastal cities was smaller. For example, nitrate concentrations in Qingdao increased by 3.9%, and those in Yantai decreased by only 4.8%. Overall, the reduction in nitrate concentrations in coastal cities was significantly lower than in inland cities and did not align with the expected emission reduction results.

A key distinction between coastal and inland cities lies in their climatic conditions, which has motivated us to investigate whether differences in nitrate formation mechanisms exist between these regions. Specifically, we aimed to address the following questions. (1) Are there differences in nitrate formation mechanisms between coastal and inland cities? (2) If there are differences, how do they influence the variations in nitrate concentrations? (3) How can tailored emission control strategies be developed for coastal and inland cities on the basis of their distinct formation mechanisms to achieve effective pollution control?

Our findings indicated that changes in chemical composition significantly influenced the reaction rates of different formation pathways, thereby affecting nitrate concentration variations. This result underscored the need for developing targeted emission reduction strategies tailored to the unique chemical environments of inland and coastal cities.

Lines 104: I think finishing off the introduction with a statement of goals or objectives of the study would be important so readers know what to anticipate from this study, since the introduction was very broad.

**Reply:** We thank you for your comments and suggestions very much. We have added a concise statement of the research objectives at the end of the introduction, which can be found in lines 125–132.

Reply: The Multiresolution Emission Inventory for China (MEIC) is an emission inventory developed and maintained by Tsinghua University. This inventory includes a wide range of anthropogenic emission sources, including transportation, industry, and energy. The inventory provides emission data for various greenhouse gases and air pollutants across China. Further information can be accessed on the official website (http://meicmodel.org/). The full name and description of MEIC have been added to the revised manuscript (lines 153–155).

Lines 129: I think MEGAN needs a citation (rather than a link).

**Reply:** Thank you for your valuable suggestions. We have added the following reference to the revised version:

Guenther, A., Karl, T., Harley, P., Wiedinmyer, C., Palmer, P. I., and Geron, C.: Estimates of global terrestrial isoprene emissions using MEGAN (Model of Emissions of Gases and Aerosols from Nature). Atmospheric Chemistry and Physics, 6(11), 3181-3210. doi: 10.5194/acp-6-3181-2006, 2006.

Lines 144-150: What layer height is the IRR calculations used for interpreting nitrate formation? The layer closest to the surface?  An integrated column from the surface to some layer height? Further, the IRR will calculate the nitrate production within a grid cell while not including the influence of transported nitrate.  This could bias the interpretation if compared to isotope data. The readers should check out previous CMAQ work that have used IRR for simulation of nitrate formationd (Walters et al., Modeling the Oxygen Isotope Anomaly ($\Delta^{17}O$) of Reactive Nitrogen in the Community Multiscale Air Quality Model: Insights into Nitrogen Oxide Chemistry in the Northeastern United States, ES&T-Air, 1(6), 451-463, 2024).

**Reply:** Thank you for your insightful question. Our IRR calculations were applied to the layer closest to the surface, which was approximately 31 meters in height. This layer was considered the most representative of the predominant nitrate formation processes.

The IRR module confines nitrate formation to individual grid cells and does not account for the influence of nitrate transported from upwind regions. This limitation can introduce bias when comparing model results with isotope data, which may integrate contributions over a broader spatial domain, including long-range transport effects. To address this issue, we have employed a backward trajectory analysis method that incorporates both temporal and spatial constraints to identify the source regions of nitrate and quantify their formation pathway contributions. This approach allows us to better account for the influence of regional transport on nitrate formation, particularly in areas such as Qingdao and Beijing, where spatial heterogeneity in formation pathways is significant.

Our results demonstrate that backward trajectory correction significantly improves the estimation of the $hetN_2O_5$ pathway, especially in regions influenced by regional transport. However, the overadjustment of the $OH+NO_2$ pathway highlights the need for further optimization of the

correction method. Key areas for improvement include refining OH radical concentration estimates, accounting for aerosol aging processes, and optimizing the quantification of regional transport. Currently, the overly broad consideration of transport regions may obscure local formation signals, suggesting a need for more precise spatial delineation in transport modeling. We appreciate the reviewer's suggestion to refer to previous CMAQ work by Walters et al. (2024). These scholars achieved relatively accurate results without considering transport effects, indicating that if the trends in nitrate formation pathways are similar across regions and vertical layers, the impact of transport on overall simulation accuracy may be limited. However, in regions with significant spatial heterogeneity, such as Qingdao and Beijing, incorporating transport effects through backward trajectory analysis remains crucial for improving model performance. This detailed discussion has been added to Section 3.5 (lines 447–488) of the revised manuscript.

Walters, W. W., Pye, H. O. T., Kim, H., and Hastings, M. G.: Modeling the Oxygen Isotope Anomaly ($\Delta^{17}O$) of Reactive Nitrogen in the Community Multiscale Air Quality Model: Insights into Nitrogen Oxide Chemistry in the Northeastern United States, ACS ES&T Air, 1, 451-463, 10.1021/acsestair.3c00056, 2024.

**Lines 162-163:** Are these observations compared to near surface simulations or with a different layer height?

**Reply:** These observational results were compared with the simulated results for the near-surface layer.

**Lines 173:** Where is the supplement?

**Reply:** We apologize for the inconvenience. We have submitted the supplementary materials as requested and ensured that all relevant content is clearly labeled. It appears that, possibly due to a system error, the reviewer may not have been able to access these materials. To resolve this issue, we have added the supplementary materials at the end of the document with all the references clearly indicated. We will verify the completeness of these materials in our submission of the revised version. We kindly invite you to review them and welcome any further comments.

**Lines 178 – 180:** This sentence reads a bit odd to me. The way it is worded, it seems like the Thermo Scientific IC system extracted the samples in ultrapure water; however, I think the authors mean the IC system was used to measure ion concentrations.

**Reply:** We apologize for the confusion. Your understanding is entirely accurate. The samples were first extracted using ultrapure water (> 18.2 MΩ·cm), after which the concentrations of water-soluble inorganic ions (including $NO_3^-$, $SO_4^{2-}$, and $NH_4^+$) were measured using a Thermo Scientific Dionex ICS-1100 ion chromatograph system. We have revised the sentence to enhance clarity and precision.

Revised sentence: Water-soluble inorganic ions, such as $NO_3^-$, $SO_4^{2-}$, and $NH_4^+$, were extracted from the samples using ultrapure water (> 18.2 MΩ·cm), and their concentrations were measured with a Thermo Scientific Dionex ICS-1100 IC system, as described in a previous study (Qi et al., 2020). This revision can be found in lines 206–209 of the revised manuscript.

Qi, J., Yu, Y., Yao, X., Gang, Y., and Gao, H.: Dry deposition fluxes of inorganic nitrogen and phosphorus in atmospheric aerosols over the Marginal Seas and Northwest Pacific, Atmos. Res., 245, 105076, 10.1016/j.atmosres.2020.105076, 2020.

Lines 180 – 181:    What were the blank values?    Also, it is mentioned that the ionic concentrations were blank corrected, but blanks would also impact the isotope data.    Are blanks corrected for in the isotope data?    If so, how?

**Reply:** We apologize for the omission of information regarding the blank samples.

First, the average concentrations of $NO_3^-$, $SO_4^{2-}$, and $NH_4^+$ in the blank sample membranes were 0.067 mg $L^{-1}$, 0.051 mg $L^{-1}$, and 0.055 mg $L^{-1}$, respectively. Under the same sampling and extraction conditions, the blank values were converted to atmospheric concentrations of approximately 0.071 μg $m^{-3}$, 0.054 μg $m^{-3}$, and 0.058 μg $m^{-3}$, respectively. The blank membranes were placed in the high-volume sampler for 2 hours without sampling during the sampling period.

On the basis of the absolute nitrogen content in the blank samples, the influence of the blank on the isotopic values of the observed samples was less than 5%; therefore, it was not considered. A detailed explanation is provided below. Our isotope blank measurements followed the same procedure as the sample isotope analysis. Specifically, in the sample measurements, after purging with high-purity nitrogen, 20 nmol of nitrogen was added to the headspace vial containing the *Pseudomonas aureofaciens* (ATCC13985) strain. For the blank measurements, no sample was added, and after 24 hours, 10 M NaOH was directly injected to quench the reaction before analysis. The peak area in the chromatogram represents the absolute amount of $N_2O$ reduced by the strain, and the $\delta^{15}N$ and $\delta^{18}O$ values correspond to the $\delta^{15}N$ and $\delta^{18}O$ values of the sample. The peak area for the samples was approximately 10, whereas the peak areas for the two blank measurements were only 0.371 and 0.336, indicating an influence on the isotope values of less than 5%, which was negligible and thus not considered.

Lines 191-202:    Where is this information at?    This is critical to be reviewed since the calculation will have a major influence on the interpretation of the results.

**Reply:** This information is provided in the supplementary materials. As explained above, the reviewer was unable to access the supplementary materials for some reason. We have attached the supplementary materials at the end of the manuscript and ensured that all the referenced content is clearly indicated. We will double-check the materials during the next submission to ensure that everything is complete and accurate.

Lines 208- 2010: What are these benchmarks?
Table 1: I don't know what the benchmark section of the table means.

**Reply:** Thank you for your question. In this study, the benchmark values (Lines 244-248 in the revised manuscript) used are based on benchmarks set in previous studies (Emery et al.,2001, Emery et al., 2017, Huang et al., 2021) to evaluate the accuracy and validity of the meteorological model results. For example, the benchmark for T2 (2-meter temperature) is "≤±0.5°C," indicating that the model's predicted temperature should not deviate from observations by more than ±0.5°C. These benchmark values help assess whether the model's prediction errors are within an acceptable range, ensuring the model's accuracy.

For the explanation of the "Benchmark" section in Table 1, we listed the standard benchmark values for each meteorological parameter to assess the statistical performance of the model results. Specifically, the error for T2 (2-meter temperature) should not exceed ±0.5°C, the error for wind

speed (WS10) should be within ±0.5 m/s, and the IOA value should be greater than 0.6. The error for the wind direction (WD10) should be within ±10°. We have clarified the definitions and sources of the benchmark values, with specific standards provided in Table 1 and calculation formulas detailed in Table S3 (supporting information), ensuring that readers an better understand their significance.

Emery, C., Tai, E., and Yarwood, G.: Enhanced meteorological modeling and performance evaluation for two Texas ozone episodes, Prepared for the Texas natural resource conservation commission, by ENVIRON International Corporation, 2001.

Emery, C., Liu, Z., Russell, A. G., Odman, M. T., Yarwood, G., and Kumar, N.: Recommendations on statistics and benchmarks to assess photochemical model performance, Journal of the Air & Waste Management Association, 67, 582-598, 10.1080/10962247.2016.1265027, 2017.

Huang, L., Zhu, Y., Zhai, H., Xue, S., Zhu, T., Shao, Y., Liu, Z., Emery, C., Yarwood, G., Wang, Y., Fu, J., Zhang, K., and Li, L.: Recommendations on benchmarks for numerical air quality model applications in China – Part 1: $PM_{2.5}$ and chemical species, Atmos. Chem. Phys., 21, 2725-2743, 10.5194/acp-21-2725-2021, 2021

**Lines 257-265:   Are there regional differences in the model efficacy of nitrate concentrations?**

**Reply:** We sincerely appreciate your constructive comments. Indeed, the performance of nitrate concentration models varies across different regions, which is a common phenomenon in model simulations (Sun et al., 2022; Liu et al., 2020; Chuang et al., 2022; Xie et al., 2022). These regional differences in model performance are closely related to local pollution sources, meteorological conditions, and parameters of atmospheric chemical reaction mechanisms (Xie et al., 2022). Therefore, although model performance may vary across regions, the overall evaluation results remain acceptable (Fu et al., 2020). In this study, we conducted a comprehensive evaluation of nitrate concentrations in the North China Plain, and the results demonstrated that the model's accuracy is acceptable for this region.

Sun, J., Qin, M., Xie, X., Fu, W., Qin, Y., Sheng, L., Li, L., Li, J., Sulaymon, I. D., Jiang, L., Huang, L., Yu, X., and Hu, J.: Seasonal modeling analysis of nitrate formation pathways in the Yangtze River Delta region, China, Atmospheric Chemistry and Physics, 22, 12629-12646, https://doi.org/10.5194/acp-22-12629-2022, 2022.

Liu, L., Bei, N., Hu, B., Wu, J., Liu, S., Li, X., Wang, R., Liu, Z., Shen, Z., and Li, G.: Wintertime nitrate formation pathways in the North China Plain: Importance of $N_2O_5$ heterogeneous hydrolysis, Environmental Pollution, 266, 115287, https://doi.org/10.1016/j.envpol.2020.115287, 2020.

Chuang, M.-T., Wu, C.-F., Lin, C.-Y., Lin, W.-C., Chou, C. C. K., Lee, C.-T., Lin, T.-H., Fu, J. S., and Kong, S. S.-K.: Simulating nitrate formation mechanisms during $PM_{2.5}$ events in Taiwan and their implications for the controlling direction, Atmospheric Environment, 269, https://doi.org/10.1016/j.atmosenv.2021.118856, 2022.

Fu, X., Wang, T., Gao, J., Wang, P., Liu, Y., Wang, S., Zhao, B., and Xue, L.: Persistent heavy winter nitrate pollution driven by increased photochemical oxidants in northern China, Environmental Science & Technology, 54, 3881-3889, https://doi.org/10.1021/acs.est.9b07248, 2020.

Xie, X., Hu, J., Qin, M., Guo, S., Hu, M., Wang, H., Lou, S., Li, J., Sun, J., Li, X., Sheng, L., Zhu, J., Chen, G., Yin, J., Fu, W., Huang, C., and Zhang, Y.: Modeling particulate nitrate in China: Current

findings and future directions, Environment International, 166, 107369, https://doi.org/10.1016/j.envint.2022.107369, 2022.

Lines 266- 298: This is interesting but it appears to me that a lot of these results and implications have already been published in previous works. Can the authors focus on this section to have more of a focus on what information is new compared to what has already been published?

**Reply**: Thank you for the valuable suggestion. Some results and discussions of the increased proportion of nitrate overlap with those of existing studies. However, we focused on nitrate concentrations, which demonstrated significantly different reductions between coastal and inland regions. Specifically, we have clarified the significant differences in nitrate concentration trends between coastal and inland cities, with inland cities experiencing a more pronounced reduction (4.6% to 41.7%) than coastal cities, where nitrate reductions were smaller (e.g., Yantai decreased by 4.8%, whereas Qingdao increased by 3.9%). This regional differentiation in nitrate concentration trends has not been extensively explored in previous studies, and our research provides new insights into the complex interplay between emission controls and atmospheric chemistry. We have streamlined this section and expanded the comparative analysis of nitrate formation mechanisms. These revisions aim to clearly demonstrate the new insights and value of our work, particularly in understanding the regional differences in nitrate pollution dynamics. These changes can be found in lines 316–325 of the revised manuscript.

Lines 312-313: I am unsure what is meant by "reduced by -2.1% to 7.8%". Did some site reduce and some increased?

**Reply:** We apologize for the confusion. Some sites decreased and some increased. This phenomenon reflects the varying changes in the contribution of the $hetN_2O_5$ pathway to nitrate formation in different cities. According to Table 3, the contribution increased in Zhengzhou from 2013 to 2018, whereas it decreased in six cities: Beijing (-4.4%), Tianjin (-4.6%), Shijiazhuang (-5.9%), Jinan (-7.8%), Qingdao (-4.5%), Yantai (-2.2%), and Zhengzhou (+2.1%). We have revised the text (lines 334–336 of the revised manuscript) to clarify this point and avoid any potential misunderstanding. Thank you for bringing this to our attention.

Lines 382- 385: Are there major takeaways or implication for this finding? Does this imply that NOx oxidation is more efficient at the coastal site compared to urban or vice versa? Will this impact the change in nitrate concentrations due to emission regulations?

**Reply:** Thank you for the insightful question. Our study revealed significant differences in nitrate formation pathways between coastal and inland regions. Table 3 shows that the $TNO_3$ production rate is higher in coastal cities than in inland cities, indicating that the $NO_x$ oxidation efficiency is greater in coastal regions. Given the differences in nitrate formation mechanisms across regions, targeted emission reduction strategies can be implemented on the basis of the dominant formation pathways to effectively control nitrate concentrations.

Lines 394-396: the $d^{18}O$ value of atmospheric oxidants can vary widely. What values were chosen and are there temperature dependence factors that need to be accounted for? Further, recent work has shown that nighttime emissions of NO can carry over the emission $d^{18}O$ value, lowering $d^{18}O$ and $\Delta^{17}O$ compared to the assumption of complete photochemical cycle (Albertin et al.,

Measurement report: Nitrogen isotopes (d$^{15}$N) and first quantification of oxygen isotope anomalies (Δ$^{17}$O, d$^{18}$O) in atmospheric nitrogen dioxide, Atmos, Chem. Phys, 21, 10477-10497, 2021). This would be very important for urban areas with large nighttime NO emissions. Was this accounted for?

**Reply:** Your consideration is entirely correct. We apologize for not providing a sufficiently detailed description of our methods, which may have caused confusion. In our supplementary materials, we have included more comprehensive information on the methodology.

First, there are temperature dependence factors for the d$^{18}$O value. The $\delta^{18}$O values for atmospheric oxidants such as ·OH and O$_3$ are shown in Table R1. We calculated the endmember $\delta^{18}$O and $\delta^{15}$N values for each formation pathway, and incorporated temperature dependence factors into the thermodynamic fractionation calculations based on Luo et al. (2022).

Additionally, as you noted, numerous scholars have reported significant diurnal variations in atmospheric $\delta^{18}$O-NO$_2$, including the studies you reference (Albertin et al., 2021; Albertin et al., 2024) and those conducted in Hefei (31.82 °N, 117.28 °E; Fig. 2(a); Zhang et al., 2025) and Nanchang (28.68 °N, 115.93 °E; Fig. 2(b); Cao, 2022), China. During the day, $\delta^{18}$O–NO$_2$ is significantly greater than that at night, because of the prominent $\delta^{18}$O–O$_3$ signal in the photochemical cycling of NO and NO$_2$ during the day; conversely, at night, more $\delta^{18}$O–NO$_2$ signals from emissions are present (Walters et al., 2016). However, numerous observational studies of $\delta^{18}$O–NO$_3^-$ do not reveal significant diurnal variations, especially in winter. Examples include Tianjin (39.11°N, 117.16°E; Fig. 2(c); Feng et al., 2020) in the North China Plain (NCP) and Nanjing (32.22°N, 118.75°E; Fig. 2(d); Zhang et al., 2022). This phenomenon may occur due to its longer atmospheric lifetime and diffusion. Therefore, in our seasonal study, the diurnal variation was not considered in the calculations. We have referenced the widely used $\delta^{18}$O-NOx value of 117 ± 5‰ (Michalski et al., 2014) for calculating the endmember values of the ·OH+NO$_2$ and hetN$_2$O$_5$ pathways to distinguish the contributions of the two pathways. In future studies, we will attempt to account for the diurnal variations in $\delta^{18}$O-NO$_2$ and their impact on the generated $\delta^{18}$O–NO$_3^-$.

Table R1 Values of $\delta^{18}$O from atmospheric components

| Components | Values (‰) | References |
| --- | --- | --- |
| O$_3$ | From 80 to 130 | Michalski et al., 2011 |
| O$_2$ | 23.5 | Kroopnick and Craing; 1972 |
| H$_2$O (g) in Beijing winter | −27.9 | Wen et al., 2010 |
| H$_2$O (g) in Qingdao winter | −18.6 | Wang et al., 2022 |
| ·OH in Beijing winter | From −72.4 to −64.9 | $\delta^{18}$O-OH = $\delta^{18}$O-H$_2$O$_{(g)}$ + 1000($^{18}\alpha_{X/Y}$ − 1) |
| ·OH in Qingdao winter | From −61.2 to −57.8 | (Walters and Michalski, 2016) |

[Figure]

Figure R2 The diurnal values of atmospheric $\delta^{18}O$-NO$_2$ in Hefei winter (Zhang et al., 2025) (a) and in Nanchang summer (Cao, 2022) (b), and $\delta^{18}O$-NO$_3^-$ in Tianjin winter (Feng et al., 2020) (c) and in Nanjing winter (Zhang et al., 2022) (d).

Albertin, S., Savarino, J., Bekki, S., Barbero, A., and Caillon, N.: Measurement report: Nitrogen isotopes ($\delta^{15}N$) and first quantification of oxygen isotope anomalies ($\Delta^{17}O$, $\delta^{18}O$) in atmospheric nitrogen dioxide, Atmos. Chem. Phys., 21, 10477-10497, 10.5194/acp-21-10477-2021, 2021.

Albertin, S., Savarino, J., Bekki, S., Barbero, A., Grilli, R., Fournier, Q., Ventrillard, I., Caillon, N., and Law, K.: Diurnal variations in oxygen and nitrogen isotopes of atmospheric nitrogen dioxide and nitrate: implications for tracing NOx oxidation pathways and emission sources, Atmos. Chem. Phys., 24, 1361-1388, 10.5194/acp-24-1361-2024, 2024.

Cao, L. Nitrogen and oxygen isotope tracing of urban atmospheric nitrate sources and atmospheric processes—taking Beijing and Nanchang as examples. Dissertation for master's degree of East China University of Technology.

Feng, X., Li, Q., Tao, Y., Ding, S., Chen, Y., and Li, X.: Impact of Coal Replacing Project on atmospheric fine aerosol nitrate loading and formation pathways in urban Tianjin: Insights from chemical composition and $^{15}N$ and $^{18}O$ isotope ratios, Sci. Total Environ., 708, 134797, 10.1016/j.scitotenv.2019.134797, 2020.

Kroopnick, P., and Craig, H.: Atmospheric Oxygen: Isotopic Composition and Solubility Fractionation, Science, 175, 54-55, 10.1126/science.175.4017.54, 1972.

Luo, L., Liao, T., Zhang, X., Wu, Y., Li, J., Zhang, R., Zheng, Z., and Kao, S.: Quantifying the formation pathways of nitrate in size-segregated aerosols during winter haze pollution, Gondwana Res., https://doi.org/10.1016/j.gr.2022.11.015, 2022.

Michalski, G., Bhattacharya, S. K., and Girsch, G.: NOx cycle and the tropospheric ozone isotope anomaly: an experimental investigation, Atmos. Chem. Phys., 14, 4935-4953, 10.5194/acp-14-4935-2014, 2014.

Michalski, G., Bhattacharya, S. K., DF Mase., 2011. Oxygen isotope dynamics of atmospheric nitrate and its precursor molecules. Springer Berlin Heidelberg.

Walters, W. W., and Michalski, G.: Theoretical calculation of oxygen equilibrium isotope

fractionation factors involving various NO molecules, OH, and $H_2O$ and its implications for isotope variations in atmospheric nitrate, Geochim. Cosmochim. Ac., 191, 89-101, 10.1016/j.gca.2016.06.039, 2016.

Walters, W. W., Simonini, D. S., and Michalski, G.: Nitrogen isotope exchange between NO and $NO_2$ and its implications for $\delta^{15}N$ variations in tropospheric NOx and atmospheric nitrate, Geophys. Res. Lett., 43, 440-448, 10.1002/2015GL066438, 2016.

Wang, Y., Cui, B., Li, D., Wang, Y., Yu, W., and Zong, H.: Stable Isotopes Reveal Water Vapor Sources of Precipitation over the Jiaolai Plain, Shandong Peninsula, China, Asia-Pac. J. Atmos. Sci., 58, 227-241, 10.1007/s13143-021-00253-2, 2022.

Wen, X. F., Zhang, S. C., Sun, X. M., Yu, G. R., and Lee, X.: Water vapor and precipitation isotope ratios in Beijing, China, Journal of Geophysical Research: Atmospheres, 115, 10.1029/2009JD012408, 2010.

Zhang, Y., Zhang, W., Fan, M., Li, J., Fang, H., Cao, F., Lin, Y., Wilkins, B. P., Liu, X., Bao, M., Hong, Y., and Michalski, G.: A diurnal story of $\Delta^{17}O(NO_3^-)$ in urban Nanjing and its implication for nitrate aerosol formation, npj Climate and Atmospheric Science, 5, 10.1038/s41612-022-00273-3, 2022.

Zhang, Z., Zhou, T., Jiang, Z., Ma, T., Su, G., Ruan, X., Wu, Y., Cao, Y., Wang, X., Liu, Z., Li, W., Zhang, H., Lin, M., Liu, P., and Geng, L.: High-Resolution Measurements of Multi-Isotopic Signatures ($\delta^{15}N$, $\delta^{18}O$, and $\Delta^{17}O$) of Winter $NO_2$ in a Megacity in Central China, Environ. Sci. Technol., 59, 3634-3644, 10.1021/acs.est.4c07724, 2025.

Lines 405-406: What were the "Other" pathways? Do you have an idea if their $d^{18}O$ values of the formed nitrate are close to the het pathway? If not then the model compared to the observations would appear biased towards accurately getting the het reaction pathway correct.

Reply: We apologize for not providing a sufficient explanation regarding the pathway calculations for stable isotopes. P7–P10 were roughly defined as "other" pathways (line 425–427 in the revised manuscript) in Table R2 in our previous study (Luo et al., 2020), and we compiled theoretical calculations of the thermodynamic fractionation of $\delta^{18}O$ endmember. As you noted, the endmember values of $\delta^{18}O$ were too close to those of $hetN_2O_5$; for example, in the calculations for the winter of 2018 in Qingdao, the end-member $\delta^{18}O$ value for the $hetN_2O_5$ pathway was 102.6 ± 4.2‰, whereas those for P7, P8, and P9 were 105.8 ± 4.8‰, 126.4 ± 4.8‰, and 135.0 ± 4.8‰, respectively. This consideration remains immature in actual pathway calculations; extensive numbers of researchers have primarily used $\cdot OH+NO_2$ and $hetN_2O_5$ as the two main formation pathways to calculate atmospheric $NO_3^-$ formation (Li et al., 2023; Xiao et al., 2025). We followed the two-pathway approach.

In contrast, the CMAQ model considers a comprehensive set of reaction rates and computes the refined pathway contribution, which is its principal strength. Consequently, stable isotopes, which serve as direct observational evidence, provide qualitative but robust confirmation of the validity of the CMAQ model. The two methods indicated that in Beijing during the winter of 2018, the contribution of the $hetN_2O_5$ pathway decreased with that in 2013, whereas the contribution of $hetN_2O_5$ in the coastal city of Qingdao was greater than that in the

inland city of Beijing (Figure R3).

**Table R2** Calculated $\delta^{18}O$ values of NOy for each nitrate production pathway (Luo et al., 2020).

| | Pathway | Expression |
|---|---|---|
| R1 | $NO + O_3 \rightarrow NO_2 + O_2$ | $\delta^{18}O\text{-}NO_2 = \phi\ \delta^{18}O\text{-}O_3 + (1\text{-}\phi)\ \delta^{18}O\text{-}R/HO_2$ |
| R2 | $NO + RO_2/HO_2 \rightarrow NO_2 + O_2$ | |
| R3 | $NO_2 + O_3 \rightarrow NO_3$ | $\delta^{18}O\text{-}NO_3 = 2/3\ \delta^{18}O\text{-}NO_2 + 1/3\ \delta^{18}O\text{-}O_3$ |
| R4 | $NO_2 + NO_3 \rightarrow N_2O_5$ | $\delta^{18}O\text{-}N_2O_5 = 2/5\ \delta^{18}O\text{-}NO_2 + 3/5\ \delta^{18}O\text{-}NO_3$ |
| R5 | $NO_2 + OH \rightarrow HNO_3$ | $\delta_{18}O\text{-}HNO3 = 2/3\ \delta^{18}O\text{-}NO_2 + 1/3\ \delta^{18}O\text{-}OH$ |
| R6 | $N_2O_5 + H_2O \rightarrow HNO_3$ | $\delta^{18}O\text{-}HNO_3 = 5/6\ \delta^{18}O\text{-}N_2O_5 + 1/6\ \delta^{18}O\text{-}H_2O$ |
| R7 | $NO_3 + HC/DMS \rightarrow HNO_3$ | $\delta^{18}O\text{-}HNO_3 = \delta^{18}O\text{-}NO_3$ |
| R8 | $N_2O_5 + Cl^- \rightarrow pNO_3^-$ | $\delta^{18}O\text{-}HNO_3 = \delta^{18}O\text{-}N_2O_5$ |
| R9 | $ClNO_3 + H_2O \rightarrow HNO_3$ | $\delta^{18}O\text{-}HNO_3 = 2/3\ \delta^{18}O\text{-}NO_2 + 1/3\ \delta^{18}O\text{-}O_3$ |
| R10 | $NO_2 + H_2O \rightarrow HNO_3$ | $\delta^{18}O\text{-}HNO_3 = \delta^{18}O\text{-}NO_2$ |

[Figure]

**Figure R3.** Boxplots of $\delta^{18}O$ of atmospheric $NO_3^-$ collected in Beijing in the winters of 2013 and 2018 and in Qingdao in the winter of 2018 under different nitrate conditions. The shadows of red and blue indicate the ranges of $\delta^{18}O\text{-}NO_3^-$ generated via the daytime and nocturnal pathways, respectively. Categories that share common letters do not differ in significance, which is set to 0.05

Xiao H W, Chen T S, Zhang Q J, et al. Changes in the Dominant Contributions of Nitrate Formation and Sources During Haze Episodes: Insights From Dual Isotopic Evidence[J]. Journal of Geophysical Research: Atmospheres, 2025, 130(2): e2024JD042175.

Li T, Li J, Sun Z, et al. High contribution of anthropogenic combustion sources to atmospheric inorganic reactive nitrogen in South China evidenced by isotopes[J]. Atmospheric Chemistry and Physics, 2023, 23(11): 6395-6407.

Luo L, Pan Y, Zhu R, et al. Assessment of the seasonal cycle of nitrate in PM2.5 using chemical compositions and stable nitrogen and oxygen isotopes at Nanchang, China[J]. Atmospheric Environment, 2020, 225: 117371.

Lines 415-416:   It would be important for the partitioning of $HNO_3$ to $pNO_3$, but it wouldn't have a major impact on the formation of nitrate.   Though it could influence the aerosol properties that could influence $N_2O_5$ reactions on aerosol surface for HNO3 production.

**Reply:** We appreciate your careful review of our manuscript. While $NH_3$ has a limited direct influence on nitrate formation, it is pivotal in converting $HNO_3$ to particulate nitrate ($pNO_3$) at sufficient ammonia concentrations. In the revised manuscript, we updated this statement to: " The availability of $NH_3$ was considered a critical factor governing the partitioning of $HNO_3$ to particulate nitrate ($pNO_3$)." lines 497–498.

Lines 449-450: I think it might be better worded to say $NH_3$ plays a critical role in influence particulate nitrate concentrations.   Using "formation" is slightly confusing for this work, because some much time and effort was devoted to talking about nitrate formation via oxidation chemistry, which isn't the same use of formation in this context.

**Reply:** We sincerely appreciate the reviewer's insightful comments. We have revised the sentence to more precisely describe the role of ammonia ($NH_3$) in influencing particulate nitrate concentrations. The modified sentence now reads: " In summary, $NH_3$ played a critical role in influencing particulate nitrate concentrations by affecting the gas–particle conversion of $HNO_3$ in the NCP region, although its availability was sufficient." (Lines 527-529)

Lines 458 – 459: The rate formation of HNO3 from the $NO_2$ + OH reaction: $d[HNO3]/dt = k(NO_2+OH)[NO_2][OH]$ shows that it depends on both $[NO_2]$ and $[OH]$

[Figure]

Figure 4 Spatial distributions of the $NO_2/OH$ molar ratio in the NCP region during the winters of 2013 and 2018

**Reply:** We are truly grateful for your insightful comments and constructive suggestions. The reaction rate of $HNO_3$ formation via $NO_2$ + OH ($d[HNO_3]/dt = k(NO_2 + OH)[NO_2][OH]$) is evidently dependent on the concentrations of both $NO_2$ and OH. However, $NO_2$ concentrations remain much higher than OH radical concentrations, with the $NO_2/OH$ molar ratio typically exceeding $10^8$. This result indicates that $NO_2$ is in excess during this reaction. Therefore, we conclude that the reaction rate is influenced primarily by the concentration of OH radicals.

Lines 501-505: Why did [O₃] increase during this period?

**Reply:** Currently, the increase in ozone ($O_3$) concentrations is attributed primarily to both meteorological variations and anthropogenic influences, with the latter playing a more significant role. Studies indicate that meteorological changes and anthropogenic emissions jointly drive the rise in $O_3$, with anthropogenic contributions being more substantial. For example, Li et al. (2019) reported that $O_3$ concentrations in the North China Plain (NCP) increased by 3.3 ppb yr⁻¹ ($p < 0.01$) from 2013 to 2019, with meteorological factors contributing 1.4 ppb yr⁻¹ ($p = 0.02$) and anthropogenic influences accounting for 1.9 ppb yr⁻¹ ($p < 0.01$). Similarly, Liu et al. (2020) demonstrated that, from 2013 to 2020, both meteorological variations (3.6 μg m⁻³) and anthropogenic emissions (6.7 μg m⁻³) contributed to the increase in the maximum daily 8-hour average ozone (MDA8 $O_3$) across China, with anthropogenic emissions playing a more dominant role.

Li, K., Jacob, D. J., Shen, L., Lu, X., De Smedt, I., and Liao, H.: Increases in surface ozone pollution in China from 2013 to 2019: anthropogenic and meteorological influences, Atmos. Chem. Phys., 20, 11423-11433, 10.5194/acp-20-11423-2020, 2020.

Liu, Y., Geng, G., Cheng, J., Liu, Y., Xiao, Q., Liu, L., Shi, Q., Tong, D., He, K., and Zhang, Q.: Drivers of Increasing Ozone during the Two Phases of Clean Air Actions in China 2013–2020, Environmental Science & Technology, 57, 8954-8964, 10.1021/acs.est.3c00054, 2023.

Line 510-512: "uptake" coefficient?

**Reply:** Thank you for pointing this out. We have revised the term to "uptake" coefficient on line 591.

Lines 617-618: I think this is an inadequate Data availability statement.

**Reply:** Thank you for your valuable feedback regarding the data availability statement. We have updated the data availability statement to provide more comprehensive information on how to access the data. All datasets supporting the findings of this study are available through the following channels. Primary datasets and analysis results are available from the corresponding authors upon reasonable request. All plotting data and essential research data have been deposited in a publicly accessible repository on the Baidu Cloud (Data link: https://pan.baidu.com/s/153rcdB-vTidH-14PPaXu-A; Access code: egus).

Supporting Information

**Enhanced Atmospheric Oxidation and Particle Reductions Driving Changes to Nitrate Formation Mechanisms across Coastal and Inland Regions of North China**

Zhenze Liu[1,2], Jianhua Qi[1,2], Yuanzhe Ni[1], Likun Xue[3], Xiaohuan Liu[1,2]

[1] Key Laboratory of Marine Environment and Ecology, Ministry of Education, Ocean University of China, Qingdao 266100, China

[2] Laboratory for Marine Ecology and Environmental Science, Qingdao Marine Science and Technology Center, Qingdao 266237, China

[3] Environment Research Institute, Shandong University, Qingdao, Shandong, 266237, China

*Correspondence to*: Jianhua Qi (qjianhua@ouc.edu.cn), Xiaohuan Liu (liuxh1983@ouc.edu.cn)

**Text S1**

Our isotope blank measurements followed the same procedure as the sample isotope analysis. Specifically, in the sample measurements, after purging with high-purity nitrogen, 20 nmol of nitrogen was added to the headspace vial containing the *Pseudomonas aureofaciens* (ATCC13985) strain. For the blank measurements, no sample was added, and after 24 hours, 10 M NaOH was directly injected to quench the reaction before the analysis. The peak area in the chromatogram represents the absolute amount of $N_2O$ reduced by the strain, and the $\delta^{15}N$ and $\delta^{18}O$ values correspond to the $\delta^{15}N$ and $\delta^{18}O$ values of the sample. The peak area for the samples was around 10, while the peak areas for the two blank measurements were only 0.371 and 0.336, indicating an influence on the isotope values of less than 5%, which is negligible and thus not considered.

[Figure]

**Figure S1** The diurnal values of atmospheric $\delta^{18}O-NO_2$ in Hefei winter (Zhang et al., 2025) (a) and in Nanchang summer (Cao, 2022) (b), and $\delta^{18}O-NO_3^-$ in Tianjin winter (Feng et al., 2020) (c) and in Nanjing winter (Zhang et al., 2022) (d).

**Text S2**

In most studies, the tropospheric $\delta^{15}$N-NOx was often assumed as 0‰ following Walters and Michalski (2016), Luo et al. (2023) and Deng et al. (2024). In addition, the tropospheric $\delta^{18}$O-H$_2$O$_{(g)}$ in Beijing in winter was determined as -27.9‰ in Wen et al. (2010), and in Qingdao, it was determined as -18.6‰ in Wang et al. (2022). The tropospheric $\delta^{18}$O-NOx ranged from 112‰ to 122‰ (Michalski et al., 2014; Walters and Michalski, 2016). The $f_{NO2}$ values in Beijing and Qingdao were 0.655 (Luo et al., 2023) and 0.786 (Lian et al., 2022) in winter, respectively.

$$\delta^{15}N - NO_3^- = \gamma \times [\delta^{15}N - NO_3^-]_{OH} + (1 - \gamma) \times [\delta^{15}N - NO_3^-]_{N_2O_5}$$
$$= \gamma \times [\delta^{15}N - HNO_3]_{OH} + (1 - \gamma) \times [\delta^{15}N - HNO_3]_{N_2O_5} \tag{S1}$$

$$\delta^{18}O - NO_3^- = \gamma \times [\delta^{18}O - NO_3^-]_{OH} + (1 - \gamma) \times [\delta^{18}O - NO_3^-]_{N_2O_5}$$
$$= \gamma \times [\delta^{18}O - HNO_3]_{OH} + (1 - \gamma) \times [\delta^{18}O - HNO_3]_{N_2O_5} \tag{S2}$$

$$[\delta^{15}N - HNO_3]_{OH} = \delta^{15}N - NO_2$$
$$= 1000 \times \left[ \frac{\left( {}^{15}a_{NO_2/NO} - 1 \right)\left( 1 - f_{NO_2} \right)}{\left( 1 - f_{NO_2} \right) + \left( {}^{15}a_{NO_2/NO} \times f_{NO_2} \right)} \right] + \delta^{15}N - NOx \tag{S3}$$

$$[\delta^{15}N - HNO_3]_{N_2O_5} = 1000 \times \left( {}^{15}a_{N_2O_5/NO_2} - 1 \right) + \delta^{15}N - NOx \tag{S4}$$

$$[\delta^{18}O - HNO_3]_{OH} = \frac{2}{3} \times [\delta^{18}O - NO_2]_{OH} + \frac{1}{3} \times [\delta^{18}O - OH]_{OH}$$
$$= \frac{2}{3} \times \left[ \frac{1000 \times \left( {}^{18}\alpha_{NO_2/NO} - 1 \right) \times \left( 1 - f_{NO_2} \right)}{\left( 1 - f_{NO_2} \right) + \left( {}^{18}a_{NO_2/NO} \times f_{NO_2} \right)} + [\delta^{18}O - NOx] \right]$$
$$+ \frac{1}{3} \times \left[ \left( \delta^{18}O - H_2O_{(g)} \right) + 1000 \times \left( {}^{18}\alpha_{OH/H_2O_{(g)}} - 1 \right) \right] \tag{S5}$$

$$[\delta^{18}O - HNO_3]_{N_2O_5} = \delta^{18}O - NO_2 + 1000 \times \left( {}^{18}\alpha_{N_2O_5/NO_2} - 1 \right) \tag{S6}$$

$$1000\left( {}^{m}a_{x/y} - 1 \right) = \frac{A}{T^4} \times 10^{10} + \frac{B}{T^3} \times 10^8 + \frac{C}{T^2} \times 10^6 + \frac{D}{T} \times 10^4 \tag{S7}$$

**Table S1 Values of $\delta^{18}O$ from atmospheric components**

| Components | Values (‰) | References |
|---|---|---|
| $O_3$ | From 80 to 130 | Michalski et al., 2011 |
| $O_2$ | 23.5 | Kroopnick and Craing; 1972 |
| $H_2O$ (g) in Beijing winter | −27.9 | Wen et al., 2010 |
| $H_2O$ (g) in Qingdao winter | −18.6 | Wang et al., 2022 |
| ·OH in Beijing winter | From −72.4 to −64.9 | $\delta^{18}O\text{-OH} = \delta^{18}O\text{-}H_2O_{(g)} + 1000(^{18}\alpha_{X/Y} - 1)$ |
| ·OH in Qingdao winter | From −61.2 to −57.8 | (Walters and Michalski, 2016) |

**Table S2 $^{15}\alpha_{A/B}$ and $^{18}\alpha_{A/B}$ regression coefficients as a function of the temperature (150 K ≤ T ≤ 450 K) (Walters and Michalski, 2015, 2016)**

|  |  | A | B | C | D |
|---|---|---|---|---|---|
| $^{15}\alpha_{A/B}$ | $N_2O_5/NO_2$ | 0.69398 | -1.9859 | 2.3876 | 0.16308 |
|  | $NO_2/NO$ | 3.8834 | -7.7299 | 6.0101 | -0.17928 |
| $^{18}\alpha_{A/B}$ | $NO/NO_2$ | -0.04129 | 1.1605 | -1.8829 | 0.74723 |
|  | $\cdot OH/H_2O_{(g)}$ | 2.1137 | -3.8026 | 2.5653 | 0.5941 |
|  | $N_2O_5/NO_2$ | -0.54136 | 0.13073 | 1.2477 | -0.1272 |

**Table S3 Equations for calculating the statistical evaluation indices**

| Statistical index | Formula |
|---|---|
| 1. Mean Bias | $$MB = \frac{1}{N}\sum_{1}^{N}\left(Sim - Obs\right)$$ |
| 2. Root Mean Square Error | $$RMSE = \sqrt{\frac{1}{N}\sum_{1}^{N}(Sim-Obs)^2}$$ |
| 3. Index of agreement, IOA | $$IOA = 1 - \frac{\sum_{i=1}^{N}(Sim-Obs)^2}{\sum_{i=1}^{N}(|Sim-\overline{Obs}|+|Obs-\overline{Obs}|)^2}$$ |
| 4. Normalized Mean Bias | $$NMB = \frac{1}{N}\sum_{1}^{N}\left(\frac{Sim-Obs}{Obs}\right)$$ |
| 5. Normalized Mean Error | $$NME = \frac{1}{N}\sum_{1}^{N}\left|\frac{Sim-Obs}{Obs}\right|$$ |
| 6. Correlation coefficient (R) | $$R = \frac{1}{N}\sum_{i=1}^{N}\left[\frac{(Sim-\overline{Sim})(Obs-\overline{Obs})}{S_p S_o}\right]$$ $$S_P = \left[\frac{1}{N}\sum_{i=1}^{N}(Sim-\overline{Sim})^2\right]^{\frac{1}{2}}$$ $$S_o = \left[\frac{1}{N}\sum_{i=1}^{N}(Obs-\overline{Obs})^2\right]^{\frac{1}{2}}$$ |

**Table S4 Sources of nitrate observation data for the winter of 2013 and the winter of 2018 in the NCP**

| City | Winter, 2013 | Winter, 2018 |
|------|------|------|
| Beijing | Song et al. (2019) | Fan et al. (2020) |
| Tianjin | Yao et al. (2020) | Observation |
| Shijiahzuang | Wang et al. (2016) | Zhou et al. (2020) |
| Jinan | Cheng et al. (2021) | Observation |
| Zhengzhou | Wei et al. (2019) | Dong et al. (2020) |
| Qingdao | Observation | Observation |
| Yantai | / | / |

The $NO_3^-$ observation data collected during the winter of 2018 for Tianjin were sourced from direct observations by the group of Li Xiaodong at Tianjin University (sampling site: Building 19 rooftop, Tianjin University; coordinates: 39.11°N, 117.16°E). The $NO_3^-$ observation data collected during the winter of 2018 in Jinan were sourced from observations by the group of Xue Likun at Shandong University (sampling site: Jinan City Environmental Monitoring Station; coordinates: 36.66°N, 117.05°E). For Qingdao, $NO_3^-$ observation data for both the winter of 2018 and the winter of 2013 were derived from our own observations.

[Figure]

**Figure S2 Concentrations of PM₂.₅ and its components in seven major cities in the NCP region during the winters of 2013 and 2018**

[Figure]

**Figure S3 Spatial distribution of the NO₂/OH molar ratio in the NCP region during the winters of 2013 and 2018**

[Figure]

**Figure S4 Relative humidity in the NCP and seven major cities (2013, 2018)**

**Reference**

Cheng, M., Tang, G., Lv, B., Li, X., Wu, X., Wang, Y., and Wang, Y.: Source apportionment of PM$_{2.5}$ and visibility in Jinan, China, Journal of Environmental Sciences, 102, 207-215, doi:10.1016/j.jes.2020.09.012, 2021.

Deng, M., Wang, C., Yang, C., Li, X., and Cheng, H.: Nitrogen and oxygen isotope characteristics, formation mechanism, and source apportionment of nitrate aerosols in Wuhan, Central China, Science of The Total Environment, 921, 170715, doi:10.1016/j.scitotenv.2024.170715, 2024.

Dong, Z., Su, F., Zhang, Z., and Wang, S.: Observation of chemical components of PM$_{2.5}$ and secondary inorganic aerosol formation during haze and sandy haze days in Zhengzhou, China, J Journal of Environmental Sciences, 88, 316-325, doi:10.1016/j.jes.2019.09.016, 2020.

Fan, M.-Y., Zhang, Y.-L., Lin, Y.-C., Cao, F., Zhao, Z.-Y., Sun, Y., Qiu, Y., Fu, P., and Wang, Y.: Changes of Emission Sources to Nitrate Aerosols in Beijing After the Clean Air Actions: Evidence From Dual Isotope Compositions, Journal of Geophysical Research: Atmospheres, 125, e2019JD031998, doi:10.1029/2019JD031998, 2020.

Lian, C., Wang, W., Chen, Y., Zhang, Y., Zhang, J., Liu, Y., Fan, X., Li, C., Zhan, J., Lin, Z., Hua, C., Zhang, W., Liu, M., Li, J., Wang, X., An, J., and Ge, M.: Long-term winter observation of nitrous acid in the urban area of Beijing, Journal of Environmental Sciences, 114, 334-342, doi:10.1016/j.jes.2021.09.010, 2022.

Luo, L., Wu, S., Zhang, R., Wu, Y., Li, J., and Kao, S.-j.: What controls aerosol δ15N-NO3−? NOx emission sources vs. nitrogen isotope fractionation, Science of The Total Environment, 871, 162185, doi:10.1016/j.scitotenv.2023.162185, 2023.

Michalski, G., Bhattacharya, S. K., and Girsch, G.: NOx cycle and the tropospheric ozone isotope anomaly: an experimental investigation, Atmos. Chem. Phys., 14, 4935-4953, doi:10.5194/acp-14-4935-2014, 2014.

Song, W., Wang, Y.-L., Yang, W., Sun, X.-C., Tong, Y.-D., Wang, X.-M., Liu, C.-Q., Bai, Z.-P., and Liu, X.-Y.: Isotopic evaluation on relative contributions of major NOx sources to nitrate of PM2.5 in Beijing, Environmental Pollution, 248, 183-190, doi:10.1016/j.envpol.2019.01.081, 2019.

Walters, W. W. and Michalski, G.: Theoretical calculation of nitrogen isotope equilibrium exchange fractionation factors for various NOy molecules, Geochimica et Cosmochimica Acta, 164, 284-297, doi:10.1016/j.gca.2015.05.029, 2015.

Walters, W. W. and Michalski, G.: Theoretical calculation of oxygen equilibrium isotope fractionation factors involving various NOy molecules, OH, and $H_2O$ and its implications for isotope variations in atmospheric nitrate, Geochimica et Cosmochimica Acta, 191, 89-101, doi:10.1016/j.gca.2016.06.039, 2016.

Wang, X., Zhou, Y., Cheng, S., and Wang, G.: Characterization and regional transmission impact of water-soluble ions in $PM_{2.5}$ during winter in typical cities, China Environmental Science, 36, 2289-2296, 2016.

Wang, Y., Cui, B.-l., Li, D.-s., Wang, Y.-x., Yu, W.-x., and Zong, H.-h.: Stable Isotopes Reveal Water Vapor Sources of Precipitation over the Jiaolai Plain, Shandong Peninsula, China, Asia-Pacific Journal of Atmospheric Sciences, 58, 227-241, doi:10.1007/s13143-021-00253-2, 2022.

Wei, X., Gao, m., and Tong, j.: Characterization of water-soluble ions of $PM_{2.5}$ in Zhengzhou, Chinese journal of quantum, 36, 495-499, 2019.

Wen, X.-F., Zhang, S.-C., Sun, X.-M., Yu, G.-R., and Lee, X.: Water vapor and precipitation isotope ratios in Beijing, China, Journal of Geophysical Research: Atmospheres, 115, doi:10.1029/2009JD012408, 2010.

Yao, Q., Liu, Z., Han, S., Cai, Z., Liu, J., Hao, T., Liu, J., Huang, X., and Wang, Y.: Seasonal variation and secondary formation of size-segregated aerosol water-soluble inorganic ions in a coast megacity of North China Plain, Environmental Science and Pollution Research, 27, 26750-26762, doi:10.1007/s11356-020-09052-0, 2020.

Zhou, J., Duan, J., Wang, J., Liu, H., LI, M., and Jin, W.: Analysis of pollution characteristics and sources of $PM_{2.5}$ during heavey pollution in Shijiazhuang city around New Year's Day 2019, Environmental Science, 41, 39-49, doi:10.13227/j.hjkx.201906085, 2020.

Cao, L. Nitrogen and oxygen isotope tracing of urban atmospheric nitrate sources and atmospheric processes—taking Beijing and Nanchang as examples. Dissertation for master's degree of East China University of Technology.

Feng, X., Li, Q., Tao, Y., Ding, S., Chen, Y., and Li, X.: Impact of Coal Replacing Project on atmospheric fine aerosol nitrate loading and formation pathways in urban Tianjin: Insights from chemical composition and [15]N and [18]O isotope ratios, Sci. Total Environ., 708, 134797, 10.1016/j.scitotenv.2019.134797, 2020.

Zhang, Y., Zhang, W., Fan, M., Li, J., Fang, H., Cao, F., Lin, Y., Wilkins, B. P., Liu, X., Bao, M., Hong, Y., and Michalski, G.: A diurnal story of $\Delta^{17}O(NO_3^-)$ in urban Nanjing and its implication for nitrate

aerosol formation, npj Climate and Atmospheric Science, 5, 10.1038/s41612-022-00273-3, 2022.

Zhang, Z., Zhou, T., Jiang, Z., Ma, T., Su, G., Ruan, X., Wu, Y., Cao, Y., Wang, X., Liu, Z., Li, W., Zhang, H., Lin, M., Liu, P., and Geng, L.: High-Resolution Measurements of Multi-Isotopic Signatures ($\delta^{15}$N, $\delta^{18}$O, and $\Delta^{17}$O) of Winter $NO_2$ in a Megacity in Central China, Environ. Sci. Technol., 59, 3634-3644, 10.1021/acs.est.4c07724, 2025.

Kroopnick, P., and Craig, H.: Atmospheric Oxygen: Isotopic Composition and Solubility Fractionation, Science, 175, 54-55, 10.1126/science.175.4017.54, 1972.

Michalski, G., Bhattacharya, S. K., and Girsch, G.: NOx cycle and the tropospheric ozone isotope anomaly: an experimental investigation, Atmos. Chem. Phys., 14, 4935-4953, 10.5194/acp-14-4935-2014, 2014.

Michalski, G., Bhattacharya, S. K., DF Mase., 2011. Oxygen isotope dynamics of atmospheric nitrate and its precursor molecules. Springer Berlin Heidelberg.

Walters, W. W., and Michalski, G.: Theoretical calculation of oxygen equilibrium isotope fractionation factors involving various NO molecules, OH, and $H_2O$ and its implications for isotope variations in atmospheric nitrate, Geochim. Cosmochim. Ac., 191, 89-101, 10.1016/j.gca.2016.06.039, 2016.

---

## Author Comment (AC3)

**Responses to reviewers (original comments by reviewers are in blue).**

**Reviewer #1:**

Comments on Liu et al., 2024

General summary:

Overall, I think there is clear evidence that a lot of work has been done and much data produced. The writing is clear. The technical analysis seems logical and results reported well, although I cannot speak much to the actual modelling technical components as that is outside my expertise.

My primary critique revolves around the structure and narrative of the paper. Namely, there is so much information presented that it is sometimes difficult to keep track of what the overall main point is that the authors are trying to get across. I think this is exemplified in two ways: first, the title itself about "Exploring…mechanisms", while accurate to the content, highlights that there is less focus on a definitive point/conclusion than just summarizing and overviewing a lot of results from modelling. Second, there is a massive Results section, but no Discussion. There is some discussion happening within the Results, but the paper would likely improve by having less space dedicated to describing every result from the modelling and more space on what those results mean for things like policy implications, the need to treat interior cities differently from coastal ones in regulations, etc. Overall, I would suggest that the authors take a fresh look at the content of the paper as a whole narrative and re-evaluate if everything in the results needs to be included and described at the detail it currently is at.

That said, I do think that there is a good information here, and a valuable contribution. And I have largely minor critiques on the technical side and scientific content. But for the authors' sake, I think the paper's eventual impact could be greatly improved by focusing the narrative, simplifying/summarizing some of the base results further, and speaking more to the broader implications of this work.

**Reply:** Thank you for your valuable suggestions for improvement. We agree that the current results section had been overly detailed, which may have detracted from the overall focus of the paper. In response to your feedback, we have implemented the following adjustments:

**1. Revised the Title**

In line with your suggestions, we had revised the title into: "Enhanced Atmospheric Oxidation and Particle Reductions Driving Changes to Nitrate Formation Mechanisms Across Coastal and Inland Regions of North China."

**2. Streamlined the Results Section**

In Section 3.2, we removed some comparisons with previous literature to highlight the unique findings of this study. We streamlined the content by summarizing key trends in a more concise format, with a particular focus on the differences between inland and coastal regions. Additionally, we merged relevant figures, as shown in Figure 11.

**3. Enhanced the Discussion**

To enhance the discussion, we enriched the results section with additional commentary to emphasize the broader implications and scientific significance of our findings. In Section 3.4 (lines 413–489), we enhanced the discussion of model-isotope comparisons and their implications. Specifically, we proposed concrete policy recommendations, such as intensifying control measures targeting photochemical reactions in inland cities (lines 584–588). We proposed that coastal cities

should focus more on regulations related to aerosol surface area (lines 630–636).

Additionally, in Section 3.6, we conducted a comparative analysis of emission reduction experiments between Beijing and Qingdao, providing targeted strategies for improving air quality based on the unique characteristics of each city.

**Major points:**

I cannot see a supplemental section on the Preprint review page? This made it impossible to examine things referenced in the methods. (Apologies if this was a mistake on my part).

Results: There is a huge amount of information and data both presented and discussed. While I commend the authors for being upfront with their data, it can be a bit overwhelming at times and causes some of the focus to be lost. I would recommend looking back over this section to determine what exactly are the main points and stories you are aiming to get across, and pare down any information and number discussion that distracts away from those points. Perhaps greater summarization of regional trends (e.g., inland vs. coastal) rather than relaying data from multiple cities would help focus the section, too. You do this already some by focusing on Beijing vs. Qingdao, but even further summarization/simplification could help in some spots.

There are a lot of figures, and many of them are similar in theme (e.g., comparing an atmospheric chemical in 2013 and in 2018 and their difference). Perhaps combining many of these into a single, larger figure would be more effective as the reader could cross compare more easily and not hit figure fatigue.

Data availability: This is an unacceptable statement for data availability, as per ACP standards. Data are to be hosted in a publicly accessible location. See further guidance from https://www.atmospheric-chemistry-and-physics.net/policies/data_policy.html:

If the data are not publicly accessible at the time of final publication, the data statement should describe where and when they will appear, and provide information on how readers can obtain the data until then. Nevertheless, authors should make such embargoed data available to reviewers during the review process in order to foster reproducibility. The Copernicus review system allows to define such assets as 'access limited to reviewers' and reviewers must then sign that they will use such data only for the purpose of reviewing without making copies, sharing, or reusing.

In rare cases where the data cannot be deposited publicly (e.g., because of commercial constraints), a detailed explanation of why this is the case is required. The data needed to replicate figures in a paper should in any case be publicly available, either in a public database (strongly recommended), or in a supplement to the paper.

**Reply:** Thank you for your valuable feedback. In response, we have made the following revisions:

**1. Supplementary Materials**

We have submitted the supplementary materials as required and ensured that all relevant content is clearly labeled. It appears that, perhaps due to a system error, the reviewer may not have been able to access these materials. To resolve this issue, we have added the supplementary materials at the end of the document with all references clearly indicated. We will verify the completeness of these materials in our submission of revised version. We kindly invite you to review them and welcome any further comments

**2. Results Section Length**

Thank you for highlighting the issue of excessive detail in the results section. We have carefully

revised this section to eliminate repetitive descriptions of secondary information and city-specific data. Specifically, in Section 3.2, we have streamlined the content by summarizing key trends in a more concise format, particularly focusing on the differences between inland and coastal regions. In Section 3.4, we have added detailed comparisons and summaries of model simulations and isotope data for Beijing and Qingdao. Additionally, in Section 3.6, we have placed greater emphasis on the comparative analysis of emission reductions between Beijing and Qingdao, ensuring a clearer and more focused narrative.

**3. Number of Charts**

Your suggestion regarding the number of charts is greatly appreciated. To streamline the presentation and enhance data comparability, we have merged charts with similar themes. The chemical differences in air composition between 2013 and 2018, including the GR, HONO, $N_2O_5$, and $NO_x$ concentrations over the North China Plain during winter, have been merged into a single figure (Figure 11).

**4. Data Availability Statement**

Thank you for your valuable feedback. We recognize that the initial statement may have lacked sufficient detail. We have updated the data availability statement to provide more comprehensive information on how to access the data. All datasets supporting the findings of this study are available through the following channels. Primary datasets and analysis results are available from the corresponding authors upon reasonable request. All plotting data and essential research data have been deposited in a publicly accessible repository on the Baidu Cloud (Data link: https://pan.baidu.com/s/153rcdB-vTidH-14PPaXu-A; Access code: egus).

**Specific points:**

71: Is this coastal or inland Greenland?

**Reply:** Thank you for your question. The Greenland research station mentioned in the paper is located at (72.6°N, 38.5°W) within Greenland's polar environment, and it is geographically closer to the coastal regions.

73: What is it about the air mass origin that affects the nitrate formation? Or why is this being set apart and discussed here after the review of the coastal vs. inland cities? Isn't air mass origin also the primary reason for those differences? The structure of the paragraph is just confusing me a little bit here.

**Reply:** Thank you for the question. Regarding the influence of air mass origin on nitrate formation, we have addressed this issue separately in the paper because the source of the air mass is a crucial factor driving the differences between coastal and inland cities, that impacts the underlying mechanisms, specifically the effects of pollutant composition and humidity on nitrate formation pathways.

Specifically, marine air masses are typically associated with higher humidity, lower NOx mixing ratios, and longer $NO_3$ radical lifetimes, which significantly promote reactions via the het$N_2O_5$ pathway. In contrast, continental air masses usually exhibit higher NOx mixing ratios and $NH_3$ concentrations, which are more favorable for nitrate formation through the OH + $NO_2$ pathway. To better understand these regional differences, we have compared the nitrate formation mechanisms between coastal and inland cities and further investigated the influence of air mass origin in Section 3.5.3 (lines 608–611). This analysis provides a more comprehensive understanding of the underlying

driving mechanisms behind these regional disparities.

Fig. 2: Data source for terrain heights should be cited
**Reply:** Thank you for your suggestion. We have supplemented a citation for the terrain elevation data in the manuscript (lines 144–145). The data are sourced from the GEBCO Compilation Group (2024) GEBCO 2024 Grid (doi:10.5285/1c44ce99-0a0d-5f4f-e063-7086abc0ea0f).

165: Was there a specific data network that you were sourcing within that website? For example, that website is just a portal to access many different data networks, such as WMO and GHCN, and if you know the exact data source network, that could be cited here and be more clear.
**Reply:** We have clarified in the manuscript (line 146) that the terrain elevation data were obtained from the GEBCO website (https://www.gebco.net/data_and_products/gridded_bathymetry_data/) which provides access to the GEBCO 2024 grid dataset.

168: A little more information about these 68 stations would be beneficial, such as are they all within a specific region/geographic bounds? Were there any selection criteria applied to choose the stations?

[Figure]

**Reply:** These 68 observation stations were all located within the simulated D02 domain, which corresponded to the North China region. The selection of these stations was based on the availability of observational data. Specifically, all stations within this area that had observational data were included.

185: Just to confirm, are all the instrumentation specifics the same that you used here as in this cited paper? You might add a brief line or addition to the end of the sentence currently ending in "denitrifier method" to add the instrumentation used, so that the reader doesn't have to go look that basic information up in another paper.
**Reply:** Thank you for your valuable feedback. We now specify that the instrument used was the Gasbench-IRMS (Delta V, Thermo Scientific), which is the same as the one employed in our previous research. This clarification was added at the end of the sentence (lines 210–212) in the revised manuscript.

Revised sentence: The $\delta^{18}O$ and $\delta^{15}N$ values of $NO_3^-$ in the TSP samples were determined via the bacterial denitrifier method (Casciotti et al., 2002; Sigman et al., 2001) with the Gasbench-IRMS system (Delta V model, Thermo Scientific).

204: I think that some more information needs to be given here on how you used these indicators to evaluate the simulation effect. You have cited some proposed benchmarks, but it isn't clear to me readily how you will be using this information in your paper. In a very soon following section (3.1) about model evaluation where you present simulated values and some of the benchmarks, I was able to eventually infer how you were doing the evaluation, but it should really be more explicitly clear in the methodology.

**Reply:** Thank you for your comment. We have included several evaluation metrics and benchmarks in the methodology section (Table 1 and Table 2); however, due to the extensive number of evaluation formulas, we have provided a comprehensive description of the calculation methods and formulas in the supporting information. You can find these details in Table S3 of the supporting information.

220: I'm unclear exactly how the numbers being discussed here from the 68 sites were gathered and compared. Are these pairwise calculations, or overall means, or involving some sort of spatial dimension, etc? Are the comparisons all at hourly resolution, or aggregated to daily or something else? There needs to be more clarity on this, likely in the methodology of the 2.5 section. Also, how did you evaluate parameters that lacked cited benchmarks (perhaps something else that could be included in the methodology?)? For example, some of the Pearson correlation coefficients are somewhat low, for wind especially.

**Reply:** Thank you for your questions. For the data collected from the 68 observation stations, comparisons were conducted at on an hourly basis. At each station, the observed value was compared with the simulated value from the corresponding model grid cell on a point-by-point, and hour-by-hour basis. These paired comparisons were then aggregated across all 68 stations, and statistical evaluation metrics (e.g., MB, RMSE, and IOA) were generated from the entire dataset to provide an overall assessment of the model's performance.

To make these comparisons, we used the grid cell corresponding to the geographic coordinates of each observation point. This process ensured precise spatial and temporal alignment between the model simulation data and the observational data. Both data processing and comparisons were conducted on an hourly basis, without any daily averaging or other forms of temporal aggregation. Further details on this methodology can be found in Section 2.5 (lines 249–254) of the revised manuscript.

Model evaluation is both a critical and challenging part of model research. The current evaluation framework is based on findings from numerous studies. The relevant literature indicates that the current evaluation metrics are sufficient for analyzing model performance. We aim to further supplement and refine our model evaluation process through these calculations. Regarding the missing cited benchmarks, commonly used methods in the literature, such as those for wind speed and wind direction, were adopted, and we acknowledged that the observed underestimation is a common phenomenon in the model. For instance, the calculation of the Pearson correlation coefficient is intended to assist in assessing model performance. Although some parameters, particularly the correlation coefficient for wind speed, yield lower values, this phenomenon is consistent with the well-documented tendency of the WRF model to overestimate wind speed, which is an issue that is frequently reported in the literature (Tan et al., 2017; Jacobson and Kaufman, 2006). Moreover, evaluating wind direction data is complicated by its cyclic nature (since 360° and 0° represent the same direction), which may lead to some deviations in the correlation and error

metrics.

Tan, J., Zhang, Y., Ma, W., Yu, Q., Wang, Q., Fu, Q., Zhou, B., Chen, J., and Chen, L.: Evaluation and potential improvements of WRF/CMAQ in simulating multi-levels air pollution in megacity Shanghai, China, Stochastic Environmental Research and Risk Assessment, 31, 2513-2526, doi:10.1007/s00477-016-1342-3, 2017.

Jacobson, M. Z. and Kaufman, Y. J.: Wind reduction by aerosol particles, Geophysical Research Letters, 33, doi:10.1029/2006GL027838, 2006.

Section 3.4: The model has output for "Others" but your isotopic method doesn't. However, I don't see any discussion of this in this section, but I feel it needs addressed in some form. If 5-8% of reactions are "others" in the model, but you don't distinguish those in the isotopic method, does that mean that you assume you are attributing those "others" reactions to either OH+NO2 or hetN2O5? Is that baked into the uncertainties in any way, or handled specifically?

**Reply:**

Thank you for your valuable comments. We apologize for the lack of detailed information on our methods, which may have caused confusion.

The CMAQ model took a comprehensive approach by including the contributions of 5-8% of "other" pathways, which was its main advantage. Conversely in our isotopic calculations, we focused only on the $\cdot OH+NO_2$ and $hetN_2O_5$ pathways because the $\delta^{18}O$ end-member values for multiple reaction pathways were quite similar. Hence, those 5–8% of "other" pathways were attributed to the $\cdot OH+NO_2$ and $hetN_2O_5$ pathways in our isotope calculations, potentially resulting in an overestimation of these two pathways compared to the CMAQ model. Notably, that the uncertainty in the pathway contribution calculations based on stable isotopes was approximately 10–30%. Nevertheless, as a receptor model, the isotopic method, based on observational data, provided robust evidence that supported the general patterns revealed by CMAQ (Figure R1). Additionally, we supplemented isotope calculation uncertainties in the manuscript and added a detailed comparison and discussion of both isotopic and CMAQ model results in the relevant section. A detailed explanation follows.

In our previous research, we compiled theoretical calculations of the thermodynamic fractionation of $\delta^{18}O$ for the commonly considered atmospheric $NO_3^-$ formation pathways (Table R1), and P7, P8, and P9 were considered as the "other" reactions (Luo et al., 2020). However, due to the end-member values of $\delta^{18}O$ being too close to those of $hetN_2O_5$, such as in the calculations for the winter of 2018 in Qingdao, the end-member $\delta^{18}O$ value for the $hetN_2O_5$ pathway was $102.6 \pm 4.2$ ‰. Conversely, the end-member values for P7, P8, and P9 were $105.8 \pm 4.8$ ‰, $126.4 \pm 4.8$ ‰, and $135.0 \pm 4.8$ ‰, respectively. This consideration was immature in actual pathway calculations; in more extensive research (Table R2), researchers primarily consider $\cdot OH+NO_2$ and $hetN_2O_5$ the two main formation pathways when calculating atmospheric $NO_3^-$ formation.

In our isotope analysis, we employed the Stable Isotope Mixing Model in R (SIAR) to determine the relative contributions of different formation pathways to atmospheric $NO_3^-$. The main sources of uncertainty stem from the standard deviations (SD) in each pathway's end-member calculations and the posterior probability distributions generated by the model. As mentioned before, focusing on the most dominant pathways, rather than multiple pathways (Zhang et al., 2021; Luo et al., 2022), enhances the model's reliability and reduces overall uncertainty. The posterior probability for each

pathway's contribution is obtained using the Hilborn sampling-importance-resampling method. For detailed information about the model framework and computational methods, please refer to Moore and Semmens (2008). We have added an explanation of model uncertainties in the Methods section (lines 237–239) and in Section 3.4 (lines (413–430) of the manuscript.

[Figure]

**Figure R1.** Boxplot of $\delta^{18}O$ of atmospheric $NO_3^-$ collected in Beijing in the winters of 2013 and 2018 and in Qingdao in the winter of 2018 under different nitrate conditions. The shadows of red and blue indicate the ranges of $\delta^{18}O–NO_3^-$ generated via the daytime and nocturnal pathways, respectively. Categories that share common letters do not differ in significance, which is set to 0.05.

**Table R1.** Calculated $\delta^{18}O$ values of NOy for each nitrate production pathway (Luo et al., 2020).

|     | Pathway | Expression |
|-----|---------|------------|
| R1  | $NO + O_3 \rightarrow NO_2 + O_2$ | $\delta^{18}O\text{-}NO_2 = \phi\, \delta^{18}O\text{-}O_3 + (1\text{-}\phi)\, \delta^{18}O\text{-}R/HO_2$ |
| R2  | $NO + RO_2/HO_2 \rightarrow NO_2 + O_2$ | |
| R3  | $NO_2 + O_3 \rightarrow NO_3$ | $\delta^{18}O\text{-}NO_3 = 2/3\, \delta^{18}O\text{-}NO_2 + 1/3\, \delta^{18}O\text{-}O_3$ |
| R4  | $NO_2 + NO_3 \rightarrow N_2O_5$ | $\delta^{18}O\text{-}N_2O_5 = 2/5\, \delta^{18}O\text{-}NO_2 + 3/5\, \delta^{18}O\text{-}NO_3$ |
| R5  | $NO_2 + OH \rightarrow HNO_3$ | $\delta_{18}O\text{-}HNO3 = 2/3\, \delta^{18}O\text{-}NO_2 + 1/3\, \delta^{18}O\text{-}OH$ |
| R6  | $N_2O_5 + H_2O \rightarrow HNO_3$ | $\delta^{18}O\text{-}HNO_3 = 5/6\, \delta^{18}O\text{-}N_2O_5 + 1/6\, \delta^{18}O\text{-}H_2O$ |
| R7  | $NO_3 + HC/DMS \rightarrow HNO_3$ | $\delta^{18}O\text{-}HNO_3 = \delta^{18}O\text{-}NO_3$ |
| R8  | $N_2O_5 + Cl^- \rightarrow pNO_3^-$ | $\delta^{18}O\text{-}HNO_3 = \delta^{18}O\text{-}N_2O_5$ |
| R9  | $ClNO_3 + H_2O \rightarrow HNO_3$ | $\delta^{18}O\text{-}HNO_3 = 2/3\, \delta^{18}O\text{-}NO_2 + 1/3\, \delta^{18}O\text{-}O_3$ |
| R10 | $NO_2 + H_2O \rightarrow HNO_3$ | $\delta^{18}O\text{-}HNO_3 = \delta^{18}O\text{-}NO_2$ |

**Table R2.** Comparison of the contributions of major atmospheric $NO_3^-$ formation pathways in China

| Region | City | Lon (°E) | Lat (°N) | Method | Year | Season | ·OH+NO$_2$ | hetN$_2$O$_5$ (%) | NO$_3$·+HC | Reference |
|---|---|---|---|---|---|---|---|---|---|---|
| NCP | Zibo | 118.0547 | 36.8258 | Δ$^{17}$O | 2022-2023 | Winter | 17.7 | 32.6 | 49.7 | Feng et al., 2023 |
| | Zhoukou | 114.6464 | 33.6033 | | | | 53.4 | 31.1 | 15.5 | |
| | Beijing | 117.1231 | 39.0601 | Δ$^{17}$O | 2021 | Winter | 18.0 | 29.1 | 52.9 | Yan et al., 2023 |
| | | | | | | Spring | 26.9 | 35.2 | 37.9 | |
| | Beijing | 116.3660 | 39.9746 | δ$^{15}$N and δ$^{18}$O | 2017-2018 | Winter | 17.4 | 61.9 | 20.7 | Luo et al., 2023 |
| | Tianjin | 117.4177 | 39.5557 | Δ$^{17}$O | 2022 | Winter | 29.7 | 34.8 | 35.5 | Zhang et al., 2024 |
| | | | | | | Spring | 51.0 | 31.9 | 17.0 | |
| | | | | | | Summer | 41.8 | 33.6 | 24.6 | |
| | | | | | | Autumn | 13.2 | 34.7 | 52.1 | |
| | Shijiazhuang | 114.4942 | 38.0980 | δ$^{15}$N and δ$^{18}$O | 2017-2018 | Winter | 6.3 | 66.0 | 27.7 | Luo et al., 2023 |
| | Tianjin | 117.3350 | 38.9944 | δ$^{18}$O | 2019-2020 | Winter | 31.4 | 68.6 | | Zhang et al., 2022 |
| | | | | | | Spring | 57.4 | 42.6 | | |
| | | | | | | Summer | 84.1 | 15.9 | | |
| | | | | | | Autumn | 62.0 | 38.0 | | |
| | Jiaozuo | 113.2606 | 35.1865 | δ$^{18}$O | 2020 | Winter | 30.0 | 70.0 | | Li et al., 2022a |
| | | | | | | Summer | 61.0 | 39.0 | | |
| | Beijing | 116.3711 | 39.9744 | Δ$^{17}$O | 2016 | Winter | 42.6 | 25.4 | 31.7 | Fan et al., 2022 |
| | | | | 2017 | Summer | 46.2 | 23.0 | 31.4 | |
| | Beijing | 116.4167 | 40.0333 | δ$^{18}$O | 2017-2018 | Winter | 67.4 | 32.6 | | Zhang et al., 2021a |
| | Tianjin | 117.1500 | 39.0833 | | | | 58.1 | 41.9 | | |
| | Shijiazhuang | 114.6377 | 38.0128 | | | | 50.5 | 49.5 | | |

| Region | City | Longitude | Latitude | Isotope | Year | Season | | | | Reference |
|---|---|---|---|---|---|---|---|---|---|---|
| | Jinan | 117.0500 | 36.6667 | | | | 50.6 | 49.4 | | |
| | Jiaozuo | 113.2667 | 35.1833 | | | | 50.7 | 49.3 | | |
| | Beijing | 117.7000 | 40.0667 | $\delta^{15}N$ and $\delta^{18}O$ | 2017-2018 | Winter | 45.3 | 46.5 | 8.2 | Zhang et al., 2021b |
| | Shijiazhuang | 114.6377 | 38.0128 | $\delta^{18}O$ | 2017-2018 | Autumn | 51.8 | | 48.2 | Luo et al., 2021 |
| | | | | | | Winter | 22.5 | | 77.5 | |
| | | | | | | Spring | 51.5 | | 48.5 | |
| | | | | | | Summer | 71.1 | | 28.9 | |
| | Beijing | 116.3400 | 39.9300 | $\delta^{18}O$ | 2013-2014 | Spring | 18.8 | 81.2 | | Zong et al., 2020 |
| | | | | | | Summer | 41.2 | 58.8 | | |
| | | | | | | Autumn | 17.9 | 82.1 | | |
| | | | | | | Winter | 17.3 | 82.7 | | |
| | Beijing | 116.3667 | 39.9667 | $\delta^{18}O$ | 2018 | Winter | 52.0 | 48.0 | | Fan et al., 2020 |
| | Beijing | 116.4167 | 40.0333 | $\delta^{18}O$ | 2017-2018 | Winter | 48.0 | 52.0 | | Zhang et al., 2020a |
| | Beijing | 116.3800 | 39.9800 | $\delta^{18}O$ | 2013 | Spring | 56.5 | 27.2 | 16.3 | Luo et al., 2020a |
| | Jinan | 117.0500 | 36.6667 | $\delta^{18}O$ | 2018 | Summer | 39.1 | 60.9 | | Zhang et al., 2020b |
| | Beijing | 116.7000 | 40.0667 | $\Delta^{17}O$ | 2015 | Winter | 29.5 | 35.1 | 35.6 | Song et al., 2020 |
| | Beijing | 116.4167 | 40.0333 | $\delta^{18}O$ | 2017-2018 | Winter | 47.0 | 53.0 | | Zhang et al., 2020c |
| | Beijing | 116.7000 | 40.0667 | $\Delta^{17}O$ | 2014 | Spring | 27.0 | 35.0 | 38.0 | Wang et al., 2019 |
| | | | | | | Summer | 40.9 | 36.1 | 32.9 | |
| | | | | | | Autumn | 25.0 | 35.0 | 40.0 | |
| | | | | | | Winter | 31.0 | 34.0 | 34.0 | |
| | Beijing | 116.6800 | 40.4100 | $\Delta^{17}O$ | 2014-2015 | Winter | 23.5 | 76.5 | | He et al., 2018 |
| | Dongying | 118.9833 | 37.7500 | $\delta^{18}O$ | 2013 | Summer | 47.4 | 52.6 | | Zong et al., 2018 |
| Central | Wuhan | 114.3000 | 30.5300 | $\delta^{18}O$ | 2018-2019 | Spring | 35.0 | 65.0 | | Deng et al., 2024 |
| | | | | | | Summer | 69.7 | 30.3 | | |
| | | | | | | Autumn | 34.5 | 65.5 | | |

| Region | City | Longitude | Latitude | Method | Year | Season | | | | Reference |
|---|---|---|---|---|---|---|---|---|---|---|
| | | | | | | Winter | 7.8 | 92.2 | | |
| | Shiyan | 111.4000 | 33.2000 | $\delta^{18}O$ | 2021 | Spring | 64.0 | 19.0 | 17.0 | Xiao et al., 2024 |
| | | | | | | Summer | 84.0 | 8.0 | 8.0 | |
| | | | | | | Autumn | 64.0 | 19.0 | 18.0 | |
| | | | | | | Winter | 25.0 | 38.0 | 37.0 | |
| | Wuhan | 104.3600 | 30.5200 | $\delta^{18}O$ | 2013-2014 | Spring | 17.9 | 82.1 | | Zong et al., 2020 |
| | | | | | | Summer | 60.4 | 39.6 | | |
| | | | | | | Autumn | 28.5 | 71.5 | | |
| | | | | | | Winter | 11.2 | 88.8 | | |
| | Lanzhou | 103.8551 | 36.0305 | $\delta^{15}N$ and $\delta^{18}O$ | 2017-2018 | Winter | 22.2 | 57.6 | 20.4 | Luo et al., 2023 |
| | Shanghai | 121.4279 | 31.2595 | $\delta^{15}N$ and $\delta^{18}O$ | 2017-2018 | Winter | 18.3 | 47.9 | 33.8 | Luo et al., 2023 |
| | Shanghai | 121.5000 | 31.3000 | $\delta^{18}O$ | 2019 | Summer | 70.5 | 29.5 | | Huang et al., 2024 |
| | | | | | | Winter | 20.7 | 79.3 | | |
| | Shanghai | 121.5100 | 31.3400 | $\delta^{15}N$ and $\delta^{18}O$ | 2018 | Winter | 16.7 | 28.3 | 55.0 | Zhu et al., 2021 |
| | | | | | 2019 | Summer | 56.1 | 24.8 | 19.1 | |
| | Shanghai | 121.5700 | 31.2600 | $\Delta^{17}O$ | 2016 | Winter | 63.8 | | 36.2 | He et al., 2020 |
| | | | | | | Spring | 62.7 | | 37.3 | |
| YRD | Shanghai | 121.5000 | 31.2900 | $\delta^{18}O$ | 2013-2014 | Spring | 32.6 | 67.4 | | Zong et al., 2020 |
| | | | | | | Summer | 60.0 | 40.0 | | |
| | | | | | | Autumn | 27.7 | 72.3 | | |
| | | | | | | Winter | 6.2 | 93.8 | | |
| | Nanjing | 118.7000 | 32.2000 | $\Delta^{17}O$ | 2018 | Winter | 20.9 | 27.4 | 34.0 | Yu et al., 2023 |
| | Hangzhou | 120.1700 | 30.2300 | $\Delta^{17}O$ | 2015-2016 | Autumn | 18.9 | 34.0 | 47.1 | Fan et al., 2023 |
| | | | | | | Winter | 24.1 | 34.8 | 41.1 | |
| | | | | | | Spring | 18.3 | 35.0 | 46.7 | |
| | | | | | | Summer | 40.5 | 35.2 | 24.3 | |

| Region | City | | | | | | | | | Reference |
|---|---|---|---|---|---|---|---|---|---|---|
| Southeast | Xiamen | 118.0900 | 24.4360 | $\delta^{18}O$ | 2019-2021 | Cold | 20.2 | 38.2 | 21.6 | Li et al., 2022b |
| | | | | | | Warm | 30.3 | 31.5 | 18.3 | |
| | Nanchang | 115.8085 | 28.6832 | $\delta^{15}N$ and $\delta^{18}O$ | 2017-2018 | Winter | 17.2 | 43.6 | 39.2 | Luo et al., 2023 |
| | Nanchang | 115.9333 | 28.6833 | $\delta^{15}N$ and $\delta^{18}O$ | 2017-2018 | Winter | 30.4 | 36.8 | 32.8 | Zhang et al., 2021c |
| | Nanchang | 115.8085 | 28.6832 | $\delta^{18}O$ | 2017-2018 | Autumn | 18.0 | 38.6 | 43.4 | Luo et al., 2020b |
| | | | | | | Winter | 7.5 | 33.8 | 58.7 | |
| | | | | | | Spring | 33.2 | 34.5 | 32.3 | |
| | | | | | | Summer | 58.7 | 28.1 | 13.2 | |
| | Nanchang | 115.9000 | 28.7000 | $\delta^{15}N$ and $\delta^{18}O$ | 2017 | Autumn | 37.1 | 60.3 | 2.6 | Xiao et al., 2020 |
| | Ganzhou | 114.7600 | 25.6600 | $\delta^{18}O$ | 2019 | Winter | 41.6 | 29.5 | 28.9 | Cheng et al., 2022 |
| | | | | | | Summer | 73.5 | 14.1 | 12.4 | |
| Southwest | Chengdu | 104.0527 | 30.5598 | $\delta^{15}N$ and $\delta^{18}O$ | 2017-2018 | Winter | 23.5 | 40.8 | 35.7 | Luo et al., 2023 |
| | Chengdu | 104.3800 | 30.6400 | $\delta^{18}O$ | 2013-2014 | Spring | 46.5 | 53.5 | | Zong et al., 2020 |
| | | | | | | Summer | 61.8 | 38.2 | | |
| | | | | | | Autumn | 29.1 | 70.9 | | |
| | | | | | | Winter | 11.3 | 88.7 | | |
| | Kunming | 102.7000 | 25.0667 | $\delta^{18}O$ | 2017-2018 | Autumn | 85.0 | | 15.0 | Guo et al., 2021 |
| | | | | | | Winter | 74.4 | | 25.6 | |
| | Nanning | 108.2833 | 22.8333 | | | Autumn | 87.9 | | 12.1 | |
| | | | | | | Winter | 68.4 | | 31.6 | |
| | Guiyang | 106.7167 | 26.5667 | $\delta^{18}O$ | 2013-2014 2016 | Winter | 38.3 | 61.7 | | Li et al., 2021a |
| | | | | | | Summer | 80.2 | 19.8 | | |
| | Nanning | 108.3192 | 22.7990 | $\delta^{15}N$ and $\delta^{18}O$ | 2017-2018 | Winter | 47.5 | 27.8 | 24.7 | Luo et al., 2023 |
| PRD | Guangzhou | 113.3689 | 23.1938 | $\delta^{18}O$ | 2015-2018 | Spring | 70.4 | 22.8 | 6.8 | Xi et al., 2023 |
| | | | | | | Summer | 85.6 | 10.9 | 3.5 | |
| | | | | | | Autumn | 86.1 | 10.6 | 3.3 | |

| Region | City | Longitude | Latitude | Measure | Year | Season | | | | Reference |
|---|---|---|---|---|---|---|---|---|---|---|
| | | | | | | Winter | 72.0 | 21.0 | 7.1 | |
| | Guangzhou | 113.3397 | 23.1075 | $\Delta^{17}O$ | 2018 | Autumn | 61.0 | 12.0 | 27.0 | Wang et al., 2023 |
| | Guangzhou | 113.3600 | 23.1500 | $\delta^{18}O$ | 2013-2014 | Spring | 48.0 | 52.0 | | Zong et al., 2020 |
| | | | | | | Summer | 66.9 | 33.1 | | |
| | | | | | | Autumn | 33.9 | 66.1 | | |
| | | | | | | Winter | 9.7 | 90.3 | | |
| Northeast | Harbin | 126.6333 | 45.7469 | $\Delta^{17}O$ | 2022-2023 | Winter | 55.3 | 28.2 | 16.5 | Feng et al., 2023 |
| | Shenyang | 123.4300 | 41.7700 | | | Spring | 65.8 | 25.6 | 8.6 | Li et al., 2022c |
| | | | | | | Summer | 95.3 | 3.0 | 1.7 | |
| | | | | | | Autumn | 83.0 | 12.1 | 4.8 | |
| | | | | $\Delta^{17}O$ | 2015-2018 | Winter | 57.7 | 30.9 | 11.4 | |
| | Fushun | 124.9400 | 41.8500 | | | Spring | 40.0 | 34.5 | 25.5 | |
| | | | | | | Summer | 85.7 | 9.6 | 4.7 | |
| | | | | | | Autumn | 46.0 | 35.9 | 18.1 | |
| | | | | | | Winter | 37.6 | 33.2 | 29.2 | |
| | Harbin | 126.7400 | 45.7300 | | | | 69.4 | 30.6 | | Zhao et al., 2021 |
| | Changchun | 125.4000 | 43.8500 | $\delta^{18}O$ | 2017-2018 | Winter | 50.2 | 49.8 | | |
| | Harbin | 126.5300 | 45.8400 | | | | 50.1 | 49.9 | | |
| | Yushu | 126.5300 | 44.8600 | | | | 37.3 | 62.7 | | |
| | Harbin | 120.6817 | 45.7539 | $\delta^{18}O$ | 2017-2018 | Spring | 46.4 | 53.6 | | Sun et al., 2020 |
| | | | | | | Summer | 68.7 | 31.2 | | |
| | | | | | | Autumn | 56.1 | 43.9 | | |
| | | | | | | Winter | 44.8 | 55.2 | | |
| | Changchun | 125.4000 | 44.0000 | $\delta^{18}O$ | 2017-2018 | Summer | 79.7 | 20.3 | | Zhao et al., 2020 |
| | | | | | | Autumn | 56.1 | 43.9 | | |
| | | | | | | Winter | 55.9 | 44.1 | | |

| Type | Location | Longitude | Latitude | Method | Year | Season | | | | Reference |
|---|---|---|---|---|---|---|---|---|---|---|
| | | | | | | Spring | 58.0 | 42.0 | | |
| Island | Beihuangcheng Island | 120.9167 | 36.4000 | $\delta^{18}O$ | 2015 | Autumn | 35.0 | 65.0 | | Zong et al., 2017 |
| | | | | | | Winter | 24.0 | 76.0 | | |
| | | | | | | Spring | 47.0 | 53.0 | | |
| | | | | | | Summer | 68.0 | 32.0 | | |
| | Dongsha Island | 116.7167 | 20.7000 | $\delta^{18}O$ | 2013 | Spring | 66.8 | 10.2 | 23.0 | Yang et al., 2023 |
| | Pengjiayu Island | 122.1333 | 26.0500 | | | | 62.1 | 23.4 | 14.5 | |
| Marine | Bohai Sea and Yellow Sea | | 32 – 34 | $\Delta^{17}O$ | 2018 | Spring | 60.0 | 24.0 | 16.0 | Zhao et al., 2024 |
| | | | 34 – 36 | | | | 49.0 | 31.0 | 19.0 | |
| | | | 36 – 38 | | | | 26.0 | 35.0 | 39.0 | |
| | | | 38 – 40 | | | | 14.0 | 34.0 | 52.0 | |
| | Bohai Sea | | 38 – 40 | $\delta^{18}O$ | 2014 | Summer | 65.9 | 34.1 | | Zong et al., 2022 |
| | | | | | | Winter | 43.9 | 56.1 | | |
| | | | | | 2016 | Summer | 49.3 | 50.7 | | |
| | | | | | | Winter | 41.1 | 58.9 | | |
| | | | | | 2017 | Summer | 73.6 | 26.4 | | |
| | | | | | | Winter | 26.4 | 73.6 | | |
| | | | | | 2018 | Summer | 61.0 | 39.0 | | |
| | | | | | | Winter | 32.0 | 68.0 | | |
| | | | | | 2019 | Summer | 61.7 | 38.3 | | |
| | | | | | | Winter | 28.5 | 71.5 | | |
| NCP | Beijing | 116.4125 | 40.2352 | CMAQ | 2013 | Winter | 77.8 | 14.3 | | Liu et al., under review |
| | | | | | 2018 | | 85.6 | 9.9 | | |
| | Tianjin | 117.4812 | 39.0929 | | 2013 | | 67.8 | 19.7 | | |
| | | | | | 2018 | | 77.5 | 15.1 | | |
| | Shijiazhuang | 114.5562 | 37.8247 | | 2013 | | 73.8 | 14.2 | | |

| Region | City | Lon | Lat | Model | Year | Season | | | Reference |
|---|---|---|---|---|---|---|---|---|---|
| | | | | | 2018 | | 83.6 | 8.3 | |
| | Jinan | 116.9187 | 36.8712 | | 2013 | | 63.7 | 27.7 | |
| | | | | | 2018 | | 72.9 | 19.9 | |
| | Zhengzhou | 113.8719 | 34.7968 | | 2013 | | 77.6 | 10.6 | |
| | | | | | 2018 | | 79.2 | 12.7 | |
| | Qingdao | 120.5000 | 36.1533 | | 2013 | | 51.5 | 41.5 | |
| | | | | | 2018 | | 56.5 | 37.0 | |
| | Yantai | 121.4187 | 37.4197 | | 2013 | | 48.2 | 45.7 | |
| | | | | | 2018 | | 50.4 | 43.5 | |
| NCP | | 115.4817 | 36.6111 | WRF-Chem | 2021 | Winter | 51.0 | 45.0 | Yang et al., 2024 |
| NCP | | 114.8000 | 39.5000 | WRF-Chem | 2017 | Winter | 48.0 | 30.3 | Zhao et al., 2023 |
| YRD | Shanghai | 120.8662 | 28.0818 | CMAQ | 2017 | Winter | 69.3 | 28.4 | Sun et al., 2022 |
| | | | | | | Spring | 81.8 | 15.3 | |
| | | | | | | Summer | 82.9 | 12.2 | |
| | | | | | | Autumn | 86.9 | 11.1 | |
| | Nanjing | 118.6491 | 28.7744 | | | Winter | 59.2 | 36.1 | |
| | | | | | | Spring | 73.1 | 5.4 | |
| | | | | | | Summer | 74.7 | 17.9 | |
| | | | | | | Autumn | 69.7 | 25.4 | |
| | Hefei | 117.3881 | 28.5389 | | | Winter | 66.9 | 27.1 | |
| | | | | | | Spring | 78.5 | 16.5 | |
| | | | | | | Summer | 81.7 | 10.4 | |
| | | | | | | Autumn | 72.5 | 21.8 | |
| | Changzhou | 119.6087 | 28.4776 | | | Winter | 68.9 | 26.8 | |
| | | | | | | Spring | 74.9 | 20.9 | |
| | | | | | | Summer | 78.7 | 14.3 | |

| Region | City | Longitude | Latitude | Model | Year | Season | | | Reference |
|---|---|---|---|---|---|---|---|---|---|
| | | | | | | Autumn | 77.6 | 18.3 | |
| | | | | | | Winter | 59.7 | 35.5 | |
| | Hangzhou | 119.7671 | 27.1448 | | | Spring | 70.5 | 23.3 | |
| | | | | | | Summer | 76.4 | 10.7 | |
| | | | | | | Autumn | 73.8 | 21.3 | |
| YRD | Shanghai | 120.9890 | 31.0970 | F0AM | 2019 | Winter | 42.9 | 55.5 | Zang et al., 2022 |
| | | 121.5330 | 31.2280 | | | | 36.8 | 62.1 | |
| NCP | | 114.8000 | 39.5000 | WRF-Chem | 2016 | Summer | 60.4 | 39.6 | Li et al., 2021b |
| | | | | | | Winter | 91.9 | 8.1 | |
| YRD | | 121.0000 | 31.1000 | | | Summer | 75.3 | 24.7 | |
| | | | | | | Winter | 86.2 | 13.8 | |
| NCP | | 114.8000 | 39.5000 | WRF-Chem | 2016 | Winter | 53.4 | 46.6 | Liu et al., 2020 |
| Taiwan | | 120.6753 | 21.1892 | WRF-CMAQ PA | 2017 | Spring | > 45.0 | 30.0 | Chuang et al., 2022 |
| PRD | | 113.3600 | 23.1500 | WRF-CMAQ PA | 2015 | Winter | 47.0 | 34.0 | Qu et al., 2021 |
| Central | Xi'an | 108.9552 | 34.2919 | WRF-Chem | 2017 | Winter | | 24.0 | Wu et al., 2021 |
| NCP | Beijing | 116.6800 | 40.4100 | GEOS-Chem | 2014-2015 | Winter | 34.4 | 44.9 | Chan et al., 2021 |
| NCP | | 114.8000 | 39.5000 | WRF-CMAQ PA | 2017 | Winter | 43.0 | 44.0 | Fu et al., 2020 |
| NCP | Beijing | 116.6800 | 40.4100 | Box model | 2016-2017 | Winter | 68.8 | 31.2 | Chen et al., 2020 |
| YRD | Nanjing | 118.6491 | 28.7744 | Box model | 2015 | Winter | | 80 | Sun et al., 2018 |
| NCP | Beijing | 116.6800 | 40.4100 | CMAQ | 2017 | Winter | | 42 | Qiu et al., 2019 |
| NCP | | 114.8000 | 39.5000 | WRF-Chem | 2015 | Winter | | 30.1 | Liu et al., 2019 |
| PRD | | 113.3600 | 23.1500 | WRF-Chem | 2013 | Winter | | 57.4 | Li et al., 2016 |

Chan, Y.C., Evans, M.J., He, P., Holmes, C.D., Jaeglé, L., Kasibhatla, P., Liu, X.Y., Sherwen, T., Thornton, J.A., Wang, X., Xie, Z., Zhai, S., Alexander, B., 2021. Heterogeneous nitrate production mechanisms in intense haze events in the North China Plain. Journal of geophysical research. Atmospheres 126, e2021JD034688. https://doi.org/10.1029/2021JD034688.

Cheng, C., Yu, R., Chen, Y., Yan, Y., Hu, G., Wang, S., 2022. Quantifying the source and formation of nitrate in $PM_{2.5}$ using dual isotopes combined with Bayesian

mixing model: A case study in an inland city of southeast China. Chemosphere 308, 136097. https://doi.org/10.1016/j.chemosphere.2022.136097.

Chuang, M., Wu, C., Lin, C., Lin, W., Chou, C.C.K., Lee, C., Lin, T., Fu, J.S., Kong, S.S., 2022. Simulating nitrate formation mechanisms during $PM_{2.5}$ events in Taiwan and their implications for the controlling direction. Atmos Environ 269, 118856. https://doi.org/10.1016/j.atmosenv.2021.118856.

Deng, M., Wang, C., Yang, C., Li, X., Cheng, H., 2024. Nitrogen and oxygen isotope characteristics, formation mechanism, and source apportionment of nitrate aerosols in Wuhan, central China. Sci Total Environ 921, 170715. https://doi.org/10.1016/j.scitotenv.2024.170715.

Fan, M., Zhang, Y., Lin, Y., Hong, Y., Zhao, Z., Xie, F., Du, W., Cao, F., Sun, Y., Fu, P., 2022. Important role of $NO_3$ radical to nitrate formation aloft in urban Beijing: Insights from triple oxygen isotopes measured at the tower. Environ Sci Technol 56 (11), 6870-6879. https://doi.org/10.1021/acs.est.1c02843.

Fan, M.Y., Zhang, W., Zhang, Y.L., Li, J., Fang, H., Cao, F., Yan, M., Hong, Y., Guo, H., Michalski, G., 2023. Formation mechanisms and source apportionments of nitrate aerosols in a megacity of eastern China based on multiple isotope observations. Journal of Geophysical Research: Atmospheres 128, e2022JD038129. https://doi.org/10.1029/2022JD038129.

Fan, M.Y., Zhang, Y.L., Lin, Y.C., Cao, F., Zhao, Z.Y., Sun, Y., Qiu, Y., Fu, P., Wang, Y., 2020. Changes of emission sources to nitrate aerosols in Beijing after the clean air actions: Evidence from dual isotope compositions. Journal of Geophysical Research: Atmospheres 125, e2019JD031998. https://doi.org/10.1029/2019JD031998.

Feng, X., Chen, Y., Chen, S., Peng, Y., Liu, Z., Jiang, M., Feng, Y., Wang, L., Li, L., Chen, J., 2023. Dominant contribution of $NO_3$ radical to $NO_3^-$ formation during heavy haze episodes: Insights from high-time resolution of dual isotopes $\delta^{17}O$ and $\delta^{18}O$. Environ Sci Technol 57 (49), 20726-20735. https://doi.org/10.1021/acs.est.3c07590.

Fu, X., Wang, T., Gao, J., Wang, P., Liu, Y., Wang, S., Zhao, B., Xue, L., 2020. Persistent heavy winter nitrate pollution driven by increased photochemical oxidants in northern China. Environ Sci Technol 54 (7), 3881-3889. https://doi.org/10.1021/acs.est.9b07248.

Guo, W., Luo, L., Zhang, Z., Zheng, N., Xiao, H., Xiao, H., 2021. The use of stable oxygen and nitrogen isotopic signatures to reveal variations in the nitrate formation pathways and sources in different seasons and regions in China. Environ Res 201, 111537. https://doi.org/10.1016/j.envres.2021.111537.

He, P., Xie, Z., Chi, X., Yu, X., Fan, S., Kang, H., Liu, C., Zhan, H., 2018. Atmospheric $\Delta^{17}O(NO_3^-)$ reveals nocturnal chemistry dominates nitrate production in Beijing haze. Atmos Chem Phys 18 (19), 14465-14476. https://doi.org/10.5194/acp-18-14465-2018.

He, P., Xie, Z., Yu, X., Wang, L., Kang, H., Yue, F., 2020. The observation of isotopic compositions of atmospheric nitrate in Shanghai China and its implication for reactive nitrogen chemistry. Sci Total Environ 714, 136727. https://doi.org/10.1016/j.scitotenv.2020.136727.

Huang, W., Ye, X., Lv, Z., Yao, Y., Chen, Y., Zhou, Y., Chen, J., 2024. Dual isotopic evidence of $\delta^{15}N$ and $\delta^{18}O$ for priority control of vehicle emissions in a megacity of east China: Insight from measurements in summer and winter. Sci Total Environ 931, 172918. https://doi.org/10.1016/j.scitotenv.2024.172918.

Li, M., Zhang, Z., Yao, Q., Wang, T., Xie, M., Li, S., Zhuang, B., Han, Y., 2021b. Nonlinear responses of particulate nitrate to NOx emission controls in the megalopolises of China. Atmos Chem Phys 21 (19), 15135-15152. https://doi.org/10.5194/acp-21-15135-2021.

Li, Q., Li, X., Yang, Z., Cui, G., Ding, S., 2021a. Diurnal and seasonal variations in water-soluble inorganic ions and nitrate dual isotopes of $PM_{2.5}$: Implications for source apportionment and formation processes of urban aerosol nitrate. Atmos Res 248, 105197. https://doi.org/10.1016/j.atmosres.2020.105197.

Li, Q., Zhang, L., Wang, T., Tham, Y.J., Ahmadov, R., Xue, L., Zhang, Q., Zheng, J., 2016. Impacts of heterogeneous uptake of dinitrogen pentoxide and chlorine activation on ozone and reactive nitrogen partitioning: improvement and application of the WRF-Chem model in southern China. Atmos Chem Phys 16, 14875-14890. https://doi.org/10.5194/acp-16-14875-2016.

Li, X., Wu, S., Zhang, J., Schwab, J.J., 2022b. Insights into factors affecting size-segregated nitrate formation in a coastal city through measurements of dual isotopes. Atmos Environ 290, 119385. https://doi.org/10.1016/j.atmosenv.2022.119385.

Li, Y., Geng, Y., Hu, X., Yin, X., 2022a. Seasonal differences in sources and formation processes of $PM_{2.5}$ nitrate in an urban environment of North China. J Environ Sci-China 120, 94-104. https://doi.org/10.1016/j.jes.2021.08.020.

Li, Z., Walters, W.W., Hastings, M.G., Song, L., Huang, S., Zhu, F., Liu, D., Shi, G., Li, Y., Fang, Y., 2022c. Atmospheric nitrate formation pathways in urban and rural atmosphere of Northeast China: implications for complicated anthropogenic effects. Environ Pollut 296, 118752. https://doi.org/10.1016/j.envpol.2021.118752.

Liu, L., Bei, N., Hu, B., Wu, J., Liu, S., Li, X., Wang, R., Liu, Z., Shen, Z., Li, G., 2020. Wintertime nitrate formation pathways in the North China Plain: importance of $N_2O_5$ heterogeneous hydrolysis. Environ Pollut 266, 115287. https://doi.org/10.1016/j.envpol.2020.115287.

Liu, L., Wu, J., Liu, S., Li, X., Zhou, J., Feng, T., Qian, Y., Gao, J., Tie, X., Li, G., 2019. Effects of organic coating on the nitrate formation by suppressing the N2O5 heterogeneous hydrolysis: a case study during wintertime in Beijing–Tianjin–Hebei (BTH). Atmos Chem Phys 19, 8189-8207. https://doi.org/10.5194/acp-19-8189-2019.

Liu, Z., Liu, X., Ni, Y., Qi, J. Exploring Atmospheric Nitrate Formation Mechanisms during the Winters of 2013 and 2018 in the North China Region using Modeling and Isotopic Analysis. Atmos Chem Phys, under review.

Luo, L., Kao, S., Wu, Y., Zhang, X., Lin, H., Zhang, R., Xiao, H., 2020. Stable oxygen isotope constraints on nitrate formation in Beijing in springtime. Environ Pollut 263, 114515. https://doi.org/10.1016/j.envpol.2020.114515.

Luo, L., Liao, T., Zhang, X., Wu, Y., Li, J., Zhang, R., Zheng, Z., Kao, S., 2023. Quantifying the formation pathways of nitrate in size-segregated aerosols during winter haze pollution. Gondwana Res 115, 71-80. https://doi.org/10.1016/j.gr.2022.11.015.

Luo, L., Pan, Y., Zhu, R., Zhang, Z., Zheng, N., Liu, Y., Liu, C., Xiao, H., Xiao, H., 2020b. Assessment of the seasonal cycle of nitrate in $PM_{2.5}$ using chemical compositions and stable nitrogen and oxygen isotopes at Nanchang, China. Atmos Environ 225, 117371. https://doi.org/10.1016/j.atmosenv.2020.117371.

Luo, L., Zhu, R., Song, C., Peng, J., Guo, W., Liu, Y., Zheng, N., Xiao, H., Xiao, H., 2021. Changes in nitrate accumulation mechanisms as $PM_{2.5}$ levels increase on the North China Plain: a perspective from the dual isotopic compositions of nitrate. Chemosphere 263, 127915. https://doi.org/10.1016/j.chemosphere.2020.127915.

Moore, J.W., Semmens, B.X., 2008. Incorporating uncertainty and prior information into stable isotope mixing models. Ecol Lett 11 (5), 470-480. https://doi.org/10.1111/j.1461-0248.2008.01163.x.

Qiu, X., Ying, Q., Wang, S., Duan, L., Zhao, J., Xing, J., Ding, D., Sun, Y., Liu, B., Shi, A., Yan, X., Xu, Q., Hao, J., 2019. Modeling the impact of heterogeneous reactions of chlorine on summertime nitrate formation in Beijing, China. Atmos Chem Phys 19 (10), 6737-6747. https://doi.org/10.5194/acp-19-6737-2019.

Qu, K., Wang, X., Xiao, T., Shen, J., Lin, T., Chen, D., He, L., Huang, X., Zeng, L., Lu, K., Ou, Y., Zhang, Y., 2021. Cross-regional transport of $PM_{2.5}$ nitrate in the Pearl River Delta, China: Contributions and mechanisms. Sci Total Environ 753, 142439. https://doi.org/10.1016/j.scitotenv.2020.142439.

Song, W., Liu, X., Wang, Y., Tong, Y., Bai, Z., Liu, C., 2020. Nitrogen isotope differences between atmospheric nitrate and corresponding nitrogen oxides: a new constraint using oxygen isotopes. Sci Total Environ 701, 134515. https://doi.org/10.1016/j.scitotenv.2019.134515.

Sun, J., Qin, M., Xie, X., Fu, W., Qin, Y., Sheng, L., Li, L., Li, J., Sulaymon, I.D., Jiang, L., Huang, L., Yu, X., Hu, J., 2022. Seasonal modeling analysis of nitrate formation pathways in Yangtze River Delta region, China. Atmos Chem Phys 22 (18), 12629-12646. https://doi.org/10.5194/acp-22-12629-2022.

Sun, X., Zong, Z., Wang, K., Li, B., Fu, D., Shi, X., Tang, B., Lu, L., Thapa, S., Qi, H., Tian, C., 2020. The importance of coal combustion and heterogeneous reaction for atmospheric nitrate pollution in a cold metropolis in China: insights from isotope fractionation and Bayesian mixing model. Atmos Environ 243, 117730. https://doi.org/10.1016/j.atmosenv.2020.117730.

Wang, Y., Liu, J., Jiang, F., Chen, Z., Wu, L., Zhou, S., Pei, C., Kuang, Y., Cao, F., Zhang, Y., Fan, M., Zheng, J., Li, J., Zhang, G., 2023. Vertical measurements of stable nitrogen and oxygen isotope composition of fine particulate nitrate aerosol in Guangzhou city: source apportionment and oxidation pathway. Sci Total Environ 865, 161239. https://doi.org/10.1016/j.scitotenv.2022.161239.

Wang, Y.L., Song, W., Yang, W., Sun, X.C., Tong, Y.D., Wang, X.M., Liu, C.Q., Bai, Z.P., Liu, X.Y., 2019. Influences of atmospheric pollution on the contributions of major oxidation pathways to $PM_{2.5}$ nitrate formation in Beijing. Journal of Geophysical Research: Atmospheres 124, 4174-4185. https://doi.org/10.1029/2019JD030284.

Wu, C., Liu, L., Wang, G., Zhang, S., Li, G., Lv, S., Li, J., Wang, F., Meng, J., Zeng, Y., 2021. Important contribution of N2O5 hydrolysis to the daytime nitrate in Xi'an, China during haze periods: Isotopic analysis and WRF-Chem model simulation. Environ Pollut 288, 117712. https://doi.org/10.1016/j.envpol.2021.117712.

Xi, D., Xiao, Y., Mgelwa, A.S., Kuang, Y., 2023. Formation pathways and source apportionments of inorganic nitrogen-containing aerosols in urban environment: insights from nitrogen and oxygen isotopic compositions in Guangzhou, China. Atmos Environ 309, 119888. https://doi.org/10.1016/j.atmosenv.2023.119888.

Xiao, C., Sun, Y., Zhao, T., Wang, G., Li, P., Zhao, Y., Chen, F., 2024. Assessment major NOx sources to nitrate of tsp around the Danjiangkou reservoir using isotopes and a Bayesian isotope mixing model. Atmos Pollut Res 15 (7). https://doi.org/10.1016/j.apr.2024.102151.

Xiao, H.W., Zhu, R.G., Pan, Y.Y., Guo, W., Zheng, N.J., Liu, Y.H., Liu, C., Zhang, Z.Y., Wu, J.F., Kang, C.A., Luo, L., Xiao, H.Y., 2020. Differentiation between nitrate aerosol formation pathways in a southeast Chinese city by dual isotope and modeling studies. Journal of Geophysical Research: Atmospheres 125, e2020JD032604. https://doi.org/10.1029/2020JD032604.

Yan, X., Hu, B., Li, Y., Shi, G., 2023. Investigating atmospheric nitrate sources and formation pathways between heating and non-heating seasons in urban North China. Environ Res Lett 18 (3), 34006. https://doi.org/10.1088/1748-9326/acb805.

Yang, J., Qu, Y., Chen, Y., Zhang, J., Liu, X., Niu, H., An, J., 2024. Dominant physical and chemical processes impacting nitrate in Shandong of the North China Plain during winter haze events. Sci Total Environ 912, 169065. https://doi.org/10.1016/j.scitotenv.2023.169065.

Yang, S., Luo, L., Li, Y., Wang, C., Lu, B., Hsu, S., Kao, S., 2023. Dry deposition fluxes, formation mechanisms and sources of nitrate in total suspended particles in springtime on Dongsha Island, South China Sea. Journal of Earth Environment 14 (2), 193-206. https://doi.org/10.7515/jee222049.

Yu, H., Zhang, Y., Cao, F., Zhao, Z., Fan, M., Yang, X., 2023. Fog event is possibly a source rather than a sink of atmospheric nitrate aerosols: Insights from isotopic measurements in Nanjing, China. Appl Geochem 155, 105721. https://doi.org/10.1016/j.apgeochem.2023.105721.

Zang, H., Zhao, Y., Huo, J., Zhao, Q., Fu, Q., Duan, Y., Shao, J., Huang, C., An, J., Xue, L., Li, Z., Li, C., Xiao, H., 2022. High atmospheric oxidation capacity drives wintertime nitrate pollution in the eastern Yangtze River Delta of China. Atmos Chem Phys 22 (7), 4355-4374. https://doi.org/10.5194/acp-22-4355-2022.

Zhang, W., Bi, X., Zhang, Y., Wu, J., Feng, Y., 2022. Diesel vehicle emission accounts for the dominate NOx source to atmospheric particulate nitrate in a coastal city: Insights from nitrate dual isotopes of $PM_{2.5}$. Atmos Res 278, 106328. https://doi.org/10.1016/j.atmosres.2022.106328.

Zhang, W., Wu, F., Luo, X., Song, L., Wang, X., Zhang, Y., Wu, J., Xiao, Z., Cao, F., Bi, X., Feng, Y., 2024. Quantification of NOx sources contribution to ambient nitrate aerosol, uncertainty analysis and sensitivity analysis in a megacity. Sci Total Environ 926, 171583. https://doi.org/10.1016/j.scitotenv.2024.171583.

Zhang, Z., Cao, L., Liang, Y., Guo, W., Guan, H., Zheng, N., 2021c. Importance of $NO_3$ radical in particulate nitrate formation in a southeast Chinese urban city: new constraints by $\delta^{15}N$-$\delta^{18}O$ space of $NO_3^-$. Atmos Environ 253, 118387. https://doi.org/10.1016/j.atmosenv.2021.118387.

Zhang, Z., Guan, H., Luo, L., Zheng, N., Xiao, H., 2020c. Response of fine aerosol nitrate chemistry to clean air action in winter Beijing: insights from the oxygen isotope signatures. Sci Total Environ 746, 141210. https://doi.org/10.1016/j.scitotenv.2020.141210.

Zhang, Z., Guan, H., Luo, L., Zheng, N., Xiao, H., Liang, Y., Xiao, H., 2020a. Sources and transformation of nitrate aerosol in winter 2017–2018 of megacity Beijing: Insights from an alternative approach. Atmos Environ 241, 117842. https://doi.org/10.1016/j.atmosenv.2020.117842.

Zhang, Z., Guan, H., Xiao, H., Liang, Y., Zheng, N., Luo, L., Liu, C., Fang, X., Xiao, H., 2021b. Oxidation and sources of atmospheric NOx during winter in Beijing

based on $\delta^{18}O$-$\delta^{15}N$ space of particulate nitrate. Environ Pollut 276, 116708. https://doi.org/10.1016/j.envpol.2021.116708.

Zhang, Z., Jiang, Z., Guan, H., Liang, Y., Zheng, N., Guo, W., 2021a. Isotopic evidence for the high contribution of wintertime photochemistry to particulate nitrate formation in northern China. Journal of Geophysical Research: Atmospheres 126, e2021JD035324. https://doi.org/10.1029/2021JD035324.

Zhang, Z., Zheng, N., Liang, Y., Luo, L., Xiao, H., Xiao, H., 2020b. Dominance of heterogeneous chemistry in summertime nitrate accumulation: insights from oxygen isotope of nitrate ($\delta^{18}O$–$NO_3^-$). Acs Earth Space Chem 4 (6), 818-824. https://doi.org/10.1021/acsearthspacechem.0c00101.

Zhao, X., Zhao, X., Liu, P., Chen, D., Zhang, C., Xue, C., Liu, J., Xu, J., Mu, Y., 2023. Transport pathways of nitrate formed from nocturnal $N_2O_5$ hydrolysis aloft to the ground level in winter North China Plain. Environ Sci Technol 57 (7), 2715-2725. https://doi.org/10.1021/acs.est.3c00086.

Zhao, Z., Cao, F., Fan, M., Zhai, X., Yu, H., Hong, Y., Ma, Y., Zhang, Y., 2021. Nitrate aerosol formation and source assessment in winter at different regions in Northeast China. Atmos Environ 267, 118767. https://doi.org/10.1016/j.atmosenv.2021.118767.

Zhao, Z., Cao, F., Fan, M., Zhang, W., Zhai, X., Wang, Q., Zhang, Y., 2020. Coal and biomass burning as major emissions of NOx in Northeast China: implication from dual isotopes analysis of fine nitrate aerosols. Atmos Environ 242, 117762. https://doi.org/10.1016/j.atmosenv.2020.117762.

Zhao, Z.Y., Zhang, Y.L., Lin, Y.C., Song, W.H., Yu, H.R., Fan, M.Y., Hong, Y.H., Yang, X.Y., Li, H.Y., Cao, F., 2024. Continental emissions influence the sources and formation mechanisms of marine nitrate aerosols in spring over the Bohai Sea and Yellow Sea inferred from stable isotopes. Journal of Geophysical Research: Atmospheres 129, e2023JD040541. https://doi.org/10.1029/2023JD040541.

Zhu, Y., Zhou, S., Li, H., Luo, L., Wang, F., Bao, Y., Chen, Y., 2021. Formation pathways and sources of size-segregated nitrate aerosols in a megacity identified by dual isotopes. Atmos Environ 264, 118708. https://doi.org/10.1016/j.atmosenv.2021.118708.

Zong, Z., Tan, Y., Wang, X., Tian, C., Fang, Y., Chen, Y., Fang, Y., Han, G., Li, J., Zhang, G., 2018. Assessment and quantification of NOx sources at a regional background site in North China: comparative results from a Bayesian isotopic mixing model and a positive matrix factorization model. Environ Pollut 242, 1379-1386. https://doi.org/10.1016/j.envpol.2018.08.026.

Zong, Z., Tan, Y., Wang, X., Tian, C., Li, J., Fang, Y., Chen, Y., Cui, S., Zhang, G., 2020. Dual-modelling-based source apportionment of NOx in five Chinese megacities: providing the isotopic footprint from 2013 to 2014. Environ Int 137, 105592. https://doi.org/10.1016/j.envint.2020.105592.

Zong, Z., Tian, C., Sun, Z., Tan, Y., Shi, Y., Liu, X., Li, J., Fang, Y., Chen, Y., Ma, Y., Gao, H., Zhang, G., Wang, T., 2022. Long‐term evolution of particulate nitrate pollution in North China: isotopic evidence from 10 offshore cruises in the Bohai Sea from 2014 to 2019. Journal of Geophysical Research: Atmospheres 127, e2022JD036567. https://doi.org/10.1029/2022JD036567.

Zong, Z., Wang, X., Tian, C., Chen, Y., Fang, Y., Zhang, F., Li, C., Sun, J., Li, J., Zhang, G., 2017. First assessment of NOx sources at a regional background site in North China using isotopic analysis linked with modeling. Environ Sci Technol 51 (11), 5923-5931. https://doi.org/10.1021/acs.est.6b06316

**Technical points:**

**Reply:** Thank you for the suggestion. We have replaced "absorbed" with "adsorbed" in line 47.

**Reply:** Thank you for the suggestion. We have restructured the paragraph to improve the readability by splitting it into two separate sections, which can now be found in lines 59–80 and lines 105–124 of the revised manuscript.

**Reply:** Thank you for your suggestion. We have revised the text by replacing "i.e." with a colon (:) to improve clarity, as recommended. This change has been made in line 176.

**Reply:** You are correct. This method employs the Pseudomonas aureofaciens (ATCC13985) strain to completely reduce $NO_3^-$ to $N_2O$ gas for isotope analysis. We have revised the text accordingly, and the change can be found in lines 206–209.
Revised sentence: The $\delta^{18}O$ and $\delta^{15}N$ values of $NO_3^-$ in the TSP samples were determined via the bacterial denitrifier method (Casciotti et al., 2002; Sigman et al., 2001) with the Gasbench-IRMS system (Delta V model, Thermo Scientific).

**Reply:** Thank you for your suggestion. We have revised the figure caption to clarify the use of the star symbol. The figure now states that the * indicates R values significant at the $p < 0.05$ level. This change can be found in the caption of Figure 3 in the revised manuscript.

**Reply:** Thank you for your suggestion. We have updated Figure 4 by adding larger labels on the left side to clearly indicate $PM_{2.5}$, $NO_3^-$, $NH_4^+$, and $SO_4^{2-}$ for each row of data. This revision can be found in Figure 4 of the revised manuscript.

[Figure]

Figure 4 Spatial distribution and changes in PM2.5 and its components in the NCP during the winters of 2013 and 2018. The up arrows indicate increases, and the down arrows indicate decreases.

Fig 5: Period missing at end of caption. This diverging color scheme is also a bit confusing as used here, because it is the same color scheme used in Fig 4 to show representative change (pos = red, neg = green), but here it is a unidirectional scale. I'd recommend a different color scheme to avoid confusion or unintentional misleading.

**Reply:** We are sorry for the confusion. We have revised Figure 5 by updating the color scheme to avoid confusion with Figure 4 and to better align it with its unidirectional scale. Additionally, we have added a period at the end of the title. This revision can be found in Figure 5 of the revised manuscript.

[Figure]

Figure 5 Spatial distributions of the nitrate and sulfate proportions in PM$_{2.5}$ in the NCP region during the winters of 2013 and 2018.

Figure 6: Humans are pretty bad at estimating angular areas. You might consider alternatives such as treemaps or waffle charts. Not required from me, but just put here for consideration. This is also a pretty simple figure, and since you have so many figures, you might consider merging it with another or whether it is necessary.

**Reply:** Thank you for your valuable suggestion. We believe that alternative visualizations, such as treemaps or waffle charts, can indeed improve the clarity and readability of the figure. After further review, we find that Figure 6 contributes relatively little to the overall manuscript; therefore, we have removed it.

Fig 9: The color choices could be changed to improve the visual story. For example, the OH +NO$_2$ on both sides would ideally both be blue, or shades of blue. And hetN$_2$O$_5$ both be orange or shades of orange. That would make it more clear that we should be directly relating them.

**Reply:** Thank you for your insightful suggestion. We have revised the figure according to suggestion. This revision can be found in Figure 9 of the revised manuscript.

[Figure]

Figure 9 Comparison of the contributions of the atmospheric NO$_3^-$ formation pathways based on the dual-isotope results and model simulations for Beijing in 2013 and 2018 and for Qingdao in 2018.

Fig 10: The legend for the dot looks like it is just connected to OH Pathway, and it should be labelled ast $NO_3^-$ concentration or $[NO_3^-]$ not just $NO_3^-$. Missing a period at end of caption. Subfigures should probably either be all in one column OR the 2018/19 Qingdao be under the 2018 Beijing chart.

**Reply:** Thank you for your valuable feedback. We have updated the legend to clearly label the dots as "$NO_3^-$ concentration" and added a period at the end of the caption. Additionally, we have adjusted the layout by placing the 2018/19 Qingdao chart directly below the 2018 Beijing chart, ensuring a more logical and visually consistent arrangement. This revision can be found in Figure 8 of the revised manuscript.

[Figure]

Figure 8 Time series of the contributions of the atmospheric $NO_3^-$ formation pathways: (a, b) Beijing 2018, (c, d) Beijing 2013, and (e, f) Qingdao 2018, based on (a, c, e) dual-isotope analysis and (b, d, f) model simulations.

432: There is a comma splice in this sentence.

**Reply:** We have revised the sentence to correct this issue. In addition, we have checked the manuscript carefully to ensure proper sentence structure and improve readability in lines 518–520 of the revised version. We appreciate your careful review and valuable feedback.

Revised sentence: This hypothesis could be verified from an emission perspective. The total ammonia emissions in China increased from 9.64 to 9. 75 Tg from 2013–2015 and then gradually decreased to 9.12 Tg by 2018 (Liao et al., 2022).

Fig 13: The scaling seems poor or wrong in the difference map. The HONO concentrations only cover <2 ppb but the scaling on the difference is ±50.

**Reply:** We apologize for the confusion. The differential scale represents the percentage change in HONO concentration in 2018 relative to 2013. For clarity, we have adjusted the scale to ±100% to more accurately depict the range of HONO concentration variations. This revision has been implemented in Figure 11 of the revised manuscript.

[Figure]

Figure 11 Spatial distributions and interannual variations in the GR, HONO, $N_2O_5$, and $NO_x$ concentrations over the North China Plain during the winters of 2013 and 2018. The percentage changes (diff) represent the relative differences between 2018 and 2013.

Supporting Information

**Enhanced Atmospheric Oxidation and Particle Reductions Driving Changes to Nitrate Formation Mechanisms across Coastal and Inland Regions of North China**

Zhenze Liu[1,2], Jianhua Qi[1,2], Yuanzhe Ni[1], Likun Xue[3], Xiaohuan Liu[1,2]

[1] Key Laboratory of Marine Environment and Ecology, Ministry of Education, Ocean University of China, Qingdao 266100, China

[2] Laboratory for Marine Ecology and Environmental Science, Qingdao Marine Science and Technology Center, Qingdao 266237, China

[3] Environment Research Institute, Shandong University, Qingdao, Shandong, 266237, China

*Correspondence to*: Jianhua Qi (qjianhua@ouc.edu.cn), Xiaohuan Liu (liuxh1983@ouc.edu.cn)

**Text S1**

Our isotope blank measurements followed the same procedure as the sample isotope analysis. Specifically, in the sample measurements, after purging with high-purity nitrogen, 20 nmol of nitrogen was added to the headspace vial containing the *Pseudomonas aureofaciens* (ATCC13985) strain. For the blank measurements, no sample was added, and after 24 hours, 10 M NaOH was directly injected to quench the reaction before the analysis. The peak area in the chromatogram represents the absolute amount of $N_2O$ reduced by the strain, and the $\delta^{15}N$ and $\delta^{18}O$ values correspond to the $\delta^{15}N$ and $\delta^{18}O$ values of the sample. The peak area for the samples was around 10, while the peak areas for the two blank measurements were only 0.371 and 0.336, indicating an influence on the isotope values of less than 5%, which is negligible and thus not considered.

[Figure]

**Figure S1** The diurnal values of atmospheric $\delta^{18}O-NO_2$ in Hefei winter (Zhang et al., 2025) (a) and in Nanchang summer (Cao, 2022) (b), and $\delta^{18}O-NO_3^-$ in Tianjin winter (Feng et al., 2020) (c) and in Nanjing winter (Zhang et al., 2022) (d).

**Text S2**

In most studies, the tropospheric $\delta^{15}$N-NOx was often assumed as 0‰ following Walters and Michalski (2016), Luo et al. (2023) and Deng et al. (2024). In addition, the tropospheric $\delta^{18}$O-H$_2$O$_{(g)}$ in Beijing in winter was determined as -27.9‰ in Wen et al. (2010), and in Qingdao, it was determined as -18.6‰ in Wang et al. (2022). The tropospheric $\delta^{18}$O-NOx ranged from 112‰ to 122‰ (Michalski et al., 2014; Walters and Michalski, 2016). The f$_{NO2}$ values in Beijing and Qingdao were 0.655 (Luo et al., 2023) and 0.786 (Lian et al., 2022) in winter, respectively.

$$\delta^{15}N - NO_3^- = \gamma \times [\delta^{15}N - NO_3^-]_{OH} + (1-\gamma) \times [\delta^{15}N - NO_3^-]_{N_2O_5}$$
$$= \gamma \times [\delta^{15}N - HNO_3]_{OH} + (1-\gamma) \times [\delta^{15}N - HNO_3]_{N_2O_5} \tag{S1}$$

$$\delta^{18}O - NO_3^- = \gamma \times [\delta^{18}O - NO_3^-]_{OH} + (1-\gamma) \times [\delta^{18}O - NO_3^-]_{N_2O_5}$$
$$= \gamma \times [\delta^{18}O - HNO_3]_{OH} + (1-\gamma) \times [\delta^{18}O - HNO_3]_{N_2O_5} \tag{S2}$$

$$[\delta^{15}N - HNO_3]_{OH} = \delta^{15}N - NO_2$$
$$= 1000 \times \left[ \frac{\left(^{15}a_{NO_2/NO} - 1\right)\left(1 - f_{NO_2}\right)}{\left(1 - f_{NO_2}\right) + \left(^{15}a_{NO_2/NO} \times f_{NO_2}\right)} \right] + \delta^{15}N - NOx \tag{S3}$$

$$[\delta^{15}N - HNO_3]_{N_2O_5} = 1000 \times \left(^{15}a_{N_2O_5/NO_2} - 1\right) + \delta^{15}N - NOx \tag{S4}$$

$$[\delta^{18}O - HNO_3]_{OH} = \frac{2}{3} \times [\delta^{18}O - NO_2]_{OH} + \frac{1}{3} \times [\delta^{18}O - OH]_{OH}$$
$$= \frac{2}{3} \times \left[ \frac{1000 \times \left(^{18}\alpha_{NO_2/NO} - 1\right) \times \left(1 - f_{NO_2}\right)}{\left(1 - f_{NO_2}\right) + \left(^{18}a_{NO_2/NO} \times f_{NO_2}\right)} + [\delta^{18}O - NOx] \right]$$
$$+ \frac{1}{3} \times \left[ \left(\delta^{18}O - H_2O_{(g)}\right) + 1000 \times \left(^{18}\alpha_{OH/H_2O_{(g)}} - 1\right) \right] \tag{S5}$$

$$[\delta^{18}O - HNO_3]_{N_2O_5} = \delta^{18}O - NO_2 + 1000 \times \left(^{18}\alpha_{N_2O_5/NO_2} - 1\right) \tag{S6}$$

$$1000\left(^m a_{x/y} - 1\right) = \frac{A}{T^4} \times 10^{10} + \frac{B}{T^3} \times 10^8 + \frac{C}{T^2} \times 10^6 + \frac{D}{T} \times 10^4 \tag{S7}$$

**Table S1 Values of $\delta^{18}O$ from atmospheric components**

| Components | Values (‰) | References |
|---|---|---|
| $O_3$ | From 80 to 130 | Michalski et al., 2011 |
| $O_2$ | 23.5 | Kroopnick and Craing; 1972 |
| $H_2O$ (g) in Beijing winter | −27.9 | Wen et al., 2010 |
| $H_2O$ (g) in Qingdao winter | −18.6 | Wang et al., 2022 |
| ·OH in Beijing winter | From −72.4 to −64.9 | $\delta^{18}O\text{-OH} = \delta^{18}O\text{-}H_2O_{(g)} + 1000(^{18}\alpha_{X/Y} - 1)$ |
| ·OH in Qingdao winter | From −61.2 to −57.8 | (Walters and Michalski, 2016) |

**Table S2 $^{15}\alpha_{A/B}$ and $^{18}\alpha_{A/B}$ regression coefficients as a function of the temperature (150 K ≤ T ≤ 450 K) (Walters and Michalski, 2015, 2016)**

|  |  | A | B | C | D |
|---|---|---|---|---|---|
| $^{15}\alpha_{A/B}$ | $N_2O_5/NO_2$ | 0.69398 | -1.9859 | 2.3876 | 0.16308 |
|  | $NO_2/NO$ | 3.8834 | -7.7299 | 6.0101 | -0.17928 |
| $^{18}\alpha_{A/B}$ | $NO/NO_2$ | -0.04129 | 1.1605 | -1.8829 | 0.74723 |
|  | $\cdot OH/H_2O_{(g)}$ | 2.1137 | -3.8026 | 2.5653 | 0.5941 |
|  | $N_2O_5/NO_2$ | -0.54136 | 0.13073 | 1.2477 | -0.1272 |

**Table S3 Equations for calculating the statistical evaluation indices**

| Statistical index | Formula |
|---|---|
| 1. Mean Bias | $MB = \dfrac{1}{N}\sum_{1}^{N}\left(\text{Sim} - \text{Obs}\right)$ |
| 2. Root Mean Square Error | $RMSE = \sqrt{\dfrac{1}{N}\sum_{1}^{N}(Sim - Obs)^2}$ |
| 3. Index of agreement, IOA | $IOA = 1 - \dfrac{\sum_{i=1}^{N}(Sim - Obs)^2}{\sum_{i=1}^{N}(|Sim - \overline{Obs}| + |Obs - \overline{Obs}|)^2}$ |
| 4. Normalized Mean Bias | $NMB = \dfrac{1}{N}\sum_{1}^{N}\left(\dfrac{Sim - Obs}{Obs}\right)$ |
| 5. Normalized Mean Error | $NME = \dfrac{1}{N}\sum_{1}^{N}\left|\dfrac{Sim - Obs}{Obs}\right|$ |
| 6. Correlation coefficient (R) | $R = \dfrac{1}{N}\sum_{i=1}^{N}\left[\dfrac{(Sim - \overline{Sim})(Obs - \overline{Obs})}{S_p S_o}\right]$

 $S_P = [\dfrac{1}{N}\sum_{i=1}^{N}(Sim - \overline{Sim})^2]^{\frac{1}{2}}$

 $S_o = [\dfrac{1}{N}\sum_{i=1}^{N}(Obs - \overline{Obs})^2]^{\frac{1}{2}}$ |

**Table S4 Sources of nitrate observation data for the winter of 2013 and the winter of 2018 in the NCP**

| City | Winter, 2013 | Winter, 2018 |
|---|---|---|
| Beijing | Song et al. (2019) | Fan et al. (2020) |
| Tianjin | Yao et al. (2020) | Observation |
| Shijiahzuang | Wang et al. (2016) | Zhou et al. (2020) |
| Jinan | Cheng et al. (2021) | Observation |
| Zhengzhou | Wei et al. (2019) | Dong et al. (2020) |
| Qingdao | Observation | Observation |
| Yantai | / | / |

The $NO_3^-$ observation data collected during the winter of 2018 for Tianjin were sourced from direct observations by the group of Li Xiaodong at Tianjin University (sampling site: Building 19 rooftop, Tianjin University; coordinates: 39.11°N, 117.16°E). The $NO_3^-$ observation data collected during the winter of 2018 in Jinan were sourced from observations by the group of Xue Likun at Shandong University (sampling site: Jinan City Environmental Monitoring Station; coordinates: 36.66°N, 117.05°E). For Qingdao, $NO_3^-$ observation data for both the winter of 2018 and the winter of 2013 were derived from our own observations.

[Figure]

**Figure S2 Concentrations of PM₂.₅ and its components in seven major cities in the NCP region during the winters of 2013 and 2018**

[Figure]

**Figure S3 Spatial distribution of the NO₂/OH molar ratio in the NCP region during the winters of 2013 and 2018**

[Figure]

**Figure S4 Relative humidity in the NCP and seven major cities (2013, 2018)**

**Reference**

Cheng, M., Tang, G., Lv, B., Li, X., Wu, X., Wang, Y., and Wang, Y.: Source apportionment of $PM_{2.5}$ and visibility in Jinan, China, Journal of Environmental Sciences, 102, 207-215, doi:10.1016/j.jes.2020.09.012, 2021.

Deng, M., Wang, C., Yang, C., Li, X., and Cheng, H.: Nitrogen and oxygen isotope characteristics, formation mechanism, and source apportionment of nitrate aerosols in Wuhan, Central China, Science of The Total Environment, 921, 170715, doi:10.1016/j.scitotenv.2024.170715, 2024.

Dong, Z., Su, F., Zhang, Z., and Wang, S.: Observation of chemical components of $PM_{2.5}$ and secondary inorganic aerosol formation during haze and sandy haze days in Zhengzhou, China, J Journal of Environmental Sciences, 88, 316-325, doi:10.1016/j.jes.2019.09.016, 2020.

Fan, M.-Y., Zhang, Y.-L., Lin, Y.-C., Cao, F., Zhao, Z.-Y., Sun, Y., Qiu, Y., Fu, P., and Wang, Y.: Changes of Emission Sources to Nitrate Aerosols in Beijing After the Clean Air Actions: Evidence From Dual Isotope Compositions, Journal of Geophysical Research: Atmospheres, 125, e2019JD031998, doi:10.1029/2019JD031998, 2020.

Lian, C., Wang, W., Chen, Y., Zhang, Y., Zhang, J., Liu, Y., Fan, X., Li, C., Zhan, J., Lin, Z., Hua, C., Zhang, W., Liu, M., Li, J., Wang, X., An, J., and Ge, M.: Long-term winter observation of nitrous acid in the urban area of Beijing, Journal of Environmental Sciences, 114, 334-342, doi:10.1016/j.jes.2021.09.010, 2022.

Luo, L., Wu, S., Zhang, R., Wu, Y., Li, J., and Kao, S.-j.: What controls aerosol δ15N-NO3−? NOx emission sources vs. nitrogen isotope fractionation, Science of The Total Environment, 871, 162185, doi:10.1016/j.scitotenv.2023.162185, 2023.

Michalski, G., Bhattacharya, S. K., and Girsch, G.: NOx cycle and the tropospheric ozone isotope anomaly: an experimental investigation, Atmos. Chem. Phys., 14, 4935-4953, doi:10.5194/acp-14-4935-2014, 2014.

Song, W., Wang, Y.-L., Yang, W., Sun, X.-C., Tong, Y.-D., Wang, X.-M., Liu, C.-Q., Bai, Z.-P., and Liu, X.-Y.: Isotopic evaluation on relative contributions of major NOx sources to nitrate of PM2.5 in Beijing, Environmental Pollution, 248, 183-190, doi:10.1016/j.envpol.2019.01.081, 2019.

Walters, W. W. and Michalski, G.: Theoretical calculation of nitrogen isotope equilibrium exchange fractionation factors for various NOy molecules, Geochimica et Cosmochimica Acta, 164, 284-297, doi:10.1016/j.gca.2015.05.029, 2015.

Walters, W. W. and Michalski, G.: Theoretical calculation of oxygen equilibrium isotope fractionation factors involving various NOy molecules, OH, and $H_2O$ and its implications for isotope variations in atmospheric nitrate, Geochimica et Cosmochimica Acta, 191, 89-101, doi:10.1016/j.gca.2016.06.039, 2016.

Wang, X., Zhou, Y., Cheng, S., and Wang, G.: Characterization and regional transmission impact of water-soluble ions in $PM_{2.5}$ during winter in typical cities, China Environmental Science, 36, 2289-2296, 2016.

Wang, Y., Cui, B.-l., Li, D.-s., Wang, Y.-x., Yu, W.-x., and Zong, H.-h.: Stable Isotopes Reveal Water Vapor Sources of Precipitation over the Jiaolai Plain, Shandong Peninsula, China, Asia-Pacific Journal of Atmospheric Sciences, 58, 227-241, doi:10.1007/s13143-021-00253-2, 2022.

Wei, X., Gao, m., and Tong, j.: Characterization of water-soluble ions of $PM_{2.5}$ in Zhengzhou, Chinese journal of quantum, 36, 495-499, 2019.

Wen, X.-F., Zhang, S.-C., Sun, X.-M., Yu, G.-R., and Lee, X.: Water vapor and precipitation isotope ratios in Beijing, China, Journal of Geophysical Research: Atmospheres, 115, doi:10.1029/2009JD012408, 2010.

Yao, Q., Liu, Z., Han, S., Cai, Z., Liu, J., Hao, T., Liu, J., Huang, X., and Wang, Y.: Seasonal variation and secondary formation of size-segregated aerosol water-soluble inorganic ions in a coast megacity of North China Plain, Environmental Science and Pollution Research, 27, 26750-26762, doi:10.1007/s11356-020-09052-0, 2020.

Zhou, J., Duan, J., Wang, J., Liu, H., LI, M., and Jin, W.: Analysis of pollution characteristics and sources of $PM_{2.5}$ during heavey pollution in Shijiazhuang city around New Year's Day 2019, Environmental Science, 41, 39-49, doi:10.13227/j.hjkx.201906085, 2020.

Cao, L. Nitrogen and oxygen isotope tracing of urban atmospheric nitrate sources and atmospheric processes—taking Beijing and Nanchang as examples. Dissertation for master's degree of East China University of Technology.

Feng, X., Li, Q., Tao, Y., Ding, S., Chen, Y., and Li, X.: Impact of Coal Replacing Project on atmospheric fine aerosol nitrate loading and formation pathways in urban Tianjin: Insights from chemical composition and [15]N and [18]O isotope ratios, Sci. Total Environ., 708, 134797, 10.1016/j.scitotenv.2019.134797, 2020.

Zhang, Y., Zhang, W., Fan, M., Li, J., Fang, H., Cao, F., Lin, Y., Wilkins, B. P., Liu, X., Bao, M., Hong, Y., and Michalski, G.: A diurnal story of $\Delta^{17}O(NO_3^-)$ in urban Nanjing and its implication for nitrate

aerosol formation, npj Climate and Atmospheric Science, 5, 10.1038/s41612-022-00273-3, 2022.

Zhang, Z., Zhou, T., Jiang, Z., Ma, T., Su, G., Ruan, X., Wu, Y., Cao, Y., Wang, X., Liu, Z., Li, W., Zhang, H., Lin, M., Liu, P., and Geng, L.: High-Resolution Measurements of Multi-Isotopic Signatures ($\delta^{15}N$, $\delta^{18}O$, and $\Delta^{17}O$) of Winter $NO_2$ in a Megacity in Central China, Environ. Sci. Technol., 59, 3634-3644, 10.1021/acs.est.4c07724, 2025.

Kroopnick, P., and Craig, H.: Atmospheric Oxygen: Isotopic Composition and Solubility Fractionation, Science, 175, 54-55, 10.1126/science.175.4017.54, 1972.

Michalski, G., Bhattacharya, S. K., and Girsch, G.: NOx cycle and the tropospheric ozone isotope anomaly: an experimental investigation, Atmos. Chem. Phys., 14, 4935-4953, 10.5194/acp-14-4935-2014, 2014.

Michalski, G., Bhattacharya, S. K., DF Mase., 2011. Oxygen isotope dynamics of atmospheric nitrate and its precursor molecules. Springer Berlin Heidelberg.

Walters, W. W., and Michalski, G.: Theoretical calculation of oxygen equilibrium isotope fractionation factors involving various NO molecules, OH, and $H_2O$ and its implications for isotope variations in atmospheric nitrate, Geochim. Cosmochim. Ac., 191, 89-101, 10.1016/j.gca.2016.06.039, 2016.